# Fibroblastic reticular cells direct the initiation of T cell responses via CD44

Xavier Y. X. Sng[1], Valentina Voigt[1,2], Iona S. Schuster[1,2], Peter Fleming[1,2], Felix A. Deuss[3,10], Mohammed H. Abuwarwar[3], Serani L. H. van Dommelen[2], Georgia E. G. Neate[1], Riley M. Arnold[1], Harry L. Horsnell[4], Sheridan Daly[2], Bagher Golzarroshan[3], Antiopi Varelias[5], Stewart D. Lyman[6], Anthony A. Scalzo[2], Geoffrey R. Hill[7,8], Scott N. Mueller[4], Matthew E. Wikstrom[2], Richard Berry[3,10], Jamie Rossjohn[3,9], Anne L. Fletcher[3], Christopher E. Andoniou[1,2,11 ✉] & Mariapia A. Degli-Esposti[1,2,11 ✉]

The movement of dendritic cells and T cells within secondary lymphoid organs is critical for the development of adaptive immune responses[1,2]. Central to this process is the fibroblastic reticular cell (FRC) network, which forms a highly organized conduit system that facilitates the movement of and interactions between dendritic cells and T cells[3–6]. Previous studies have partly characterized how FRCs support these interactions[7,8]. However, the molecular mechanisms that operate under physiological conditions remain unknown. Here we show that the viral protein m11, encoded by the herpesvirus murine cytomegalovirus (CMV), inhibits antiviral immunity by targeting the FRC network and interfering with a critical function of cellular CD44. We found that m11 binds to CD44 and established that m11 perturbs the molecular interactions of CD44 with its natural ligand, hyaluronic acid. The interaction of m11 with CD44 impairs the trafficking of dendritic cells within the spleen, thereby impeding efficient priming of naive T cells and the initiation of antiviral CD8 T cell responses. The targeting of CD44 by CMV reveals CD44 as a molecule that is essential to the functioning of the FRC network and uncovers a previously unrecognized stroma-based mechanism that is critical for the generation of effective T cell responses.

The regulation of immune responses is remarkably precise, especially within the adaptive T cell compartment that is central to antiviral defence. To initiate these responses, multiple signals must be integrated in secondary lymphoid organs such as the lymph nodes and spleen[5,9]. These tissues contain a structured stromal cell network that supports interactions between antigen presenting cells and T cells[4]. Fibroblastic reticular cells (FRCs) are central to this network and provide the structural and functional cues that support effective T cell responses[3,5]. They facilitate the migration and interaction of immune cells, with podoplanin having a significant role[7,8]. Additional cellular receptors may also contribute, although experimental evidence is lacking.

Viruses hijack cellular functions to their advantage. Over millennia of co-evolution, viruses, and cytomegaloviruses (CMVs) in particular, have evolved mechanisms to subvert host immunity, including the expression of proteins that bind and modify the functions of cellular receptors[10,11]. Here we used CMV infection to examine viral–host protein interactions as a strategy to identify novel cellular functions and processes.

## A viral ligand for cellular CD44

We used an expression cloning approach to identify the cellular binding partners of CMV proteins. Viral proteins expressed on infected cells are of particular interest, as they may engage cellular receptors to promote immune evasion. m11 is a predicted 299-amino-acid type I transmembrane protein encoded by murine CMV (MCMV) (GenBank: CAP08055.1), with a 212-amino-acid extracellular domain and a 66-amino-acid cytoplasmic domain (Fig. 1a), whose function is unknown. To examine m11 expression, we generated specific monoclonal antibodies (M-627 and 7G5; Methods). m11 was detected on the surface of MCMV-infected cells (Fig. 1b,c), indicating that it may engage cellular receptors and function as an immune evasion molecule.

To identify cellular proteins that bind m11, we screened cDNA expression libraries from diverse mouse cell lines and tissues with m11–Fc, a fusion protein comprising the extracellular domain of m11 (amino acids 1–212) fused to a human IgG1 Fc domain, and identified a cDNA encoding mouse CD44, a transmembrane glycoprotein involved in

[1]Infection and Immunity Program and Department of Microbiology, Biomedicine Discovery Institute, Monash University, Clayton, Victoria, Australia. [2]Centre for Experimental Immunology, Lions Eye Institute, Nedlands, Western Australia, Australia. [3]Infection and Immunity Program and Department of Biochemistry and Molecular Biology, Biomedicine Discovery Institute, Monash University, Clayton, Victoria, Australia. [4]Department of Microbiology and Immunology, The University of Melbourne, The Peter Doherty Institute for Infection and Immunity, Melbourne, Victoria, Australia. [5]QIMR Berghofer Medical Research Institute, Herston, Queensland, Australia. [6]Lyman BioPharma Consulting, Seattle, WA, USA. [7]Translational Science and Therapeutics Division, Fred Hutchinson Cancer Center, Seattle, WA, USA. [8]Division of Medical Oncology, University of Washington, Seattle, WA, USA. [9]Institute of Infection and Immunity, Cardiff University, School of Medicine, Cardiff, UK. [10]Present address: Audax Biosciences, Melbourne, Victoria, Australia. [11]These authors contributed equally: Christopher E. Andoniou, Mariapia A. Degli-Esposti. ✉e-mail: chris.andoniou@monash.edu; mariapia.degli-esposti@monash.edu

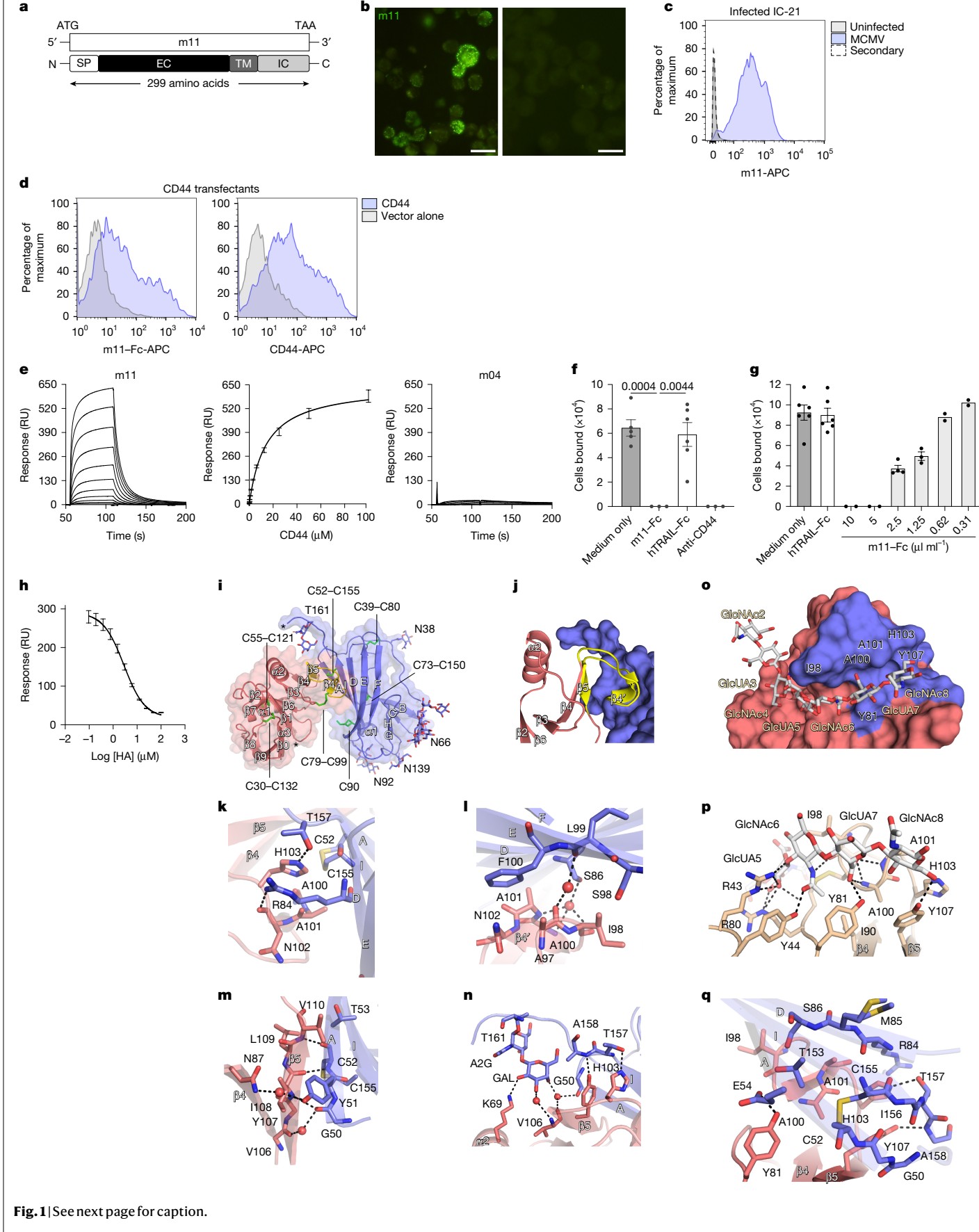

**Fig. 1 | See next page for caption.**

**Fig. 1 | The m11 protein is expressed on the surface of MCMV-infected cells and binds CD44. a**, Schematic of m11 domains. SP, signal peptide; EC, extracellular domain; TM, transmembrane domain; IC, intracellular domain. **b**, Unpermeabilized IC-21 cells infected with MCMV (multiplicity of infection (MOI) = 10) (left) or uninfected (right) were stained with anti-m11. Scale bars, 20 μm. **c**, Histograms of IC-21 stained with anti-m11. **d**, Histograms of transfected COS-7 cells stained with m11–Fc or anti-CD44. Data in **b–d** are representative of three independent experiments. **e**, SPR showing CD44 binding to immobilized m11 or m04. $K_d$ was calculated from two independent experiments. RU, resonance units. **f**, Hyaluronic acid binding to CD44 on EL4 T cells activated with PMA plus ionomycin and incubated with m11–Fc or control human TRAIL–Fc fusion (hTRAIL–Fc) (10 μg ml⁻¹). Anti-CD44 (KM114, 10 μg ml⁻¹) served as a control. Graphs show mean ± s.e.m.; significance tested by two-sided *t*-test (medium only, $n = 5$; hTRAIL–Fc, $n = 6$; m11–Fc and anti-CD44, $n = 3$). **g**, Inhibition of hyaluronic acid binding to EL4 cells activated with PMA plus ionomycin and pre-incubated with serial dilutions of crosslinked m11–Fc, control Fc or medium only (medium only, $n = 6$; hTRAIL–Fc, $n = 6$; m11–Fc 10 and 5 μg ml⁻¹, $n = 2$; 5 μg ml⁻¹, $n = 2$;

2.5 μg ml⁻¹, $n = 4$; 1.25 μg ml⁻¹, $n = 3$; 0.62 μg ml⁻¹, $n = 2$; 0.31 μg ml⁻¹, $n = 2$). **h**, Inhibition curve showing binding of 12.5 μM CD44 to immobilized m11 in the presence of 0–100 μM hyaluronic acid (HA). Average of two independent experiments, performed in duplicate. **i**, CD44 (salmon) bound to m11 (blue); CD44 hook and β5-strand are highlighted (yellow). Glycans are shown as sticks, cysteine residues are shown as green sticks and asterisks indicate C termini. **j**, CD44 hook engaging m11 (blue). **k–n**, Close-up views of the m11–CD44 interface. Black dashed lines indicate hydrogen bonds and red spheres represent water molecules. Overlap between the m11- and hyaluronic acid-binding sites on CD44. **o**, The CD44–hyaluronic acid complex (Protein Data Bank (PDB) ID: 2JCR) showing CD44 (solid surface) and hyaluronic acid (grey sticks). CD44 residues contacting m11 are in blue; those contacting both m11 and hyaluronic acid are in black. **p**, Hyaluronic acid interactions with CD44. **q**, The m11–CD44 interface in the same orientation as in **p**, showing hyaluronic acid-binding CD44 residues (salmon sticks) and interactions with m11 (blue sticks). Black dashed lines indicate hydrogen bonds.

cell adhesion and signalling[12]. Binding of m11 to mouse CD44 was confirmed in transfected COS-7 cells (Fig. 1d).

The binding affinity of m11 for mouse CD44 was measured by surface plasmon resonance (SPR). The CD44 ectodomain (residues 23–174) bound m11 (residues 28–164) with a dissociation constant ($K_d$ (± s.e.m)) of 14.3 ± 0.3 μM (Fig. 1e) but showed no binding to m04[13], confirming specificity (Fig. 1e). These results indicate that m11 directly engages the CD44 ectodomain, including the hyaluronic acid-binding Link module[14], independent of other cellular or viral components.

## m11 blocks hyaluronic acid binding to CD44

The principal ligand for CD44 is hyaluronic acid[15], a linear glycosaminoglycan. CD44–hyaluronic acid interactions regulate adhesion, migration and proliferation[16], but their role in antiviral immunity remains unclear. We therefore examined whether m11 affects CD44 function by competing with hyaluronic acid. CD44 on the surface of EL4 cells binds hyaluronic acid[14] when the cells are activated with phorbol myristate acetate (PMA) and ionomycin. Activated EL4 cells showed significantly reduced adhesion to hyaluronic acid-coated plates in the presence of m11–Fc (Fig. 1f). A similar reduction occurred in the presence of anti-CD44 KM114 monoclonal antibody, which masks the hyaluronic acid-binding site, whereas an irrelevant human TRAIL (hTRAIL)–Fc fusion had no effect (Fig. 1f). The blocking effect of m11–Fc was dose-dependent (Fig. 1g). In a reciprocal experiment, we tested whether hyaluronic acid could block the m11–CD44 interaction. Binding of a fixed concentration of CD44 (12.5 μM) to immobilized m11 decreased to 10–15% of the maximal response after incubation with 50–100 μM hyaluronic acid (Fig. 1h). Together these data indicate that m11 and hyaluronic acid bind CD44 in a mutually exclusive manner. By acting as a competitive inhibitor of CD44–hyaluronic acid interactions, m11 is expected to interfere with CD44-mediated functions.

## The m11–CD44 structure shows unique docking

We solved the structure of the soluble m11–CD44 complex (Extended Data Table 1), in which the CD44 fold within the complex was similar to that of the unliganded receptor[17] (Fig. 1i). m11 engaged CD44 through a flattened surface that interacted primarily with a 'hook-like' region of CD44 (Fig. 1j). This m11–CD44 contact zone formed three distinct groups of interacting residues. The primary interaction zone was focused around the apex of the CD44 hook. Arg84 of m11 had a central role, whereby it made extensive interactions with Ala101, Asn102 and His103 of CD44 (Fig. 1k). These interactions were further supported by a water-mediated hydrogen-bonding network (Fig. 1l). Additional interactions with the CD44 hook included Ser98 and Phe100 of m11,

which flanked Ala97 of CD44 (Fig. 1l). The second interaction zone was centred around the base of the CD44 hook (Fig. 1m). Comparison with the structure of unliganded CD44[17] revealed that this region becomes flattened by m11, indicative of an induced-fit mode of interaction (Extended Data Fig. 1). The third cluster of interacting residues included Thr157, which anchored the CD44 hook to the base of the β5-strand of m11 (Fig. 1n). Thus, m11 uses a three-pronged strategy to specifically target the hook region of CD44.

The blocking data indicate that binding of m11 and hyaluronic acid to CD44 are mutually exclusive, suggesting that the two ligands might compete for the same binding site. Hyaluronic acid binds to CD44 within a shallow groove located between the hook region and the β5–β6 loop (Fig. 1o). Four of the sugar residues (GlcUA5–GlcNAc8) make the majority of the interactions with CD44, while the remainder of the glycan chain projects away into the solvent. The CD44–hyaluronic acid interaction is dominated by an extensive interaction network with the predominantly aliphatic residues that line the hook and a stretch of aromatic and basic amino acids located at the base of the groove (Fig. 1p). Notably, 6 of the 14 CD44 residues that bind to hyaluronic acid also participate in interactions with m11 (Fig. 1q). Of these, the majority are located within the CD44 hook (Ile98, Ala100, Ala101 and His103) or the β5-strand (Tyr107). Comparison of the CD44–hyaluronic acid and m11–CD44 structures indicates that m11 would physically clash with three of the four sugar residues that mediate CD44 binding, thereby providing a rationale for how m11 blocks the CD44–hyaluronic acid interaction. Although the hyaluronic acid and m11 binding sites overlap considerably, the nature of the interactions is distinct, suggesting that m11 does not function by molecular mimicry. We therefore refer to m11 as viral CD44 binding protein (vCD44BP).

## vCD44BP helps the virus evade CD8 T cells

Having established that vCD44BP binds CD44 and blocks CD44–hyaluronic acid interactions, we examined whether this activity benefits the virus. We constructed an MCMV mutant with a premature stop codon in the m11 open reading frame (ΔvCD44BP). The ΔvCD44BP virus replicated in permissive cell types with kinetics that were indistinguishable from those of the parent virus (Extended Data Fig. 2), indicating that vCD44BP is dispensable for viral entry and replication in vitro.

We next examined the role of vCD44BP in vivo by comparing viral loads after infection with wild-type MCMV, the ΔvCD44BP mutant, and a revertant virus in which vCD44BP expression was restored (REV). Viral loads in the spleen were equivalent for all three viruses at 2 and 4 days post-infection (dpi) (Fig. 2a). Thereafter, the ΔvCD44BP virus showed significantly reduced loads compared with wild-type and REV viruses. Thus, although vCD44BP does not affect viral replication directly, it is required to sustain viral loads in vivo.

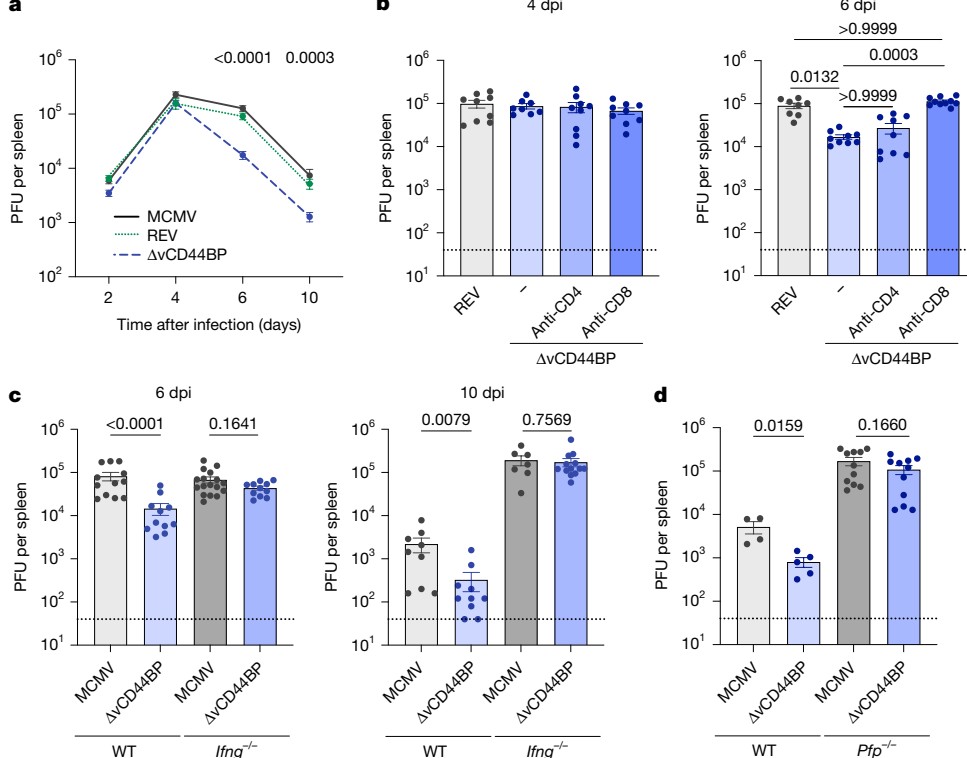

**Fig. 2 | vCD44BP attenuates viral clearance by interfering with CD8 T cell functions. a**, Viral loads in spleens of BALB/c mice infected with wild-type (MCMV), ΔvCD44BP or ΔCD44BP-REV (REV) MCMV viruses (MCMV: 2 dpi, $n = 11$; 4 dpi, $n = 14$; 6 dpi, $n = 13$; 10 dpi, $n = 13$; ΔvCD44BP: 2 dpi $n = 11$; 4 dpi, $n = 14$; 6 dpi, $n = 15$; 10 dpi, $n = 20$; REV: 2 dpi, $n = 12$; 4 dpi, $n = 8$; 6 dpi, $n = 10$; 10 dpi, $n = 10$). PFU, plaque-forming units. **b**, Viral loads in spleens of BALB/c mice depleted of CD4 (with GK1.5) or CD8 (with 53.5.8) T cells and infected with REV or ΔvCD44BP (4 dpi: REV, anti-CD4 and anti-CD8, $n = 9$; undepleted (−), $n = 8$; 6 dpi: REV, $n = 8$; undepleted and anti-CD4, $n = 9$; anti-CD8, $n = 10$). **c**, Viral loads in spleens of BALB/c (wild-type (WT)) and BALB/c $Ifng^{-/-}$ mice infected

with MCMV or ΔvCD44BP (6 dpi: MCMV in WT, $n = 12$; ΔvCD44BP in WT, $n = 11$; MCMV in $Ifng^{-/-}$, $n = 17$; ΔvCD44BP in $Ifng^{-/-}$, $n = 11$; 10 dpi: MCMV in WT, $n = 9$; ΔvCD44BP in WT, $n = 10$; MCMV in $Ifng^{-/-}$, $n = 7$; ΔvCD44BP in $Ifng^{-/-}$, $n = 13$). Data in **a**–**c** are pooled from at least three independent experiments. **d**, Viral loads at 6 dpi in spleens of BALB/c wild-type and $Pfp^{-/-}$ mice infected with MCMV or ΔvCD44BP; pooled from two independent experiments (MCMV in WT $n = 4$; ΔvCD44BP in WT, $n = 5$; MCMV in $Pfp^{-/-}$ and ΔvCD44BP in $Pfp^{-/-}$, $n = 11$). Mice were infected with $1 \times 10^4$ PFU, except in **d**, where $2 \times 10^3$ PFU was used. Graphs show mean ± s.e.m.; significance tested by two-sided Mann–Whitney test, or Kruskal–Wallis (in **b** only).

The timing of reduced ΔvCD44BP viral loads suggested a role for adaptive immune responses, which limit infection starting from 6 dpi[18]. As expected, depleting CD4 or CD8 T cells did not affect viral loads at 4 dpi (Fig. 2b, left). By 6 dpi, however, CD8 T cell depletion rendered the ΔvCD44BP viral loads equivalent to those of the REV virus, eliminating the improved control of ΔvCD44BP (Fig. 2b, right). CD4 T cell depletion had no effect, and improved control of the ΔvCD44BP virus was still observed (Fig. 2b, right).

CD8 T cells limit infection via secretion of the antiviral cytokine IFNγ and the release of cytotoxic granzymes and perforin (encoded by *Prf1* (also known as *Pfp*))[18]. Consistent with the role of these molecules, in IFNγ- or perforin-deficient mice, the ΔvCD44BP virus replicated to wild-type levels at 6 and 10 dpi (Fig. 2c,d). In perforin-deficient mice, viral replication could not be assessed beyond 6 dpi as mice succumb to infection[18,19]. These data indicate that vCD44BP impairs viral control by affecting antiviral CD8 T cell responses.

## CD44–vCD44BP impairs dendritic cell migration

To determine how vCD44BP affects antiviral CD8 T cell responses, we examined CD8 T cells using flow cytometry (Extended Data Fig. 3a,b). Total CD8 T cells (Fig. 3a) and virus-specific (IE1-tetramer⁺) CD8 T cell numbers (Fig. 3b) were significantly higher in ΔvCD44BP-infected mice. Analysis of effector and memory potential[20] of virus-specific CD8 T cells showed that proportionally ΔvCD44BP infection generated fewer memory precursor effector cells (MPECs; CD127^hi) and more

short-lived effector cells (SLECs; CD127^low) than wild-type MCMV infection (Fig. 3c and Extended Data Fig. 3c). SLEC numbers in spleens were significantly higher following ΔvCD44BP infection (Fig. 3d), whereas MPEC numbers did not differ across infections (Fig. 3d). Accordingly, total and virus-specific CD8 T cell numbers at later times (40 and 120 dpi) were equivalent between wild-type and ΔvCD44BP MCMV infections (Extended Data Fig. 3d–g). Thus, vCD44BP affects the generation of effector cells, suggesting that CD8 T cell priming is impaired when vCD44BP interferes with CD44-mediated functions.

Effective CD8 T cell priming requires interactions between naive T cells and dendritic cells. As reported[21,22], numbers of conventional type 1 dendritic cells (cDC1s) and conventional type 2 dendritic cells (cDC2s) (the two major dendritic cell subsets) declined after MCMV infection, but this reduction was equivalent in wild-type and ΔvCD44BP infections (Fig. 3e and Extended Data Fig. 4a,b), and thus independent of vCD44BP. Similarly, expression of antigen presentation and co-stimulatory proteins on dendritic cells did not differ between infections (Extended Data Fig. 4c). We therefore examined whether dendritic cell localization was affected. As in our infection model priming occurs in the spleen, we assessed the distribution of dendritic cells in the red pulp and white pulp, the key functional and structural compartments of the spleen. A short-pulse intravascular (IV) injection of anti-CD11c antibody was used to distinguish dendritic cells in the red pulp (IV⁺) from those in the white pulp[23,24], and a stromal dissociation protocol facilitated the separation of lymphocytes from stroma. The percentage of dendritic cells within the white pulp (IV⁻) was significantly

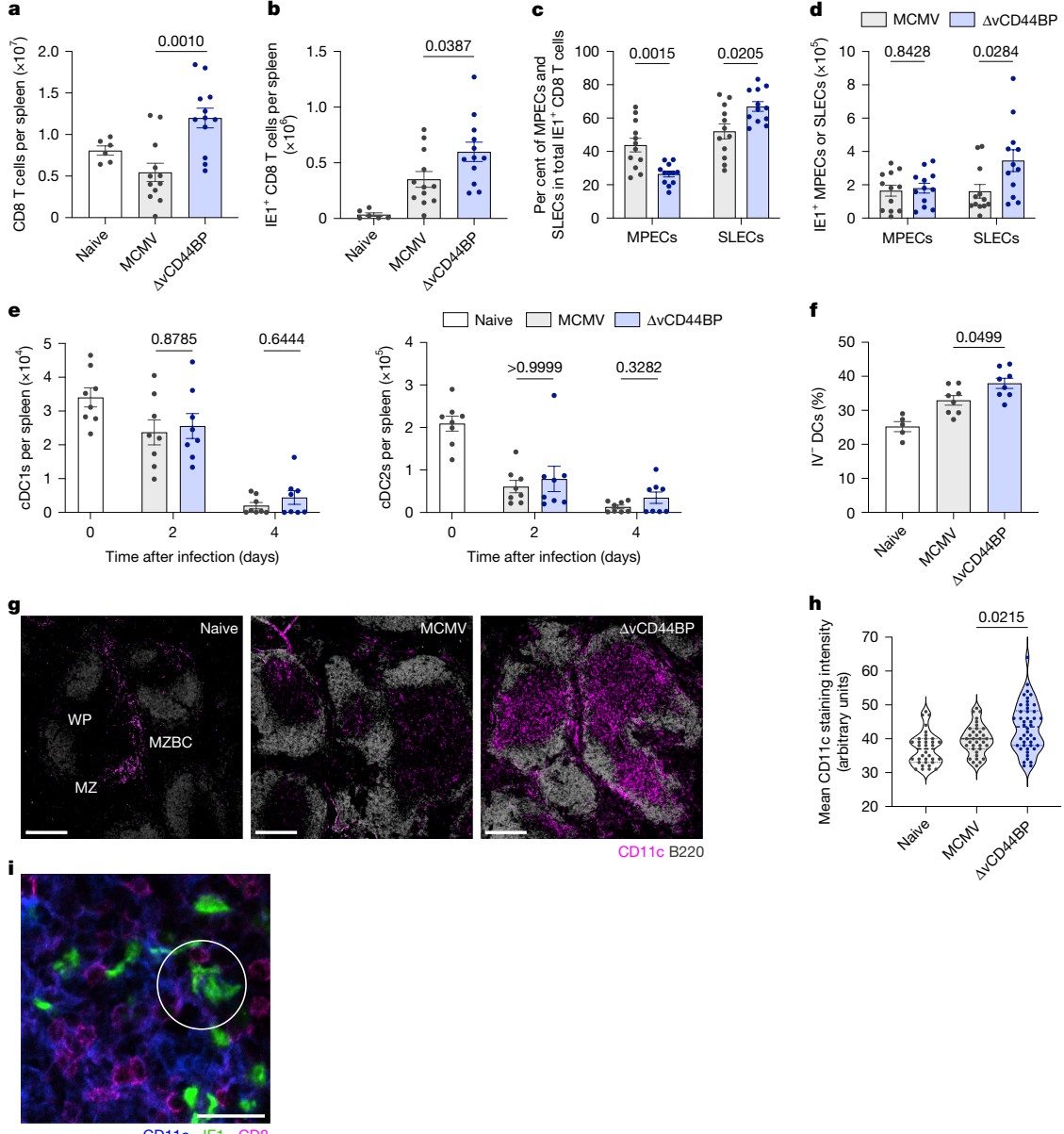

**Fig. 3 | vCD44BP impairs antiviral CD8 T cell responses by interfering with dendritic cell migration. a–d**, Numbers of CD8 T cells (**a**), virus-specific IE1-tetramer⁺ CD8 T cells (**b**) and percentage (**c**) and number (**d**) of MPEC (CD127^hi) and SLEC (CD127^low) IE1⁺ virus-specific CD8 T cells in spleens of uninfected mice or mice infected with wild-type or ΔvCD44BP MCMV (7 dpi). Data in **a**–**d** are pooled from three independent experiments (naive, *n* = 6; MCMV and ΔvCD44BP, *n* = 12). **e**, Numbers of cDC1s and cDC2s in spleens of uninfected mice or mice infected with MCMV or ΔvCD44BP. Data are pooled from two independent experiments (*n* = 8 per group). **f**, Frequency of IV⁻ dendritic cells in spleens of uninfected mice or mice at 36 h after infection with MCMV or ΔvCD44BP. Data are pooled from two independent experiments (naive, *n* = 5; MCMV and ΔvCD44BP, *n* = 8). **g**, Spleen

sections from uninfected mice and mice infected with MCMV or ΔvCD44BP (2 dpi). ImageJ images processed with Imaris (for display purposes only) are shown. Scale bars, 250 μm. WP, white pulp; MZ, marginal zone; MZBC, marginal zone bridging channel. **h**, Mean CD11c staining intensities in T cell zones at 2 dpi. **i**, Representative confocal image showing CD8, IE1 and CD11c in the splenic white pulp of MCMV-infected mice (2 dpi). The white circle highlights an area in which MCMV-infected IE1⁺ cells are in close contact with CD11c⁺ dendritic cells and CD8⁺ T cells. The image is representative of data from three independent experiments. Scale bar, 20 μm. All mice were infected with 5 × 10³ PFU. Graphs show mean ± s.e.m.; violin plots show median and quartiles; significance tested by two-sided Mann–Whitney test.

higher in spleens from ΔvCD44BP-infected mice compared with those infected with wild-type MCMV (Fig. 3f and Extended Data Fig. 4d). Since lymphocyte isolation methods do not recover all cells from tissues[23], we confirmed dendritic cell localization by microscopy. As expected, in naive mice, dendritic cells localized to the marginal zone (at the red pulp–white pulp interface) and bridging channels (Fig. 3g, left). Once activated, dendritic cells migrate into the white pulp to present antigen to naive T cells[24]. After infection, CD11c staining increased in the white pulp (Fig. 3g, middle and right), consistent with dendritic cell migration into the T cell zone. Staining was significantly more

prominent in the white pulp of mice infected with ΔvCD44BP virus, indicating increased dendritic cell migration (Fig. 3g,h and Extended Data Fig. 4e). In BALB/c mice, where acute infection is not limited by Ly49H-dependent natural killer (NK) cell responses, cDC1s and cDC2s are crucial to generate protective CD8 T cell responses[25–28]. Since total dendritic cell numbers did not change, but more dendritic cells, including cDC1s and cDC2s (Extended Data Fig. 4f–h), localized to the white pulp in ΔvCD44BP-infected mice, we conclude that vCD44BP impairs dendritic cell recruitment into the splenic white pulp, thereby limiting dendritic cell–T cell interactions. This is likely to contribute to the

increased numbers of virus-specific SLEC CD8 T cells observed after ΔvCD44BP infection. Accordingly, at 2 dpi, MCMV-infected cells in the white pulp closely associated with dendritic cells, which, in turn, were closely associated with CD8 T cells (Fig. 3i).

## vCD44BP disrupts FRC–dendritic cell interactions

Dendritic cells trafficking within lymphoid organs is directed by a highly organized network of fibroblastic stromal cells, including FRCs, which form the scaffold guiding dendritic cell and T cell migration[29]. MCMV infects stromal cells, including FRCs[30–32], and alters lymphocyte organization within the spleen[30,31]. We confirmed infection of stromal cells, including FRCs, both in vitro (Fig. 4a) and in vivo (Fig. 4b,c and Extended Data Fig. 5a–d). In vivo, T cell zone reticular cells comprised the majority (around 50%) of infected FRCs (Extended Data Fig. 5c,d). Notably, these analyses identified strain-specific phenotypic differences in FRC subsets, with red pulp reticular cells (RPRCs) from BALB/c mice lacking Ly6C, an identifier of RPRCs in C57BL/6[33] (Extended Data Fig. 5b). Although stromal cells in both the red pulp and white pulp were infected, those in the white pulp, which constitute a small portion of the overall fibroblastic population, were preferentially targeted by MCMV (Extended Data Fig. 5e). Infected white pulp FRCs, identified by podoplanin expression, were also detected and quantified in situ in infected spleens (Fig. 4d,e and Extended Data Fig. 5f,g).

Next, we measured CD44 expression in an FRC line and in primary splenic FRCs. CD44 expression on FRCs was equivalent after infection with wild-type and ΔvCD44BP MCMV (Extended Data Fig. 6a,b), and was not affected by vCD44BP expression (Extended Data Fig. 6c,d). Podoplanin distribution regulates FRC reticular network contractility[7], and CD44 has been implicated as a podoplanin partner protein[34,35], but the role of CD44 in FRC functionality remains unknown. We therefore examined whether the cellular distribution of CD44 is modified by MCMV infection, and specifically by vCD44BP. Confocal imaging showed that CD44 redistributed to the apical surface in FRCs infected with the ΔvCD44BP virus, but not in cells infected with wild-type MCMV (Extended Data Fig. 6e). The cell morphology index, a measure of changes in cell perimeter relative to cell area (a round cell has an index of 1), was significantly lower in cells infected with wild-type virus compared with those infected with ΔvCD44BP (Extended Data Fig. 6f). Thus, vCD44BP alters the distribution of CD44 in infected FRCs, preventing its redistribution after infection. By altering CD44 distribution in FRCs, vCD44BP might impair dendritic cell migration and the resulting dendritic cell–T cell interactions required for optimal T cell priming.

Co-expression and co-localization of vCD44BP and CD44 were confirmed in infected FRCs, including primary splenic FRCs (Fig. 4f,g and Extended Data Fig. 6g,h). Furthermore, expression of vCD44BP required CD44 expression, with significantly reduced vCD44BP expression observed in infected CD44-deficient splenic FRCs (Fig. 4h). Thus, *cis* interactions between CD44 and vCD44BP occur in infected FRCs.

FRC network remodelling is required for effective immune responses, and dendritic cell migration on this network is essential for efficient T cell priming. We hypothesized that by interfering with CD44 on FRCs, vCD44BP would affect morphological remodelling of the FRC network, thereby explaining the observed differences in dendritic cell migration. Analysis of the splenic FRC network after infection with wild-type or ΔvCD44BP MCMV revealed differential remodelling (Fig. 4i), with ΔvCD44BP infection resulting in longer individual podoplanin processes and more complex podoplanin[+] branches, reflected in increased branch length (Fig. 4j, left) and branch points (Fig. 4j, right). Changes in the network were not accompanied by differences in FRC numbers or FRC proliferation, which were equivalent in the two infections (Extended Data Fig. 6i,j). Expression of CCL19 and CCL21, FRC-produced chemokines that guide dendritic cell migration[9,36], was also equivalent after infection with wild-type or ΔvCD44BP MCMV

(Extended Data Fig. 6k). Thus, vCD44BP is likely to affect dendritic cell migration by directly interfering with CD44 on FRCs. Supporting this idea, time-lapse imaging of dendritic cell movement on FRCs in 3D cultures revealed a significant reduction in dendritic cell migration when FRCs were pre-treated with vCD44BP–Fc before co-culture with dendritic cells (Fig. 4k). Pre-incubation of dendritic cells with vCD44BP–Fc prior to co-culture with FRCs had no effect (Fig. 4k), excluding the possibility that vCD44BP acts directly on dendritic cells to influence their migration. Reduced dendritic cell movement also occurred when dendritic cells were cultured with CD44-deficient FRCs, a condition that mimics the unavailability of the hyaluronic acid-binding cleft on CD44 when vCD44BP is expressed in infected FRCs (Fig. 4l). Interaction between CLEC2 on dendritic cells and podoplanin on FRCs can affect dendritic cell migration[8,29]. Podoplanin partners with CD44 to execute its functions[34,35], with interaction mediated by the transmembrane and cytosolic regions[37]. Since podoplanin expression on the CD44-deficient FRCs used in our assay was unchanged (Extended Data Fig. 6l), and the CD44–vCD44BP–Fc interaction involves the extracellular domain of CD44 (where the hyaluronic acid-binding cleft is located), we conclude that CD44 can direct dendritic cell migration independently of podoplanin–CLEC2 interactions. Consistent with dendritic cell migration being directed by CD44, the dendritic cells in the co-cultures expressed the CD44 ligand hyaluronic acid (Extended Data Fig. 6m).

## Stromal CD44 tunes CD8 T cell responses

Our data demonstrated that MCMV vCD44BP targets CD44 and alters its distribution in infected FRCs, the cells that direct dendritic cell trafficking in the spleen. They also showed that targeting of CD44 by vCD44BP impaired antiviral CD8 T cell responses. These findings suggest a critical role for stromal CD44 in the generation of antiviral T cell immunity. To substantiate this, we used bone marrow chimeras in which the stromal compartment lacked CD44, while CD44 was expressed by haematopoietic cells. In addition, we generated chimeric mice using a CRISPR–Cas9 based approach to target CD44 in haematopoietic stem cells. We infected mice with wild-type or ΔvCD44BP MCMV and examined CD8 T cell responses and viral control (Fig. 5a). Control of acute MCMV infection relies on CD8 T cells when Ly49H-mediated NK cell responses are absent. Given that our chimeras required the use of mice on the C57BL/6 background, which express Ly49H, we used TC1 mice, a congenic C57BL/6 strain that lacks Ly49H[38] as donors. In TC1→wild-type C57BL/6 (WT) chimeras, in which CD44 was present on stromal cells, viral loads were lower in mice infected with the ΔvCD44BP virus compared with wild-type MCMV, but this difference was lost in TC1→*Cd44*[−/−] chimeras, which lacked stromal CD44 (Fig. 5b).

Next, we assessed antiviral CD8 T cell responses in TC1→*Cd44*[−/−] chimeras, which lack CD44 specifically on stromal cells. Total CD8 T cell (Extended Data Fig. 7a) and M45[+] virus-specific CD8 T cell numbers (Fig. 5c and Extended Data Fig. 7b) were equivalent after infection with wild-type or ΔvCD44BP MCMV. These responses differed from those observed in TC1→WT chimeras (Fig. 5c and Extended Data Fig. 7a,b) and the parental TC1 strain used to generate the chimeras (Extended Data Fig. 7c–e). Thus, CD44 must be expressed by stromal cells for vCD44BP to impair the antiviral CD8 T cell response.

Consistent with stromal CD44–vCD44BP interactions hindering CD8 T cell responses by disrupting dendritic cell migration, the enhanced migration of dendritic cells into the splenic white pulp in TC1→WT chimeras infected with the ΔvCD44BP virus (relative to wild-type MCMV) was absent in TC1→*Cd44*[−/−] chimeras (Fig. 5d and Extended Data Fig. 7f). Indeed, in TC1→*Cd44*[−/−] chimeras, dendritic cell migration after ΔvCD44BP infection was comparable to that observed in wild-type MCMV infection and significantly lower ($P < 0.0001$) than in ΔvCD44BP-infected TC1→WT chimeras (Fig. 5d and Extended Data Fig. 7f). Thus, in the absence of CD44 expression on stromal cells, ΔvCD44BP infection did not enhance dendritic cell migration into

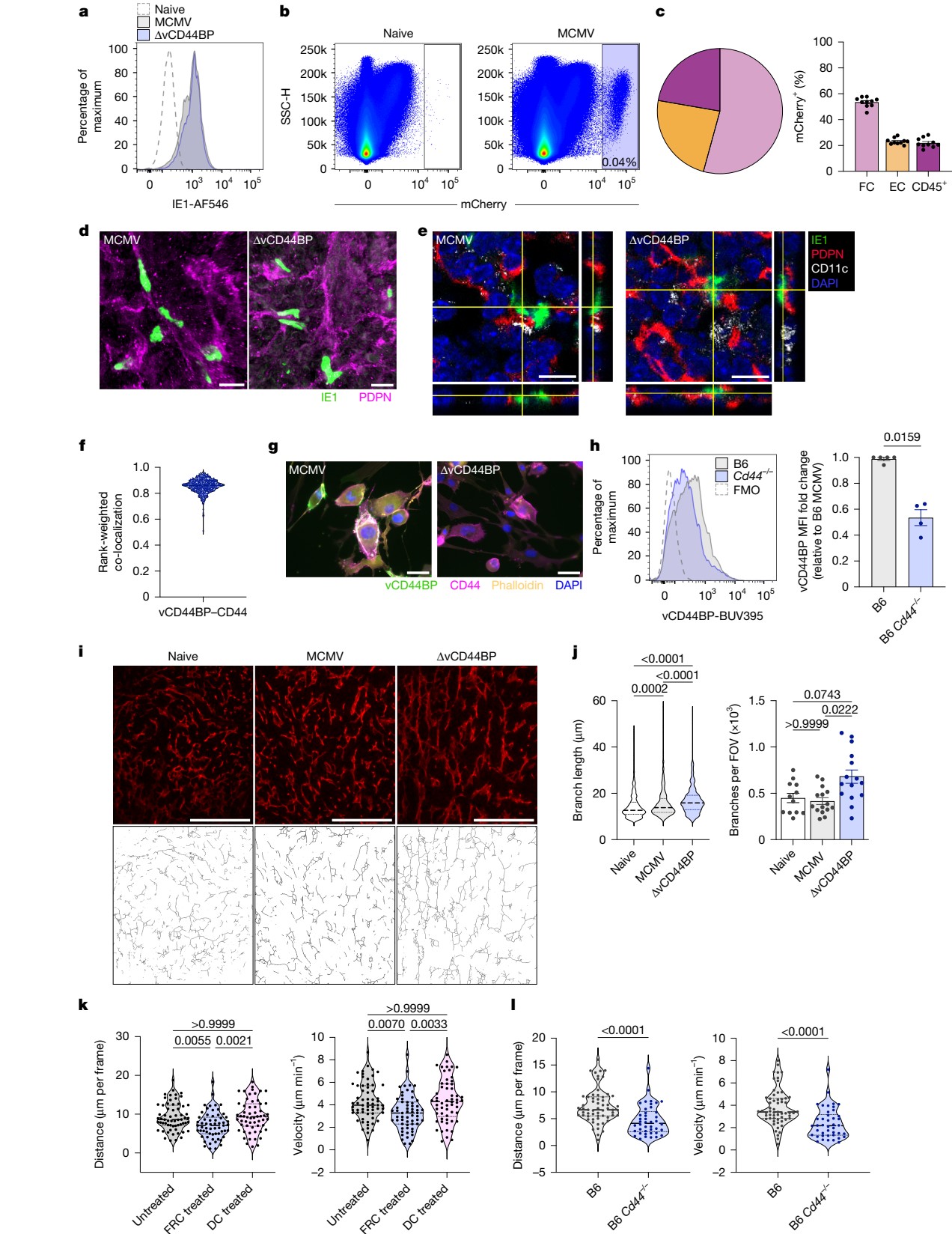

**Fig. 4** | See next page for caption.

the T cell zone. Supporting the role of stromal CD44 in dendritic cell localization, naive mice lacking stromal CD44 also showed reduced dendritic cell accumulation in the splenic white pulp (Fig. 5d and Extended Data Fig. 7f).

To examine whether vCD44BP also functions via CD44 on haematopoietic cells, including dendritic cells, we generated chimeric mice lacking CD44 expression in haematopoietic stem cells derived from TC1 mice using a CRISPR–Cas9 approach (Extended Data Fig. 8a–c).

**Fig. 4 | vCD44BP interacts with CD44 in FRCs, remodels the FRC network and alters dendritic cell migration. a**, IE1 expression in FRC2 cells 24 h after infection with wild-type or ΔvCD44BP MCMV (MOI = 3). Data are representative of two independent experiments. **b**, Representative plots of splenocytes from uninfected or MCMV-mCherry-infected BALB/c mice (24 h after infection, $1 \times 10^4$ PFU) showing mCherry$^+$ infected cells. **c**, Pie chart and graph showing frequencies of infected fibroblastic stromal cells (FC), endothelial cells (EC) and leukocytes (CD45$^+$). Data are pooled from three independent experiments (*n* = 10 mice per group). **d**, Confocal images (processed with Imaris for display purposes) of infected IE1$^+$PDPN (podoplanin)$^+$ stromal cells in splenic white pulp (2 dpi). Data are representative of three independent experiments. Scale bars, 10 μm. **e**, Confocal 3D-orthogonal views of infected FRCs (IE1$^+$PDPN$^+$) in spleens of MCMV- or ΔvCD44BP-infected mice (2 dpi). Data are representative of two independent experiments. Scale bars, 10 μm. **f**, vCD44BP–CD44 co-localization in MCMV-infected FRC2 (MOI = 3, 24 h). Image analysed by ImageStreamX; 484 infected FRCs pooled from 2 independent experiments. **g**, Images of MCMV- or ΔvCD44BP-infected primary splenic FRCs (MOI = 3, 24 h) stained for vCD44BP, CD44, actin (phalloidin). Scale bars, 50 μm. **h**, vCD44BP expression on primary splenic FRCs from B6 and *Cd44$^{-/-}$* mice infected with MCMV (MOI = 3, 24 h) assessed by flow cytometry, representative histograms and fold-change relative to B6 FRCs; pooled from four independent experiments (B6, *n* = 5; *Cd44$^{-/-}$*, *n* = 4). FMO, fluorescence minus one; MFI, mean fluorescence intensity. **i**, PDPN-stained spleen sections from uninfected, or wild-type or ΔvCD44BP MCMV-infected mice (2 dpi) and corresponding skeletonized PDPN$^+$ networks. Scale bars, 100 μm. **j**, Plots showing skeleton branch length and branch number per field of view (FOV); top 30 branch lengths (naive, 360; infected, 450) and branch numbers (naive, 12; infected, 15) from each FOV. Data are from 2 naive and 3 infected mice per group. **k**, Distance and velocity of dendritic cells (DCs) in contact with FRCs quantified by time-lapse live imaging. Data are pooled from three independent experiments (untreated, *n* = 62; vCD44BP–Fc-treated FRCs, *n* = 60; vCD44BP–Fc-treated dendritic cells, *n* = 58). **l**, Distance and velocity of dendritic cells in contact with B6 or B6 *Cd44$^{-/-}$* FRCs; pooled from two independent experiments (B6, *n* = 59; *Cd44$^{-/-}$*, *n* = 42). All mice were infected with $5 \times 10^3$ PFU unless indicated. Graphs show mean ± s.e.m.; violin plots show median and quartiles; significance tested by two-sided Mann–Whitney or Kruskal–Wallis (in **j**,**k**).

We confirmed specific loss of CD44 in leukocytes, including dendritic cells, in chimeric mice generated using guide RNAs targeting *Cd44* (g*Cd44*), but not in mice generated with a non-targeting scrambled RNA (gScr) (Extended Data Fig. 8c). We then infected the chimeric mice with wild-type or ΔvCD44BP MCMV and examined CD8 T cell responses (Extended Data Fig. 8d–f) and viral control. The reduced viral loads (Fig. 5e) and improved antiviral CD8 T cell responses (Fig. 5f and Extended Data Fig. 8e) observed after ΔvCD44BP infection were lost in mice lacking CD44 on stromal cells (Fig. 5e,f, gScr→*Cd44$^{-/-}$*). By contrast, when CD44 was absent on haematopoietic cells (g*Cd44*→WT), ΔvCD44BP-infected mice had significantly lower viral loads (Fig. 5e) and better antiviral CD8 T cell responses (Fig. 5f) compared with those infected with wild-type MCMV, a response equivalent to that observed in chimeric mice with normal CD44 expression on haematopoietic cells (gScr→WT) (Fig. 5e,f). These findings confirmed that vCD44BP delays antiviral CD8 T cell responses by targeting stromal CD44.

Consistent with fibroblastic CD44 being critical to the initiation of CD8 T cell responses, fibroblastic stromal cells expressed higher levels of CD44 than endothelial cells or haematopoietic cells involved in priming, such as dendritic cells (Extended Data Fig. 8g). Only a small fraction of endothelial cells expressed CD44, and the majority (around 90%) of MCMV-infected endothelial cells lacked CD44 expression (Extended Data Fig. 8h).

Generation of adaptive immune responses requires expansion of the FRC network; for example, immunization-induced lymph node hyperplasia is critical for effective adaptive immunity[7]. We therefore tested whether vCD44BP could alter adjuvant-driven lymph node expansion independently of viral infection.

Mice were injected with adjuvant[39] and subsequently received vCD44BP–Fc or an irrelevant Fc protein subcutaneously. Cervical (draining) and inguinal (non-draining) lymph nodes were then collected (Fig. 5g). The cervical lymph nodes draining the site of injection expanded in size and cellularity, and this response was constrained by vCD44BP–Fc treatment (Fig. 5h,i). As expected, no hyperplasia occurred in the non-draining inguinal lymph nodes, and vCD44BP–Fc had no effect (Extended Data Fig. 9a). Similar results were obtained when mice were injected with antigen plus adjuvant (Extended Data Fig. 9b,c). In these experiments, T cell responses, including antigen-specific responses in the draining lymph nodes, were significantly reduced by vCD44BP–Fc treatment (Extended Data Fig. 9d–g). These findings provide evidence that vCD44BP can impair adjuvant-driven lymph node expansion independently of infection.

Finally, to determine whether CD44 expression in FRC is broadly relevant, we examined the effect of treatment with vCD44BP–Fc in a different model of viral infection (Fig. 5j), and demonstrated a significant impairment of the influenza-specific CD8 T cell response (Fig. 5k–m and Extended Data Fig. 9h,i). Together our data establish that altering CD44 function broadly affects immune responses, including the generation of antigen-specific CD8 T cell responses.

## Discussion

We identified an MCMV-encoded protein, vCD44BP, that revealed a previously unknown mode of viral immune evasion and an unrecognized biological function of CD44. Through co-evolution with their hosts, herpesviruses have acquired genes that subvert cellular responses to enhance virus survival[10]. vCD44BP promotes virulence by interacting with CD44 and modulating its function, thereby impairing antiviral CD8 T cell responses. Accordingly, disrupting the m11 gene, which encodes vCD44BP, reduced virulence during acute infection, a phenotype that was reversed by CD8 T cell depletion.

Mechanistically, vCD44BP interferes with CD44 binding to its ligand hyaluronic acid. Crystal structure analysis of the vCD44BP–CD44 complex showed that inhibition occurs via steric occlusion of the hyaluronic acid-binding domain of CD44[17], demonstrating that vCD44BP and hyaluronic acid binding to CD44 are mutually exclusive. Furthermore, our data indicate that the CD44–vCD44BP interaction occurs in *cis* within infected FRCs. By physically occluding the hyaluronic acid-binding site on CD44, vCD44BP functions as a natural inhibitor, effectively limiting stromal CD44 accessibility to hyaluronic acid on other cells or possibly within the extracellular matrix.

Binding of vCD44BP to CD44 is indicative of an induced-fit mode of interaction, which, together with the distinct nature of the interactions, suggests that vCD44BP does not function by molecular mimicry. These findings provide insights into novel strategies to modulate CD44 function, which is particularly relevant since CD44 has been recognized as a target in cancer and autoimmunity, and there is an active search for selective CD44-blocking reagents[16,40–44].

CD44 mediates diverse processes including haematopoiesis, cell adhesion, lymphocyte migration and tumour metastasis[16,45]. Our findings reveal a mechanism by which CD44 on stromal cells participates in the initiation of adaptive immunity. Effective T cell responses rely on interactions between dendritic cells, the professional antigen presenting cells, and T cells within the T cell compartment of secondary lymphoid organs[46,47]. During MCMV infection, CD44 is targeted by vCD44BP, resulting in impaired trafficking of dendritic cells. By limiting the movement of dendritic cells in secondary lymphoid organs, vCD44BP delays the generation of CD8 T cell responses. Our findings demonstrate that this is achieved by targeting CD44 specifically on stromal cells and, by excluding a role for CD44 on endothelial cells, uncover a previously unrecognized immunomodulatory function of CD44 on fibroblastic stromal cells. CD44 is expressed by multiple fibroblastic

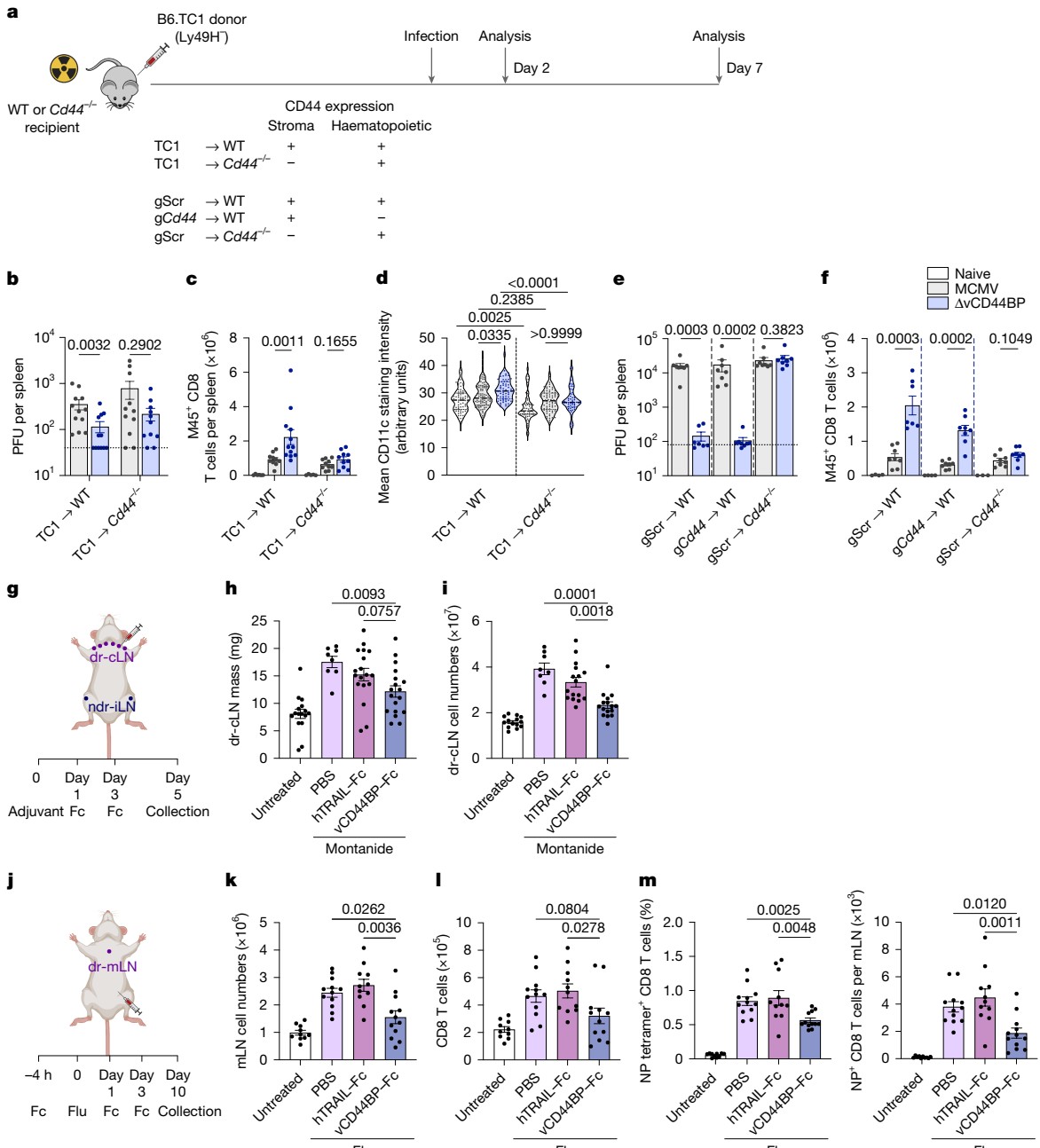

**Fig. 5 | vCD44BP–CD44 interactions in stromal cells regulate the generation of antigen-specific CD8 T cell responses. a**, Experimental setup. **b**, Viral loads in spleens of chimeric mice infected with MCMV or ΔvCD44BP virus (TC1→WT, *n* = 12; TC1→*Cd44*⁻/⁻, *n* = 11). **c**, Splenic M45⁺ virus-specific CD8 T cell numbers in chimeric mice infected with MCMV or ΔvCD44BP (7 dpi) or uninfected controls (TC1→WT: naive, *n* = 6; MCMV, *n* = 10; ΔvCD44BP, *n* = 12; TC1→*Cd44*⁻/⁻: naive, *n* = 5; MCMV and ΔvCD44BP, *n* = 10). **d**, Mean CD11c staining intensities in the white pulp of chimeric mice infected with MCMV or ΔvCD44BP (2 dpi) or uninfected controls. Number of white pulp areas examined: TC1→WT (naive, *n* = 79; MCMV, *n* = 97; ΔvCD44BP, *n* = 65); TC1→*Cd44*⁻/⁻ (naive, *n* = 58; MCMV, *n* = 80; ΔvCD44BP, *n* = 56) from 3 mice per group within one experiment. **e,f**, Viral loads (**e**) and M45⁺ CD8 T cell numbers (**f**) in spleens of chimeric mice infected with MCMV or ΔvCD44BP (7 dpi). Data are combined from two independent experiments (gScr→WT: MCMV, *n* = 8; ΔvCD44BP, *n* = 7; naive, *n* = 4; g*Cd44*→WT: MCMV, *n* = 8; ΔvCD44BP, *n* = 8; naive, *n* = 4; gScr→*Cd44*⁻/⁻: MCMV, *n* = 8; ΔvCD44BP, *n* = 8; naive, *n* = 3). All mice were infected with 5 × 10³ PFU. **g**, Experimental setup for evaluation of adjuvant-driven responses following vCD44BP–Fc-mediated CD44

inhibition. dr-cLN, draining cervical lymph nodes; ndr-iLN, non-draining inguinal lymph nodes. **h,i**, Mass of (**h**; untreated, *n* = 16; PBS, *n* = 8; hTRAIL–Fc and vCD44BP–Fc, *n* = 18) and total cell numbers in (**i**; untreated, *n* = 14; PBS, *n* = 8; hTRAIL–Fc and vCD44BP–Fc, *n* = 16) draining cervical lymph nodes of untreated mice and mice treated with Montanide that received PBS, hTRAIL–Fc or vCD44BP–Fc. Data are pooled from two or three independent experiments. **j**, Experimental setup for evaluation of immune responses to influenza (flu) infection following vCD44BP–Fc-mediated CD44 inhibition. **k–m**, Total cell numbers (**k**) and CD8 T cell numbers (**l**) and frequency and number of nucleoprotein (NP)⁺ flu-specific CD8 T cells (**m**) in draining mediastinal lymph nodes (mLNs). Data are pooled from three independent experiments (untreated, *n* = 10; PBS, *n* = 12; hTRAIL–Fc, *n* = 11 and vCD44BP–Fc, *n* = 12). Graphs show mean ± s.e.m.; violin plots show median and quartiles. Significance tested by two-sided Mann–Whitney (in **b,c,e,f**) or Kruskal–Wallis tests (in **d,h,i,k–m**). Schematics in **a,g,j** created in BioRender. Sng, X. (2025) https://BioRender.com/hgpsca9.

stromal cells, including FRCs, and has been postulated to regulate the functioning of the FRC network. Thus, it is notable that vCD44BP alters the distribution of CD44 in FRCs and affects the FRC network remodelling that is required for effective immune responses[7,8,39,48]. CD44 is an important binding partner for podoplanin[34,35], a protein that mediates the dynamic response of FRC networks to dendritic cells migrating in T cell zones[29]. When dendritic cells enter lymph nodes, podoplanin-expressing FRCs engage CLEC2-bearing dendritic cells, driving expansion of the FRC network to accommodate further leukocyte migration and proliferation[7]. In vitro studies suggest that CD44 suppresses podoplanin-driven FRC contractility[7], allowing FRCs to form protrusions and cell–cell junctions to support lymphocyte expansion[35]. Here we show that by interfering with CD44 in FRCs, vCD44BP reduced dendritic cell movement in vitro and dendritic cell trafficking to the splenic T cell zone during infection in vivo, highlighting the contribution of stromal CD44 to dendritic cell migration. These findings were corroborated by in vitro evidence that FRC-expressed CD44 influenced dendritic cell movement on FRCs. Although podoplanin is integral to interactions with dendritic cells via CLEC2 and can affect dendritic cell migration, we found that CD44 can direct dendritic cell migration independently of podoplanin–CLEC2 interactions. The role of FRC-expressed CD44 in guiding dendritic cell migration does not preclude its involvement in activities mediated by podoplanin. Considering the pivotal role of FRCs in initiating adaptive immune responses, CD44 and podoplanin are likely to function both cooperatively and independently, building redundancy in this essential system. Additionally, it is possible that FRC-expressed CD44 influences initial dendritic cell migration, and this is subsequently sustained by podoplanin–CLEC2 interactions that promote maximal network expansion and further lymphocyte influx. Thus, our findings reveal a previously unrecognized function of stromal CD44 in the initiation of adaptive immunity that is essential but operates in concert with other mechanisms (such as chemokines and the podoplanin–CLEC2 axis) to maintain robustness in a system that is critical for host survival. By targeting stromal CD44 through vCD44BP, CMV delays but does not abolish T cell responses, an evolutionarily advantageous strategy that enhances viral replication while preserving host survival.

Our data show that FRCs are a primary target of vCD44BP; however, this does not exclude the possibility that targeting of other fibroblastic stromal subsets, such as marginal zone fibroblasts, may also contribute to the impaired dendritic cell trafficking observed in vivo. Although dendritic cells express hyaluronic acid[49], and this has a central role in their migration, hyaluronic acid is also a component of the extracellular matrix, and thus FRC–extracellular matrix interactions may additionally influence dendritic cell trafficking. Furthermore, hyaluronic acid is present in the glycocalyx of FRCs, and may therefore be affected by vCD44BP, thus its role in FRC function and dendritic cell migration merits further investigation. Collectively, our study provides a molecular mechanism for dendritic cell trafficking within secondary lymphoid organs mediated by CD44–hyaluronic acid interactions and uncovers an immunomodulatory function of fibroblastic CD44. The selective targeting of fibroblastic stromal cells, rather than widespread effects on multiple cell populations, reflects a highly focused immune evasion strategy that highlights the critical role of fibroblastic CD44 in orchestrating immune responses and demonstrates that CMV has evolved a mechanism to specifically target this critical checkpoint.

To our knowledge, this is the first description of a viral immune evasion protein specifically targeting CD44. Our studies reveal that CD44 influences the generation of antiviral T cell responses beyond CMV, as shown by the ability of a soluble form of vCD44BP to impair an influenza-specific CD8 T cell response. Furthermore, vCD44BP–Fc interfered with lymph node hyperplasia and the generation of antigen-specific CD8 T cell responses in vaccination, highlighting the critical role of the stromal CD44 pathway for robust immunity outside of viral infection. Lymphadenopathy characterizes several pathological conditions, including autoimmune disease and malignancy[3,50,51]. Given that CD44 antibodies can block inflammation[52], vCD44BP may have potential as an anti-inflammatory agent.

In sum, our study reveals a viral immune evasion mechanism targeting a previously unrecognized CD44 function that relates to its expression on fibroblastic stromal cells.

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

# Methods

## Generation of m11 chimeric Fc protein and m11-specific antibodies

The MCMV m11–Fc (also referred to as vCD44BP–Fc) fusion protein was generated using Sew-PCR to attach the Fc portion of human IgG1 to the part of the m11 gene that encodes its extracellular domain (amino acids 1–212). The construct was transiently transfected into CV-1/EBNA cells and the soluble fusion protein purified using protein-A-Sepharose (Amersham Pharmacia Biotech). A human TRAIL–Fc fusion protein was used as a negative control. A monoclonal antibody (M-627) to m11 was generated by immunizing mice with the m11–Fc protein and fusing splenocytes with a myeloma partner using standard methodologies. The specificity of M-627 was verified using m11 transfected cells and by comparing cells infected with wild-type MCMV and a mutant lacking m11, referred to as ΔvCD44BP—see below. The anti-m11 monoclonal antibody (7G5) was generated by immunizing rats with the m11 ectodomain (amino acids 28–164) and fusing splenocytes with X63 myeloma cells using standard methodologies. Supernatants from hybridoma clones were initially screened by ELISA using plate bound m11–Fc protein and a secondary screen performed by flow cytometry using fibroblasts infected with either MCMV or ΔvCD44BP virus.

## Expression cloning of the m11 cognate from a CD40 ligand-stimulated B cell cDNA library

A cDNA library was generated from mouse B cells stimulated with CD40 ligand and cloned into the pDC409 vector using previously described methods[53]. Approximately 200 pools, each containing ~2,000 clones, were transfected into CV-1/EBNA cells and 2 days later the transfected cells screened using the m11–Fc protein as described[54]. One positive pool was identified, and this was subdivided into smaller pools until a single positive cDNA was obtained from the original pool. Individual clones were then sequenced, and sequences were compared to public DNA databases.

## Construction and cloning of m11 for transfection

The MCMV m11 open reading frame (ORF) was amplified by PCR based on published sequences (accession number AM886412) using purified MCMV DNA templates. The m11 PCR product was cloned into the pDC409 mammalian expression vector[54] to generate p409-m11.

## Cell lines

COS-7 and EL4 cells were cultured in Dulbecco's modified Eagle medium (DMEM, Gibco); IC-21 cells were cultured in Roswell Park Memorial Institute (RPMI-1640, Gibco); M2-10B4 and mouse embryonic fibroblasts (MEFs) were cultured in minimal essential medium (MEM, Gibco). All culture media were supplemented with 10% fetal calf serum (FCS) (Gibco) (COS-7, EL4 and IC-21) or 10% newborn calf serum (NCS) (M2-10B4 and MEF), and antibiotics (penicillin 100 μg ml$^{-1}$, CSL; gentamycin 40 μg ml$^{-1}$, Pharmacia & Upjohn).

The fibroblastic reticular cell line (FRC2) was generated by A.L.F. by isolating FRCs from the lymph nodes of C57BL/6J mice as described[55] and FRC enriched via CD45 and CD31 depletion (Miltenyi Biotec). Stromal cells were plated overnight in alpha-MEM with 10% FBS (Gibco) and then co-transfected using Lipofectamine 3000 with a piggyBac expression vector expressing SV40LT with GFP and a piggyBac transposase expression vector, hyPBase (The Sanger Centre, pCM-hyPBase).

## CD44–hyaluronic acid binding assay

The hyaluronic acid adhesion assay was performed with modifications of a previously described method[56]. In brief, 96-well plates (Costar) were coated with 100 μl of a 100 μg ml$^{-1}$ solution of hyaluronic acid (220 kDa) and incubated overnight at 37 °C. The plates were then gently washed and incubated with 200 μl of 2% BSA in PBS for 2 h at 37 °C. Separately, $2 \times 10^5$ EL4 cells previously treated for 24 h with PMA (100 ng ml$^{-1}$) and ionomycin (500 ng ml$^{-1}$) (Thermo Fisher Scientific) were added to the wells of 96-well plates containing 10 μg ml$^{-1}$ of anti-CD44 antibody (KM114, that masks the hyaluronic acid binding site), m11–Fc or the irrelevant human TRAIL–Fc, and incubated for 30 min at 4 °C. An anti-human-Fc antibody (10 μg ml$^{-1}$, Jackson ImmunoResearch) was then added to the wells containing the Fc proteins and the plates incubated for a further 30 min at 4 °C. These cells were then transferred to the hyaluronic acid-coated plates in DMEM containing 10% FCS, and incubated for 1 h at 37 °C. After three washes with PBS, 100 μl of DMEM were added back into the wells. To quantify the number of cells in the wells, an MTT colorimetric assay (Sigma-Aldrich, St Louis, MO, USA) was used and the absorbance read at 570 nm on a AD200 microplate reader (Beckman-Coulter Inc.). MTT incorporation levels reflect number of cells present in the wells as determined microscopically with the number of EL4 cells calculated using a standard curve.

## Protein chemistry

The construct encoding mouse CD44 (amino acids 23–174) was cloned into the p30 vector and expressed as inclusion bodies in TonA-BL-21 *Escherichia coli* cells. CD44 was refolded by dilution in a solution containing 4 M urea, 0.4 M L-arginine, 0.1 M EDTA, 0.1 M Tris-HCl pH 8.0 in a 5:1 mM reduced:oxidized glutathione overnight at 4 °C. Refolded CD44 was purified first via DEAE anion exchange and size-exclusion chromatography using a Superdex S75 16/600 column (GE Healthcare). The m11 ectodomain (amino acids 28–164 from MCMV strain K181) was cloned into the pFASTBac vector (Invitrogen) to include a C-terminal hexa-histidine tag. The plasmid was incorporated into a recombinant baculovirus, and the viral titre expanded in SF9 cells as described in the Bac-to-Bac manual (Invitrogen). Soluble m11 was obtain by infecting Hi5 cells with 2% P3 virus. The construct encoding m04 (amino acids 24–223 from MCMV strain G4) was cloned into the pHLSec vector to include a C-terminal hexa-histidine tag and expressed via transient transfection in human embryonic kidney 293-S cells as described[13]. Secreted proteins from mammalian and baculoviral systems were buffer-exchanged into 10 mM Tris-HCl (pH 8.0), 500 mM NaCl and purified via nickel-affinity and size-exclusion chromatography using a Superdex S200 16/600 column (GE Healthcare) in a 10 mM Tris-HCl (pH 8.0), 150 mM NaCl buffer.

## Crystallization and data collection

An equimolar mixture of m11 and CD44 was resolved using a Superdex S200 16/600 column (GE Healthcare) and pure heterodimer was concentrated to 9.65 mg ml$^{-1}$ in 20 mM Tris-HCl pH 8.0 and 150 mM NaCl. Crystals were obtained using the hanging drop vapour diffusion method from a mother liquor containing 0.2 M LiSO$_4$, 0.1 M Tris-HCl pH 8.5 and 30% (w/v) PEK 3000. Prior to data collection, crystals were cryoprotected in mother liquor supplemented with 10% (v/v) glycerol. Crystals were flash cooled using liquid nitrogen and X-ray diffraction data was recorded using a Quantum-315 CCD detector at the MX2 beamline of the Australian Synchrotron. Data were integrated by MOSFLM and scaled using SCALA within the CCP4 suite of programmes (Extended Data Table 1).

## Structure determination and refinement

The structure was determined by molecular replacement using Phaser. Isolated models for CD44 and m11 were generated from the structure of mouse CD44 (PDB ID: 2JCP) and m04 (PDB ID: 4PN6), respectively, using PyMOL (Schrödinger, Inc.). The structure was refined via iterative cycles of model building in Coot and refinement using Buster (http://globalphasing.com/buster/). N- and O-linked glycans were manually incorporated into regions of positive density that correlated to the requisite sequence motif: NX(S/T), where X is any amino acid except proline, serine or threonine. The final structure was refined to final Rfactor/Rfree values of 18.4% and 20.2%, respectively. Details of the refinement statistics are provided in Extended Data Table 1. The structure

factor file and associated atomic coordinates have been deposited in the PDB under accession code 9EJW.

## Surface plasmon resonance

SPR experiments were performed using a BIAcore 3000 system (GE Healthcare) at 25 °C with a buffer comprising 10 mM Tris-HCl pH 8.0, 300 mM NaCl, and 0.005% (v/v) surfactant P20. CM5 sensor chips (GE Healthcare) were primed with an equal mixture of EDC (1-ethyl-3-(3-dimethylaminopropyl)carbodiimide) and NHS (N-hydroxysuccinimide). The m11 and m04 ectodomains were prepared in buffer containing 150 mM NaCl and either 20 mM sodium acetate pH 5.3 or 100 mM sodium citrate pH 3.0, respectively, and approximately 2,300 response units were immobilized per flow cell. Flow cells were quenched with 20 μl of ethanolamine at a flow rate of 5 μl min$^{-1}$ and primed twice with running buffer prior to injection of analyte. Varying concentrations of CD44 or hyaluronic acid (100–0.1 μM) pre-incubated with a fixed concentration (12.5 μM) of CD44 were passed over the flow cells for 65 s, in duplicate, at a flow rate of 10 μl min$^{-1}$. The final responses were double referenced by subtracting responses from an 'empty' flow cell. The responses at equilibrium were used to construct equilibrium binding curves that fit by a single-site binding model. The calculated equilibrium dissociation constants represent the mean ± s.e.m. from two independent experiments. Data were analysed with Scrubber (BioLogic Software) and Prism (GraphPad Software).

## Generation of the ΔvCD44BP virus and revertant virus

A homologous recombination approach was used to generate the m11 mutant virus. A 5 kb Hpa1 fragment spanning residues 7114–12176 (Genbank accession AM886412) of the K181 strain of MCMV was cloned into pBluescript SK⁻. The 4.2 kb LacZ cassette from the MV10 vector was subcloned into the Nde1 site at nucleotide positions (10,948–10,953) within the m11 ORF. Following linearization, the plasmid was co-transfected into MEFs together with purified K181 MCMV DNA. Plaques were screened for β-galactosidase expression using X-gal staining and β-gal⁺ plaques were plaque purified to generate a Δm11 stock. To introduce a premature stop codon into the m11 ORF, Sew-PCR was used to generate a PCR product in which the unique Nde1 site in m11 was replaced with an Hpa1 restriction site encompassing an in-frame premature stop codon. This construct was used to co-transfect MEFs with purified Δm11 viral DNA. Viral preparations in which the LacZ cassette has been substituted by the Hpa1-containing construct were selected by identification of plaques that did not stain blue with X-gal staining. Plaques were purified to homogeneity and a single clone selected and designated ΔvCD44BP. An m11 revertant virus (REV) was generated by co-transfection of Δm11 with plasmid constructs containing wild-type K181 sequence spanning the m11 region. All mutants were sequence verified across the m11 region and restriction fragment length polymorphism analysis was performed to compare the profiles of the stop mutants, revertant and wild-type viruses.

## Mice

BALB/c and C57BL/6J mice were purchased from the Animal Resources Centre/Ozgene ARC (Perth, Western Australia, Australia) or the Walter and Eliza Hall Institute of Medical Research (Melbourne, Victoria, Australia). B6 Cd44$^{-/-}$ (ref. 57) and B6 BALB-TC1 (TC1) (H2$^b$ NK1.1 + Ly49H⁻)[38] mice were bred at Perkins Bioresources Facility (Perth, Western Australia, Australia). BALB/c.Ifng$^{-/-}$ mice and BALB/c.Prf1$^{-/-}$ mice were obtained from the Animal Services Facility at QIMR Berghofer Medical Research Institute (Queensland, Australia). Age-matched adult female mice (8–12 weeks old) were used as controls for all experiments. All animal experimentation was performed with ethics approval from Monash University Ethics Committee (MARP2); Perkins Animal Ethics Committees (for the Lions Eye Institute); University of Western Australia Animal Ethics Committee (for the Lions Eye Institute) and in accordance with NHMRC Australia Code of Practice for the Care and Use of Animals for Scientific Purposes.

## Viral infections and in vivo monoclonal antibody administration

Mice were infected intraperitoneally with MCMV (K181 strain), ΔvCD44BP, MCMV-REV or MCMV-K181-Perth-mCherry salivary gland-propagated virus ($5 \times 10^3$ or $1 \times 10^4$ PFU, except for BALB/c. Prf1$^{-/-}$ which were infected with $2 \times 10^3$ PFU owing their increased susceptibility to infection) diluted in PBS containing 0.05% FCS. For CD8 or CD4 T cell depletion studies, mice were injected intraperitoneally with anti-CD8β monoclonal antibody (clone 53.5.8, 250 μg per injection) or anti-CD4 monoclonal antibody (clone GK1.5, 500 μg per injection) at days −2, 0 and 2 relative to virus injection. Depletion of CD4 or CD8 T cells was confirmed by flow cytometric analysis.

For influenza infections, mice were infected intraperitoneally with $1.5 \times 10^7$ PFU of the influenza A virus (IAV) strain A/Puerto Rico/8/34 (H1N1, PR8) diluted in PBS.

For intravenous antibody labelling of dendritic cells, 1.5 μg of anti-CD11c APC antibody (clone HL3) was injected 36 h post-infection and mice were humanely killed three minutes later. Spleens were collected and single-cell suspensions were prepared using the stromal isolation protocol described in ref. 55, stained with the appropriate antibodies, and analysed by flow cytometry.

## Generation of bone marrow chimeras

Bone marrow cells were collected from the tibia, femur and ilium of TC1 donor mice and washed with sterile PBS. Recipient WT or B6 Cd44$^{-/-}$ mice received two doses of 500 cGy total-body irradiation, spaced 3 h apart, prior to receiving $10^7$ bone marrow cells administered by intravenous injection. Chimeric mice were housed for three months to allow full reconstitution of the haematopoietic compartment. Chimerism in the haematopoietic compartment was >95% in this system.

## CRISPR-mediated deletion of CD44

Bone marrow cells were isolated from TC1 mice and haematopoietic stem cells were enriched using an EasySep Mouse Hematopoietic Progenitor Cell Isolation Kit following the manufacturer's instructions (STEMCELL Technologies, 19856 A). The purified progenitor cells were seeded into 24 wells plates at $1 \times 10^6$ cells per well and incubated at 37 °C for 2 h in 1.5 ml of growth medium consisting of StemSpan SFEM II medium (STEMCELL Technologies, 09605) supplemented with 50 ng ml$^{-1}$ Stem cell factor (SCF) (Thermo Fisher Scientific, PMC2113L). CD44 deletion was achieved by precomplexing two Cd44 single guide RNA (sgRNA) guides (300 pmoles of Cd44 sgRNA1 plus 300 pmoles of Cd44 sgRNA2) with 36.3 pmoles of Cas9 protein (IDT, 1081059). Each reaction was in a total of 5 μl and was incubated at room temperature for 10 min. In addition, control electroporation reactions using a scrambled non-targeting sgRNA were performed. P3 Nucleofector Solution (Lonza, PBP3-00675) was prepared by mixing 16.4 μl of P3 Solution with 3.6 μl of Supplement 1 and the solution was allowed to equilibrate to room temperature. Progenitor cells were washed once in PBS before being resuspended in P3 Nucleofector Solution ($5 \times 10^5$ cells in 20 μl) and the cells added to the guide–Cas9 mixture. A total of 20 μl of the cell plus guide–Cas9 mixture was transferred to a well of a 16-well Nucleocuvette strip and pulsed using the unstimulated mouse T cell programme (4D-Nucleofector X Unit, Lonza). Multiple electroporations were performed to generate a sufficient number of cells. Immediately after electroporation, 80 μl of warm growth medium was added to each well and the strip incubated at 37 °C for 30 min. Cells were then transferred to a 6-well plate and cultured in growth medium at 37 °C for a further 2 days. Cells from individual cultures were pooled and loss of CD44 confirmed by flow cytometry before being injected into lethally irradiated C57BL/6J (CD45.1) recipient mice ($2 \times 10^5$ cells per mouse). Details of the sgRNA guides are provided in Extended Data Table 2.

## Quantification of viral loads

Viral titres were quantified as described[58]. In brief, individual organs were homogenized in MEM containing 2% NCS and the homogenate was centrifuged at 3,000$g$ for 15 min at 4 °C. The supernatant was collected, and viral titres determined by adding serial dilutions of the supernatant to a sub-confluent monolayer of M2-10B4 cells for 1 h at 37 °C. The supernatant was then removed, and cells grown in MEM + 2% NCS containing carboxy-methylcellulose for 4 days. Cells were fixed, stained with 0.5% methylene blue in 10% formaldehyde for 24 h and plaques in the monolayer counted.

## Isolation of leukocytes and fibroblastic reticular cells

Mice were humanely killed and spleens and/or lymph nodes removed. FRCs from the spleen or lymph nodes were isolated by enzymatic digestion as described previously[55]. Spleen or lymph node leukocytes were isolated by mechanical disruption of tissues, except when CD44 expression on these cells was compared to that in stromal populations (for example, fibroblastic cells and endothelial cells), in which case the stromal isolation protocol[55] was used. Prior to staining, red blood cells were lysed using an ammonium chloride–potassium lysis solution.

## Cell staining and flow cytometric analysis

Cell surface staining of single-cell suspensions was performed using fluorescently conjugated antibodies in combination with pMHCI tetramers $L^d$-IE1$_{168-176}$, or $D^b$-M45$_{985-993}$ for the identification of MCMV-specific CD8 T cells, $D^b$-NP$_{366-374}$ for the identification of IAV-specific CD8 T cells, or $K^b$-OVA$_{257-264}$ for the identification of OVA-specific CD8 T cells. Dead cells were excluded using 4′,6-diamidino-2-phenylindole hydrate (DAPI) for live cells or FVS440UV (BD Biosciences) for fixed cells. For hyaluronic acid staining, cells were fixed in 2% paraformaldehyde (PFA) and stained overnight prior to analysis. Cells were acquired on a FACSymphony A3 cell analyser running FACSDiva (BD Biosciences), and data analysis performed with FlowJo software (BD Biosciences). Gating strategies are shown in Extended Data Fig. 10. Details of the antibodies used for flow cytometry are provided in Extended Data Table 3.

## Immunofluorescence

IC-21 cells infected with MCMV (K181) were collected at 4 dpi and the cell suspension was pre-incubated on ice for 30 min with 10% normal goat serum (NGS) plus 2% FCS in PBS to block nonspecific reactivity. The cells were stained with anti-vCD44BP (M-627) followed by anti-mouse IgG biotin and Streptavidin-Alexa Fluor 488. After staining, cells were fixed in 4% PFA, dried onto glass slides and examined by epifluorescence microscopy (Olympus, BX60).

The FRC2 cell line or primary splenic FRCs were grown on glass coverslips or 96-well PhenoPlate (black, optically clear, flat bottom, tissue culture treated), infected with MCMV or ΔvCD44BP viruses for 24 h, fixed in 4% PFA and stained with antibodies, as indicated in figure legends. For vCD44BP immunofluorescent imaging, cells were stained with anti-vCD44BP (clone 7G5, 20 µg ml⁻¹) for 1 h at 37 °C, fixed in 4% PFA for 15 min at room temperature, followed by incubation with biotin-conjugated anti-rat-IgG. Samples were then treated with a tyramide signal amplification kit (Invitrogen), followed by the addition of Alexa488-conjugated Streptavidin. Samples were permeabilized in 0.1% Triton X-100 before the addition of anti-CD44 antibodies, phalloidin and DAPI. Glass coverslips were mounted with Fluoromount Aqueous Mounting Medium (Sigma). Images were acquired using the following confocal instruments: Leica SP5 and SP8 at 1,024 × 1,024 pixels at 64× magnification with 8-bit sensitivity or Leica DMi8 Inverted Microscope with 8-bit sensitivity.

For immunofluorescence studies of spleen sections, mice were humanely killed, and spleens were excised and fixed overnight in periodate-lysine-paraformaldehyde that was prepared as described[59]. Spleens were then transferred into 30% sucrose for 24 h before embedding in Tissue-Tek OCT compound, frozen on isopentane over dry ice and stored at −80 °C. Cryostat sections (6–20 µm thick) were cut and air dried before fixation with −20 °C acetone and quenching with 50 mM ammonium chloride. Sections were blocked with 10% NGS and stained with primary antibodies overnight at 4 °C. The next day, sections were washed prior to staining with secondary antibodies for 1 h at room temperature. Images were acquired using a Leica DMi8 Inverted Microscope with 8-bit sensitivity, Nikon AX R Ti2-E confocal microscope, or Carl Zeiss LSM980 confocal microscope.

Image analysis was performed using the ImageJ software. Mean fluorescence intensities of CD11c, Xcr1, or 33D1 staining were quantified within splenic white pulp regions. The cell morphology index (perimeter²/4π area) was calculated as described[8]. Skeleton analysis was performed as described[60]. First, immunofluorescence images were cropped to include the podoplanin positive area of the white pulp. Images were then de-speckled, thresholded, converted into binary images, skeletonized and analysed using the Skeletonize (2D/3D) and Analyse Skeleton (2D/3D) plugins, respectively. Details of the antibodies used for immunofluorescence are provided in Extended Data Table 4.

## 3D migration assays

Primary FRCs were isolated from C57BL/6J or B6 *Cd44*⁻/⁻ spleens and cultured as described[55]. Dendritic cells were expanded from bone marrow progenitors as described[61], labelled with Cell Tracker Deep Red (Invitrogen) and seeded at a 5:1 ratio with FRCs into a matrix of 1.8 mg ml⁻¹ Collagen I (Corning), 2.6 mg ml⁻¹ Matrigel (Corning) and 10% FBS in alpha-MEM[62]. In some assays, FRCs or dendritic cells were pre-treated with vCD44BP–Fc (20 µg ml⁻¹ vCD44BP–Fc for 40 min at 4 °C) and washed to remove unbound vCD44BP–Fc prior to co-culture. Cultures were imaged with the Opera Phenix Plus High-Content Screening System (PerkinElmer) for 4 h at 37 °C with 5% CO₂. Migration of dendritic cells in contact with FRCs was tracked with ImageJ using the Manual Tracking plugin.

## Quantification of CD44–vCD44BP co-localization

The FRC2 cell line was infected with MCMV or ΔvCD44BP for 24 h, stained with anti-CD44 and anti-vCD44BP (clone 7G5) antibodies, and cells analysed using an Amnis INSPIRE ImageStreamX instrument (Cytek Biosciences). Data analysis was performed using Image Data Exploration and Analysis Software (IDEAS) and rank-weighted co-localization was analysed using the co-localization pipeline in CellProfiler.

## vCD44BP impact on adjuvant-induced LN expansion, immunization and influenza infection

Montanide adjuvant (Seppic) (25% diluted in PBS) was administered by subcutaneous injection in the neck scruff of mice. On days 1 and 3 post-adjuvant treatment, 15 µg of vCD44BP–Fc, control human TRAIL–Fc (a human protein that does not bind in the mouse), or PBS were administered by subcutaneous injection adjacent to the adjuvant injection site. At day 5 post-adjuvant treatment, lymph nodes that drain the injection site (cervical) were isolated, pooled and weighed prior to flow cytometric analysis. As a control, non-draining lymph nodes (inguinal) were also isolated and treated as above.

Antigen-specific immune responses were assessed by emulsifying 40 µg of OVA protein (Merck Life Science) with Montanide adjuvant. On days 1 and 3 post-adjuvant treatment, 15 µg of vCD44BP–Fc, hTRAIL–Fc or PBS were administered by subcutaneous injection adjacent to the adjuvant + OVA injection site. At day 6 post-adjuvant + OVA treatment, cervical and inguinal lymph nodes were isolated and analysed by flow cytometry.

On days −1, 0 and 3 relative to influenza infection, 50 µg of vCD44BP–Fc, control hTRAIL–Fc or PBS were administered by intravenous injection. At 10 dpi, the mediastinal lymph nodes were isolated and single-cell suspensions were prepared and analysed by flow cytometry.

## Quantitative real-time PCR

Quantitative real-time PCR was performed using the SsoAdvanced Universal SYBR Green Supermix using a CFX Connect Real-Time System (Bio-Rad). The ribosomal protein L32 was used as the control housekeeping gene: forward, 5′-CATCGGTTATGGGAGCAAC-3′; reverse, 5′-GCACACAAGCCATCTACTCAT-3′. The following transcripts were detected: CCL19: forward, 5′-CCTGGGTGGATCGCATCA-3′; reverse, 5′-TGCCTTTGTTCTTGGCAGAA-3′, CCL21: forward, 5′-GCAAAGAGGG AGCTAGAAAACAGA-3′; reverse, 5′-TGGACGGAGGCCAGCAT-3′.

## Statistical analysis

All data were analysed and graphed as mean ± s.e.m. or violin plots using Prism (GraphPad), unless otherwise stated. Statistical significance was determined using either a Mann–Whitney $U$-test or a Kruskal–Wallis test with a post hoc Dunn's test for multiple comparisons, as detailed in the figure legends.

## Software and algorithms

The software packages and algorithms used in this study are listed in Extended Data Table 5.

## Reporting summary

Further information on research design is available in the Nature Portfolio Reporting Summary linked to this article.

## Data availability

The data supporting the findings reported in this study are included in the article and extended data. The structure factor file and associated atomic coordinates have been deposited in the Protein Data Bank under accession code 9EJW. Data relating to the structure of mouse CD44 (PDB ID: 2JCP) and m04 (PDB ID: 4PN6) are accessible from the PDB (https://www.rcsb.org/). The m11 sequence is available from GenBank (https://www.ncbi.nlm.nih.gov/) under accession CAP08055.1. Source data are provided with this paper.

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

**Acknowledgements** We thank N. L. La Gruta and S. J. Turner for discussions, critical reading of the manuscript and providing the influenza virus; D. Thiele for expert advice relating to CRISPR–Cas9-based gene deletion; and J. Homman-Ludiye for expert advice relating to micro imaging. We thank S. Pervan at LEI Histology and S. Ross and staff at the Bioresources Animal Facility for technical support. This work was supported by the Lions Eye Institute Flow Cytometry and Histology facilities and by the following Monash University platforms: Monash Animal Research Platform, Monash Macromolecular Crystallization Facility, FlowCore, Monash Micro Imaging, and Monash Histology. Data collection was undertaken using the MX2 beamline at the Australian Synchrotron, part of ANSTO, and made use of the Australian Cancer Research Foundation (ACRF) detector. This work was supported by funding from the National Health and Medical Research Council (NHMRC 1119298 principal research fellowship to M.A.D.-E.; research grant 353640 to A.A.S. and M.A.D.-E.; NHMRC investigator awards GNT2026377 to M.A.D.-E. and GNT2008981 to J.R.) and the Australian Research Council (ARC discovery project grant DP230102854 to M.A.D.-E.). G.R.H. is supported by NIH grant R01 AI175535. The content is solely the responsibility of the authors and does not necessarily represent the official views of the NIH.

**Author contributions** Conceptualization: M.A.D.-E. and C.E.A. with assistance from A.L.F. Experimental design: M.A.D.-E., C.E.A., A.L.F., R.B., J.R., A.A.S., S.D.L., S.N.M. and M.E.W. Investigations: X.Y.X.S., V.V., C.E.A., I.S.S., P.F., S.L.H.v.D., G.E.G.N., R.M.A., F.A.D., R.B., M.H.A., H.L.H., S.D., A.V. and B.G. Data analysis: X.Y.X.S., C.E.A., V.V., P.F., I.S.S., F.A.D., R.B., M.H.A., M.E.W., S.N.M., G.R.H., A.A.S., S.D.L., J.R., A.L.F. and M.A.D.-E. Funding acquisition: M.A.D.-E. and A.A.S. Manuscript writing: M.A.D.-E., C.E.A. and X.Y.X.S., with critical editing from A.L.F., J.R., R.B., S.N.M. and M.E.W., and contributions from all other authors.

**Competing interests** The Lions Eye Institute have submitted a PCT patent application (WO2025080875) relating to stromal CD44 regulation of T cell responses; the application involves M.A.D.-E., C.E.A., R.B., J.R. and A.A.S. The other authors declare no competing interests.

**Additional information**
**Correspondence and requests for materials** should be addressed to Christopher E. Andoniou or Mariapia A. Degli-Esposti.

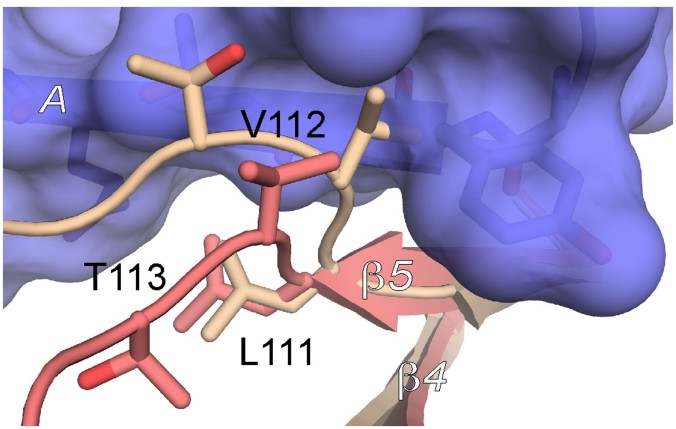

**Extended Data Fig. 1 | An m11-induced conformational change in CD44.**
Overlay of the m11-bound (salmon) and unliganded (sand) forms of mouse
CD44. The view shown is focused on the β5-strand and the β5-β6 loop. m11 is
shown as a blue surface.

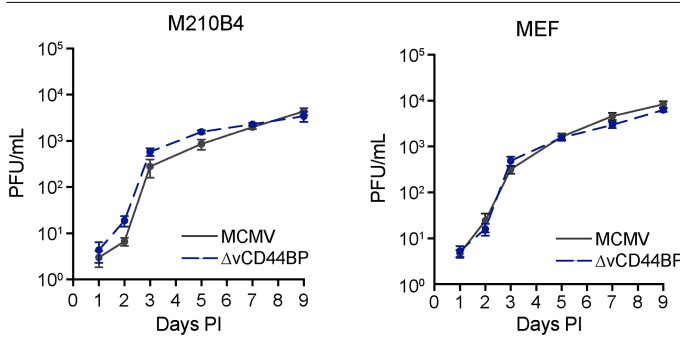

**Extended Data Fig. 2 | Viral infectivity and replication in vitro are not altered by lack of vCD44BP.** The M210B4 stromal cell line or murine embryonic fibroblasts (MEF) were infected in triplicate with MCMV (black) or ΔvCD44BP (blue dashed) at MOI 0.2 and viral loads quantified by plaque assay at the indicated timepoints. Graphs show mean ± SEM.

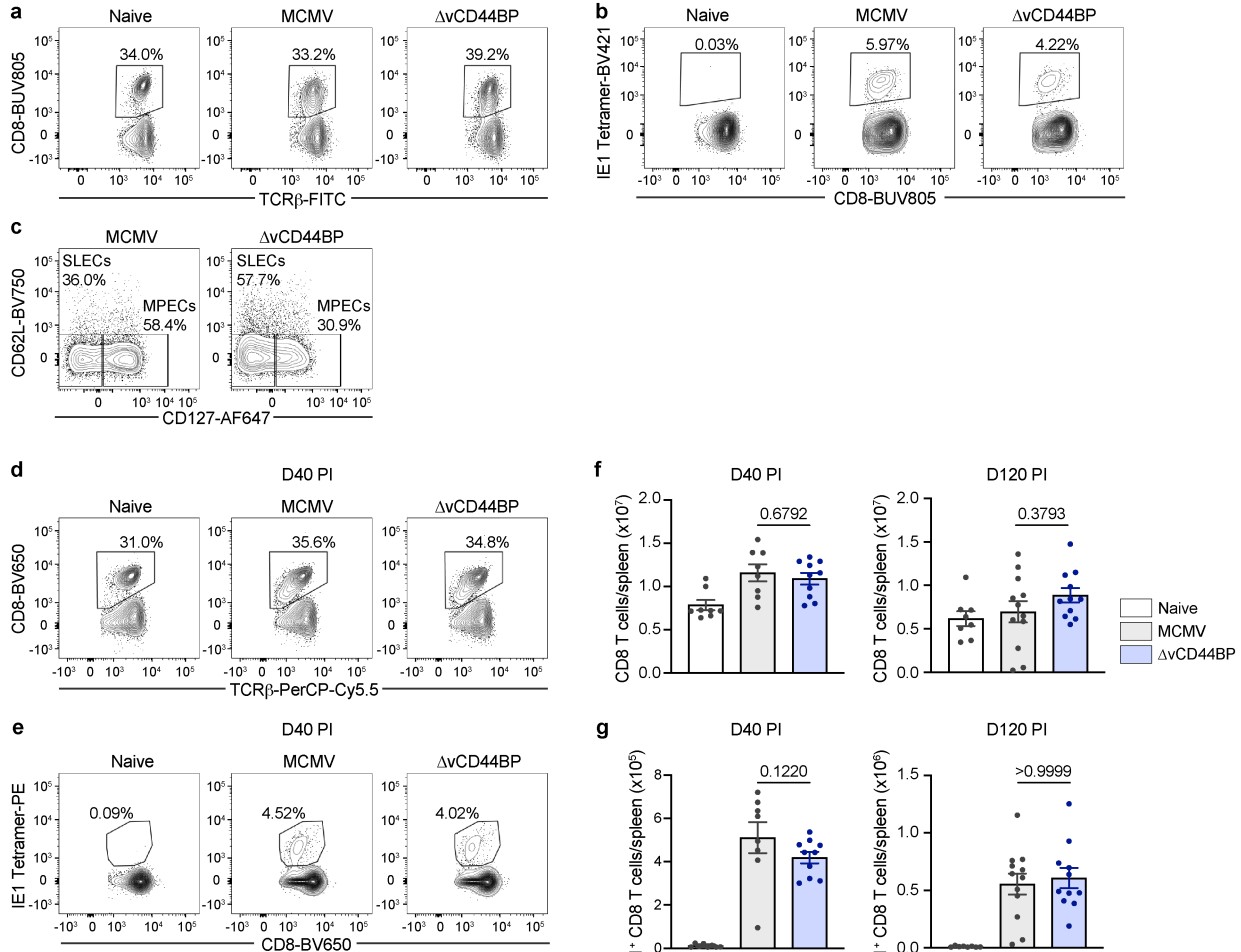

**Extended Data Fig. 3 | vCD44BP constrains effector but not memory CD8 T cell numbers.** Representative flow cytometry plots showing (**a**) CD8 T cells, (**b**) IE1+ virus specific CD8 T cells and (**c**) IE1+ SLEC (CD127lo) and MPEC (CD127hi) subsets in spleens of BALB/c mice infected with MCMV or ΔvCD44BP (day 7 PI); plots are concatenated from one experiment (naïve n = 2; MCMV and ΔvCD44BP n = 4) and representative of three independent experiments. Representative flow cytometry plots showing (**d**) CD8 T cells and (**e**) IE1+ virus specific CD8 T cells in spleens of BALB/c mice infected with MCMV or ΔvCD44BP (day 40 PI);

plots are concatenated from one experiment (naïve n = 2; MCMV and ΔvCD44BP n = 4) and representative of two independent experiments. Number of (**f**) CD8 T cells and (**g**) IE1+ virus specific CD8 T cells in spleens of BALB/c mice infected with MCMV or ΔvCD44BP at days 40 PI and 120 PI; pooled from two to four independent experiments (D40 PI: naïve n = 8; MCMV n = 8; ΔvCD44BP n = 10; D120 PI: naïve n = 8; MCMV n = 12; ΔvCD44BP n = 11). All mice were infected with 5 × 10^3 PFU. Mean is shown in flow plots; graphs show mean ± SEM; significance tested by two-sided Mann-Whitney.

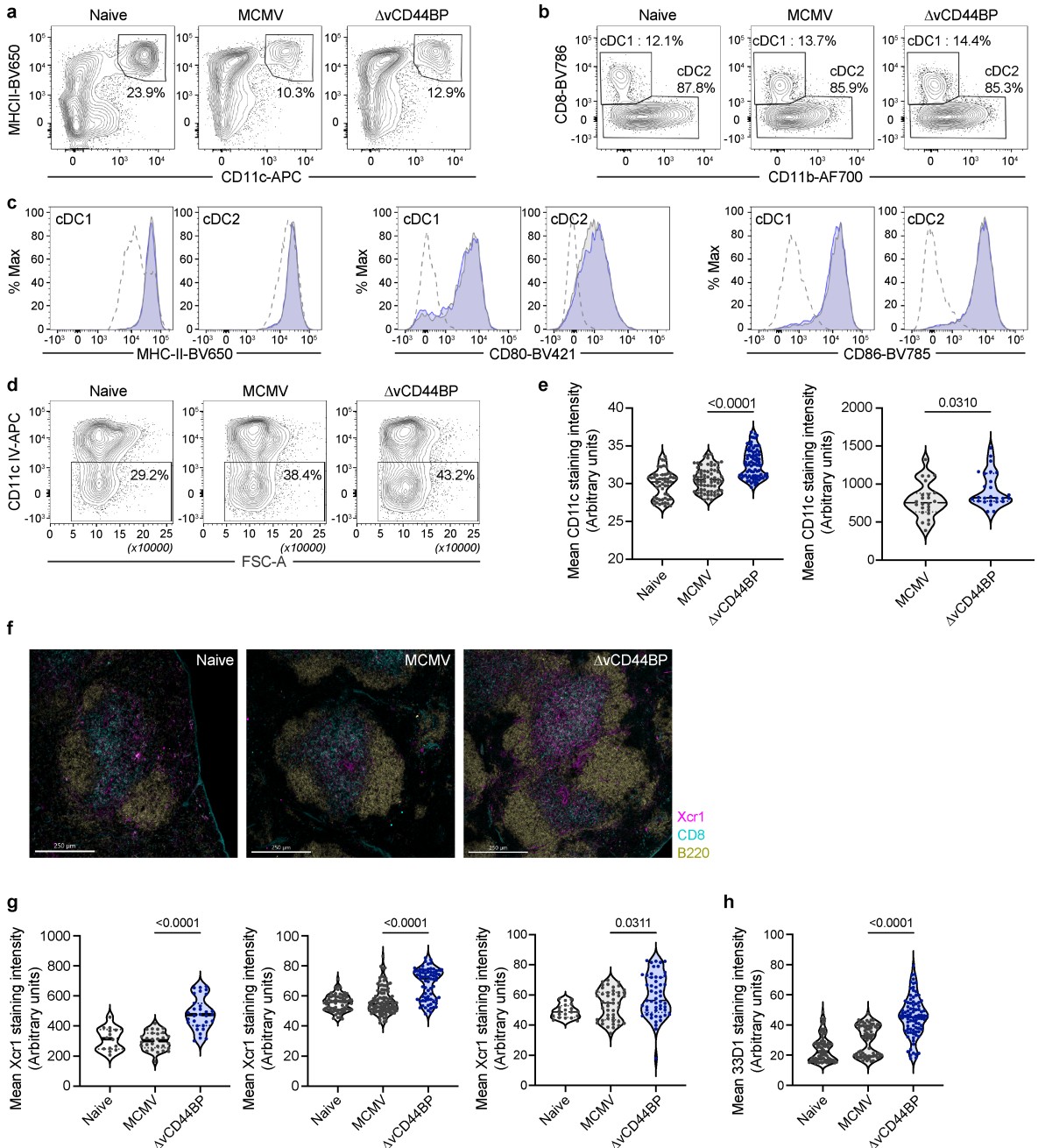

**Extended Data Fig. 4 | vCD44BP constrains DC migration but does not alter DC activation.** Representative flow cytometry plots showing (**a**) dendritic cells and (**b**) cDC1 and cDC2 subsets in spleens of BALB/c mice infected with MCMV or ΔvCD44BP (day 2 PI); plots are concatenated from one experiment (naïve n = 2; MCMV and ΔvCD44BP n = 4) and representative of two independent experiments. (**c**) Expression of MHCII, CD80 and CD86 on cDC1 (left) and cDC2 (right) at day 2 PI with MCMV (grey histogram) or ΔvCD44BP (blue); naïve uninfected control (dashed histogram); histograms are concatenated from one experiment (naïve n = 2; MCMV and ΔvCD44BP n = 3) and representative of two independent experiments. (**d**) Representative flow plots with boxed IV negative fractions of splenic CD11c⁺MHCII⁺ dendritic cells at 36 h PI; plots are concatenated from one experiment (naïve n = 2; MCMV and ΔvCD44BP n = 4

and representative of two independent experiments. (**e**) Mean CD11c staining intensities within the splenic white pulp at day 2 PI; data from two independent experiments are plotted. (**f**) Representative images (processed with Imaris for display purposes only) of spleen sections from naïve mice or mice infected with MCMV or ΔvCD44BP (day 2 PI) stained with anti-B220, anti-CD8 and anti-Xcr1 mAbs. (**g**) Mean Xcr1 (identifies cDC1) staining intensities within the white pulp are plotted and three independent experiments are shown. (**h**) Mean 33D1 (identifies cDC2) staining intensities in spleen sections are plotted from one experiment. All mice were infected with 5 × 10³ PFU. Mean is shown in flow plots; violin plots show median and quartiles; significance tested by two-sided Mann-Whitney.

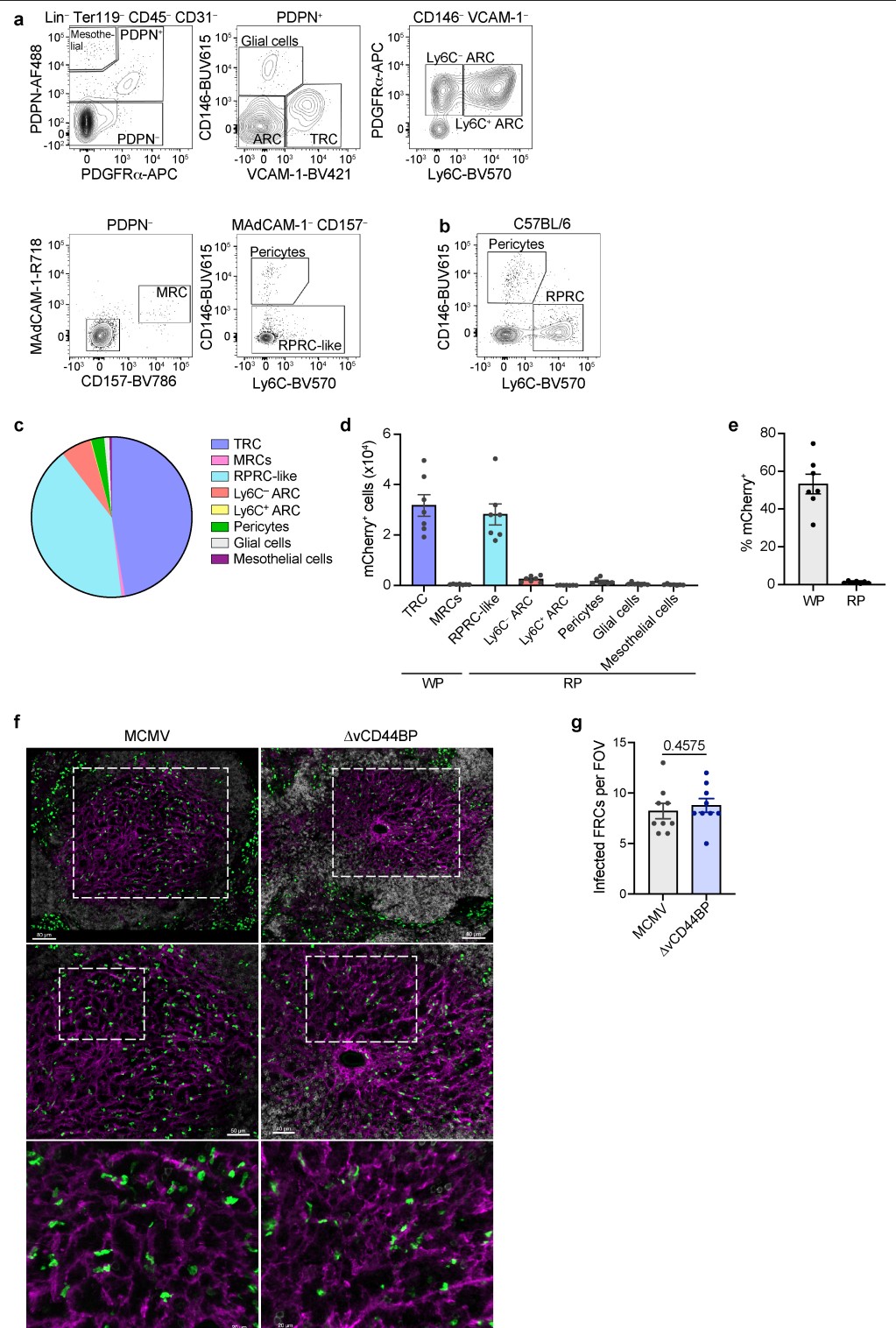

**Extended Data Fig. 5 | FRCs in the splenic WP are a major target of MCMV infection.** (**a**) Gating strategy used to characterize mouse splenic fibroblasts. (**b**) Representative flow cytometry plot showing Ly6C versus CD146 staining in the PDPN⁻ MAdCAM-1⁻ CD157⁻ fraction of C57BL/6 FRCs. (**c**) Pie chart showing the fraction and (**d**) graph showing total numbers for stromal cell populations infected by MCMV:mCherry (TRCs, T cell zone reticular cells; MRCs, marginal zone reticular cells; RPRC-like, red-pulp reticular cell-like; ARCs, adventitial reticular cells; WP, white pulp; RP, red pulp). (**e**) Proportion of stromal cells infected by MCMV in the WP compared to the RP; data (**c**–**e**) pooled from two independent experiments (n = 7 mice/group). (**f**) Representative confocal images (processed with Imaris for display purposes) of spleen sections from mice infected with MCMV or ΔvCD44BP (day 2 PI) stained with anti-B220 (grey), anti-PDPN (magenta) and anti-IE1 (green) mAbs; from top to bottom, each image shows an enlargement of relevant areas as demarcated by rectangles; representative of two independent experiments. (**g**) Quantification of infected FRC per field of view in spleens at day 2 PI (n = 9/group from two independent experiments). All mice were infected with $5 \times 10^3$ PFU, except in (**c**–**e**) where $1 \times 10^4$ PFU was used. Graphs show mean ± SEM ; in (**g**) significance tested by two-sided Mann-Whitney.

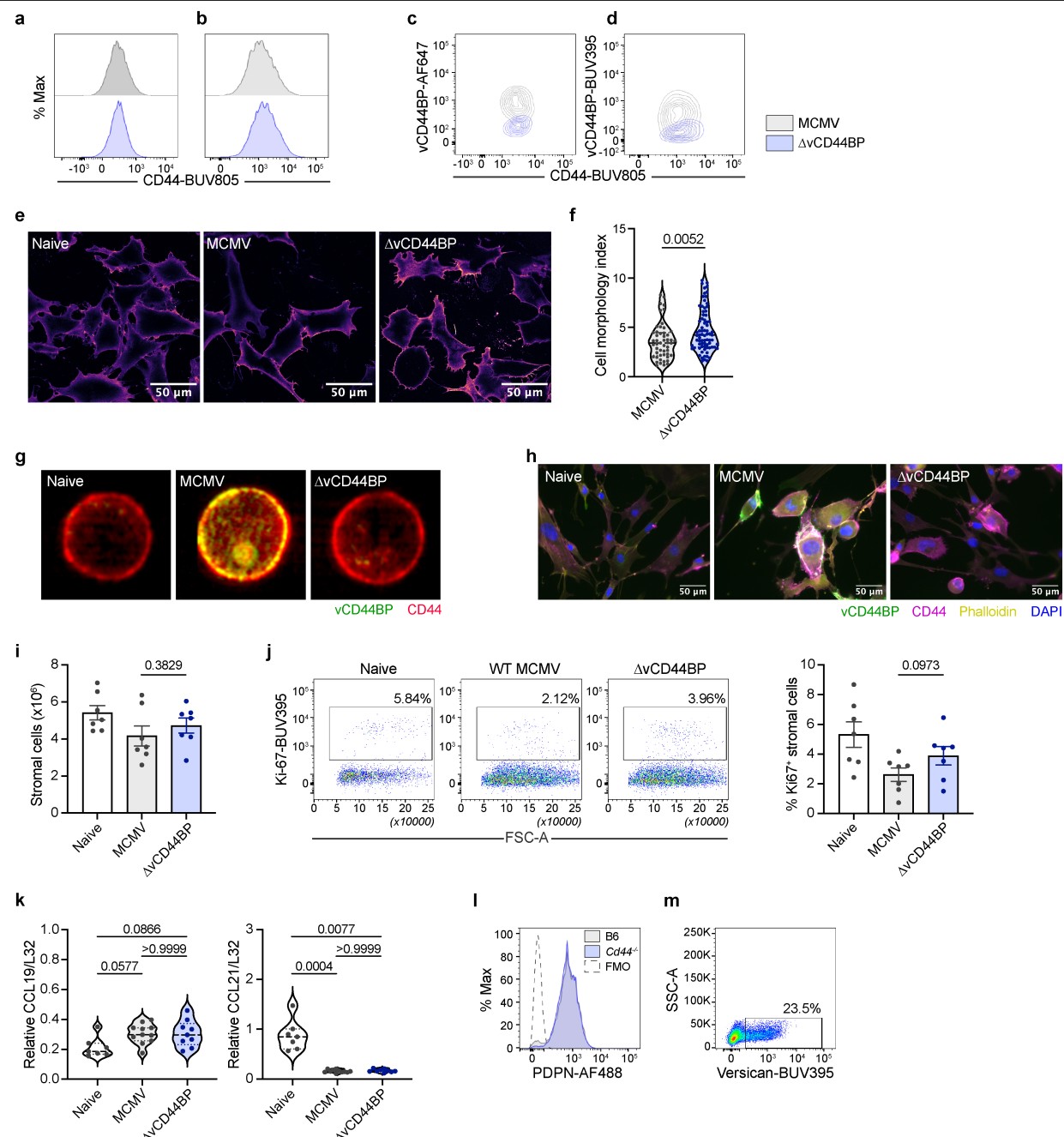

**Extended Data Fig. 6 | Impact of vCD44BP on FRCs. (a–d)** Flow cytometry data showing CD44 and vCD44BP expression on (**a, c**) the FRC2 cell line, or (**b, d**) primary splenic FRCs infected with MCMV or ΔvCD44BP (MOI 3, 24 h). (**e**) Representative confocal images of CD44 staining on uninfected and infected (MCMV or ΔvCD44BP; MOI 3, 24 h) FRC2; representative of three independent experiments. (**f**) Cell morphology index of FRC2 infected with MCMV or ΔvCD44BP (24 h); pooled from two independent experiments with a minimum of 50 infected FRCs per condition examined. (**g**) Representative images from Image StreamX; representative of two independent experiments. (**h**) Confocal images of uninfected and infected FRCs; the uninfected panel is shown for comparison to the infected panels in Fig. 4g, also reproduced here. (**i**) Number of FRCs (Lin⁻ Ter119⁻ CD45⁻ CD31⁻ PDPN⁺) in spleens of naïve mice or mice infected with MCMV or ΔvCD44BP (day 2 PI). (**j**) Concatenated pseudocolour plots of Ki-67 staining showing percentage of Ki-67⁺ FRCs; data (**i, j**) pooled from two independent experiments (n = 7/group). (**k**) Relative expression of CCL19 and CCL21 mRNA in spleen homogenates of naïve mice or mice infected with MCMV or ΔvCD44BP (day 2 PI); combined from three independent experiments (naïve n = 7, MCMV n = 11, ΔvCD44BP n = 9). (**l**) Representative histograms showing PDPN expression on primary FRCs from B6 and B6.$Cd44^{-/-}$ mice. (**m**) Representative flow cytometry plot showing Versican staining on BMDCs used in migration assays. Versican is a large proteoglycan whose G1 domain binds HA with high affinity, allowing HA expression to be detected [PMID: 28504698]. All mice were infected with 5x10³ PFU. Mean is shown in flow plots; graphs show mean ± SEM; violin plots show median and quartiles. In (**f,i** and **j**) significance tested by two-sided Mann-Whitney; in (**k**) Kruskal-Wallis was used.

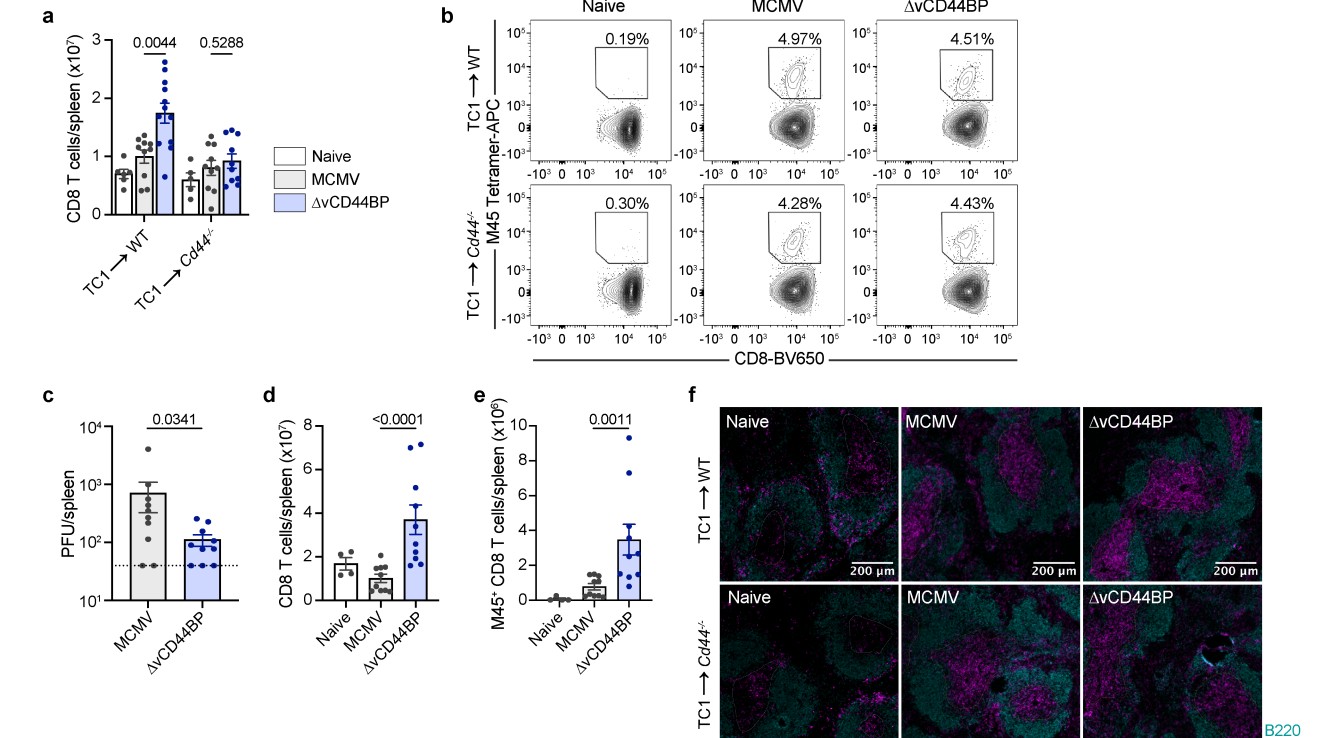

**Extended Data Fig. 7 | Stromal CD44 affects DC migration and antiviral CD8 T cell responses.** (**a**) Number of CD8 T cells in spleens of chimeric mice infected with MCMV, ΔvCD44BP (day 7 PI) or uninfected controls; combined from three independent experiments (TC1 → WT: naïve n = 6, MCMV n = 10; ΔvCD44BP n = 12; TC1 → $Cd44^{-/-}$: naïve n = 5, MCMV and ΔvCD44BP n = 10). (**b**) Representative flow cytometry plots showing splenic M45$^+$ virus specific CD8 T cells from chimeric mice infected with MCMV, ΔvCD44BP (day 7 PI) or uninfected controls; plots concatenated from one experiment (naïve n = 2; MCMV and ΔvCD44BP n = 4)

and representative of three independent experiments. (**c**) Viral loads, (**d**) number of CD8 T cells and (**e**) number of M45$^+$ virus specific CD8 T cells in spleens of B6.TC1 mice infected with MCMV, ΔvCD44BP (day 7 PI) or uninfected controls; combined from two independent experiments (naïve n = 4; MCMV and ΔvCD44BP n = 10). (**f**) Representative images of CD11c staining in spleen sections from naïve or infected (day 2 PI) chimeric mice (3 mice/group). All mice were infected with 5 × 10$^3$ PFU; Mean is shown in flow plots; graphs show mean ± SEM; significance tested by two-sided Mann-Whitney.

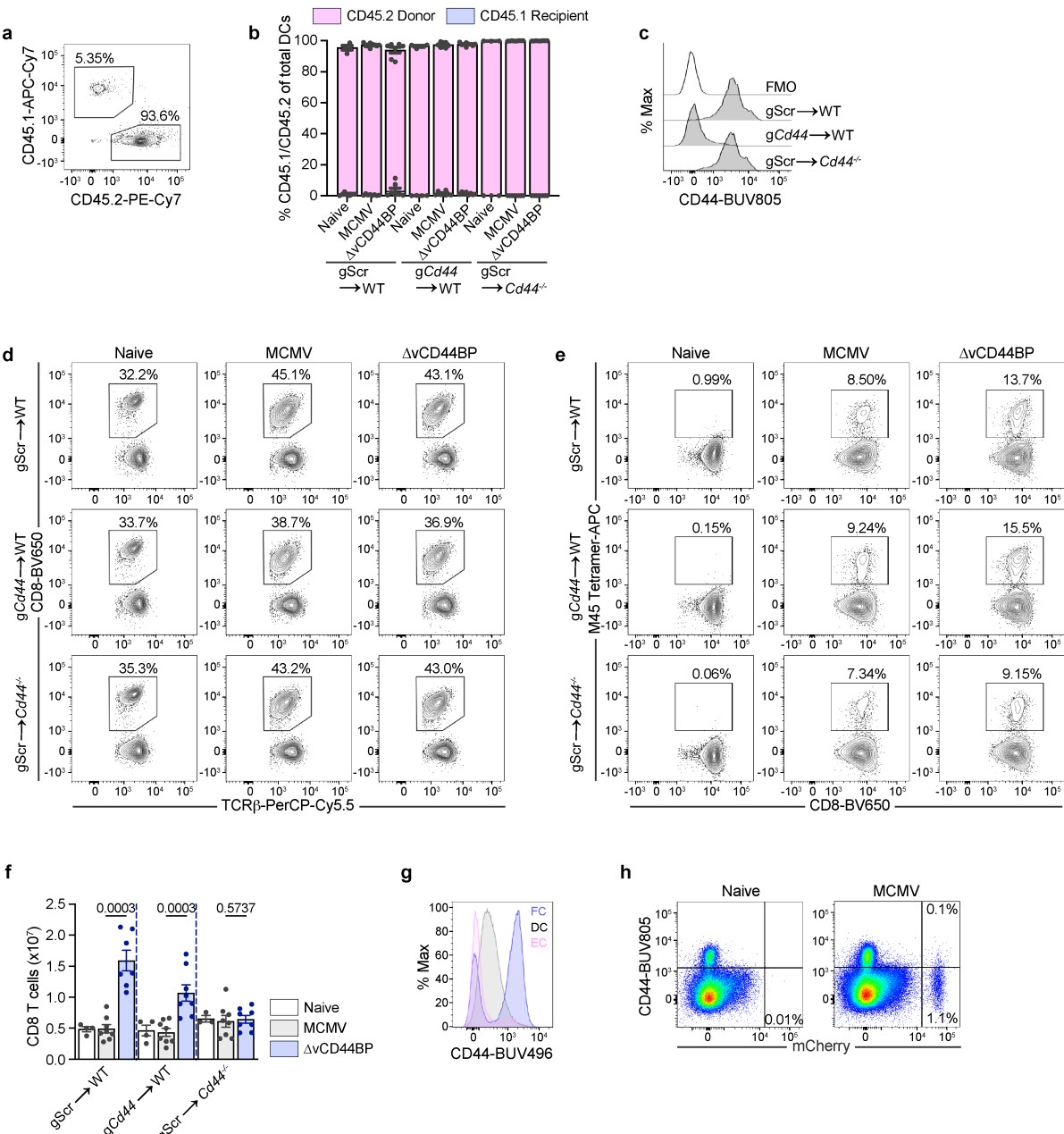

**Extended Data Fig. 8 | vCD44BP–CD44 interactions in stromal cells but not hematopoietic cells affect the generation of antiviral CD8 T cell responses.** (**a**) Representative flow cytometry plot showing proportions of donor (CD45.2) and recipient (CD45.1) derived haematopoietic cells in spleens of chimeric mice generated using the CRISPR-Cas9 based approach. (**b**) Proportions of donor (CD45.2) and recipient (CD45.1) derived DCs in spleens of chimeric mice. (**c**) Representative histograms showing CD44 expression on DCs isolated from spleens of chimeric mice by mechanical dissociation. (**d**) Representative flow cytometry plots of CD8 T cells and (**e**) M45⁺ virus specific CD8 T cells from chimeric mice infected with MCMV, ΔvCD44BP (day 7 PI) or uninfected controls. (**f**) Number of CD8 T cells in spleens of chimeric mice infected with MCMV, ΔvCD44BP (day 7 PI) or uninfected controls. Data in (**a**–**f**) combined from two independent experiments (gScr→WT: MCMV n = 8, ΔvCD44BP n = 7,

naïve n = 4; g*Cd44* → WT: MCMV n = 8, ΔvCD44BP n = 8, naïve n = 4; gScr→*Cd44*⁻/⁻: MCMV n = 8, ΔvCD44BP n = 8, naïve n = 3) and representative flow plots concatenated from one experiment (naïve n = 2; MCMV and ΔvCD44BP n = 4). (**g**) Representative histograms showing CD44 expression on splenic stromal fibroblastic cells (FC), endothelial cells (EC) and dendritic cells (DC) isolated using a stromal isolation protocol; representative of three independent experiments (naïve n = 6; MCMV and ΔvCD44BP n = 11). (**h**) Representative flow cytometry plots showing expression of CD44 and mCherry by splenic EC from uninfected or MCMV:mCherry-infected BALB/c mice (24 h PI). All mice were infected with 5 × 10³ PFU, except in (**h**) where 1 × 10⁴ PFU was used. Mean is shown in flow plots; graphs show mean ± SEM; significance tested by two-sided Mann-Whitney.

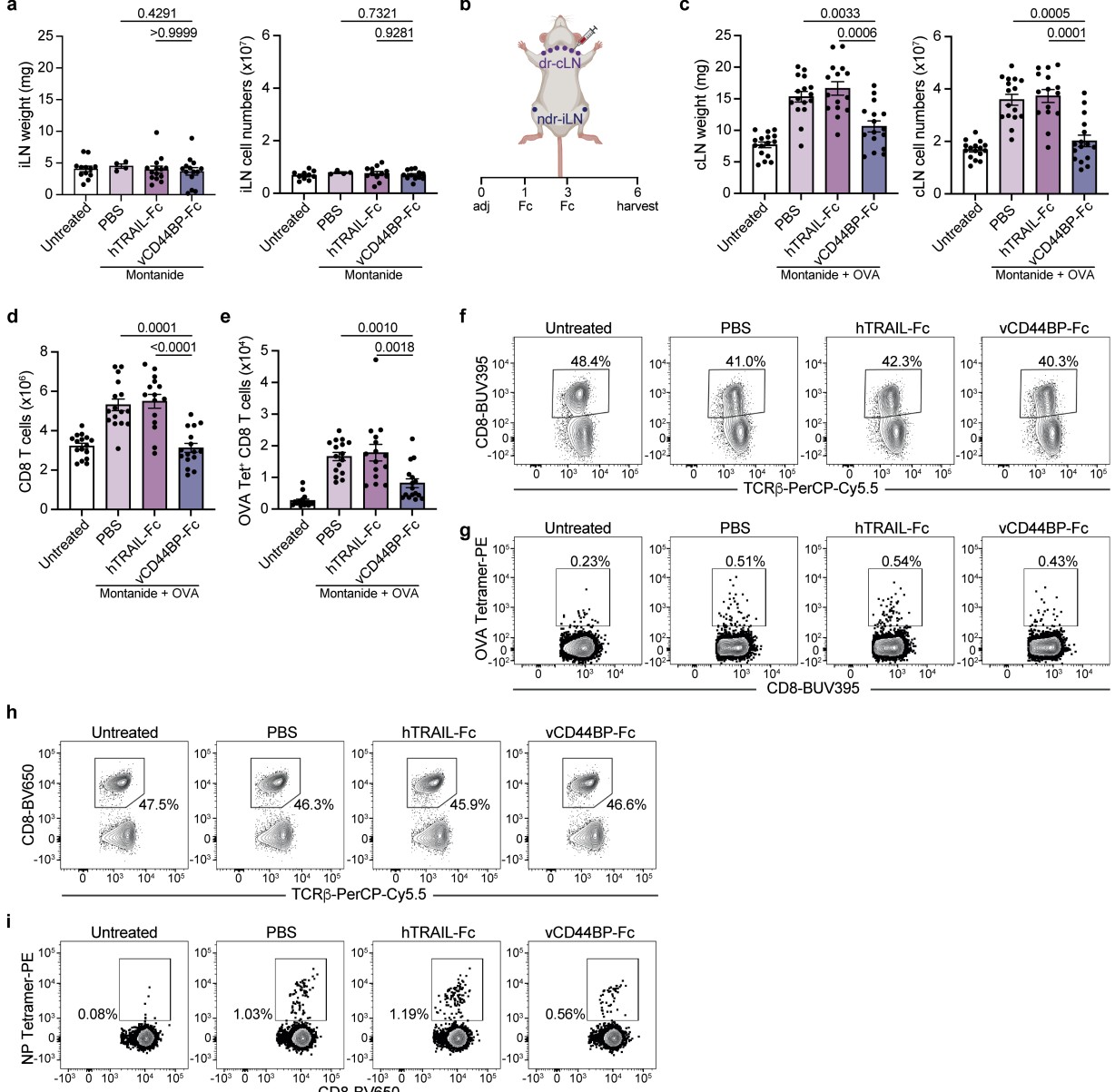

**Extended Data Fig. 9 | vCD44BP inhibits immune responses generated by adjuvant, immunization, or influenza infection.** (**a**) Weights (untreated n = 12; PBS n = 4; hTRAIL-Fc and vCD44BP-Fc n = 14) and total cell numbers (untreated n = 10; PBS n = 4; hTRAIL-Fc and vCD44BP-Fc n = 12) in the non-draining inguinal lymph nodes (iLN) of untreated and Montanide-treated mice which had received either PBS, hTRAIL-Fc or vCD44BP-Fc. (**b**) Schematic of experimental set-up: evaluating the impact of interference with CD44 function by soluble vCD44BP-Fc treatment in a vaccination setting. Created in BioRender. Sng, X. (2025) https:// BioRender.com/hgpsca9. (**c**) Weights and total cell numbers in the draining cervical lymph nodes (cLN) of untreated mice and mice treated with Montanide plus OVA that received either PBS, hTRAIL-Fc or vCD44BP-Fc. Number of (**d**) CD8 T cells, and (**e**) OVA-specific CD8 T cells in the cLN. Data (**c–e**) pooled from four independent experiments (untreated n = 16; PBS n = 16; hTRAIL-Fc n = 15 and vCD44BP-Fc n = 16 mice/group). (**f**) Representative flow cytometry plots of CD8 T cells and (**g**) OVA-specific CD8 T cells from cervical lymph nodes of untreated and Montanide plus OVA treated mice that received either PBS, hTRAIL-Fc or vCD44BP-Fc; plots concatenated from one experiment (naïve, MCMV and ΔvCD44BP, n = 4/group) and representative of four independent experiments. Representative flow plots showing (**h**) CD8 T cells and (**i**) NP-specific CD8 T cells from mediastinal lymph nodes of untreated and flu-infected mice that received either PBS, hTRAIL-Fc or vCD44BP-Fc; plots concatenated from one experiment (naïve, MCMV and ΔvCD44BP, n = 4/group) and representative of three independent experiments. Mean is shown in flow plots; graphs show mean ± SEM; significance tested by Kruskal-Wallis.

**a**    Flow cytometry gating strategy for CD8 T cells

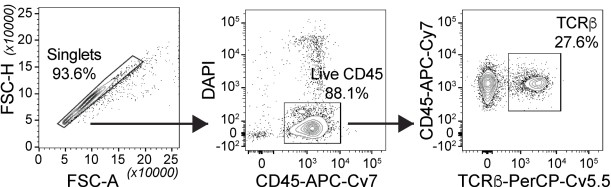

**b**    Flow cytometry gating strategy for dendritic cells

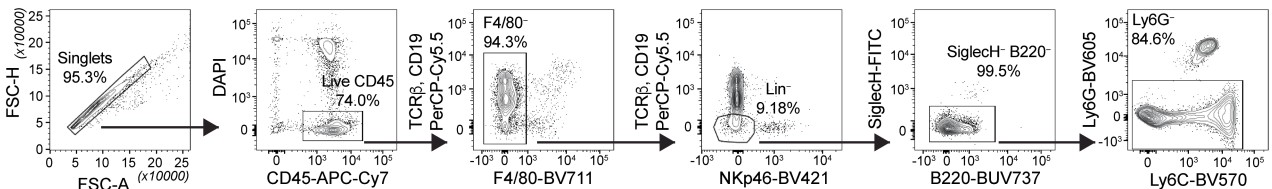

**c**    Flow cytometry gating strategy for fibroblastic stromal cells and endothelial cells

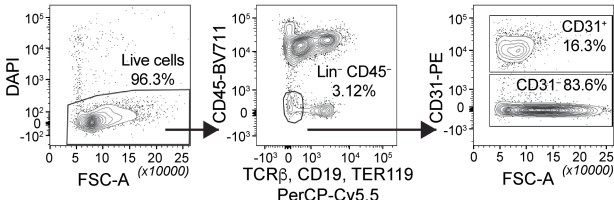

**Extended Data Fig. 10 | Representative flow cytometry gating strategies for the identification of immune, stromal and endothelial cell subsets.** (**a**) Gating strategy for phenotyping of CD8 T cells in spleen and lymph nodes in main Figs. 3a–d and 5c,f,l,m. (**b**) Gating strategy for phenotyping of splenic DCs in Fig. 3e,f. (**c**) Gating strategy for phenotyping of splenic stromal cells and endothelial cells in Fig. 4c.

**Extended Data Table 1 | X-ray crystallographic data collection and refinement statistics**

**Data collection statistics**

| | |
|---|---|
| Temperature (K) | 100 |
| X-ray source | MX2 Australian Synchrotron |
| Spacegroup | $P2_12_12_1$ |
| Cell dimensions (Å) | 34.84, 105.77, 113.00 |
| | 90, 90, 90 |
| Resolution (Å) | 47.90-1.40 (1.48-1.40) |
| Total number of observations | 1,126,366 (159,306) |
| No. unique observations | 83,336 (11,969) |
| Multiplicity | 13.5 (13.3) |
| Data completeness | 99.9 (99.7) |
| $I/\sigma_I$ | 12.4 (1.6) |
| $R_{merge}$ (%) | 10.6 (161.3) |
| $R_{pim}$ (%) | 3.0 (45.4) |
| CC (1/2) | 0.999 (0.595) |

**Refinement statistics**

| | |
|---|---|
| Non-hydrogen atoms | |
| Protein | 2,227 |
| Sugar | 164 |
| Water | 408 |
| $^1R_{factor}$ (%) | 18.38 |
| $R_{free}$ (%) | 20.24 |
| r.m.s.d from ideality | |
| Bond lengths (Å) | 0.007 |
| Bond angles (°) | 0.98 |
| Ramachandran plot | |
| Favored regions (%) | 98 |
| Allowed regions (%) | 2 |
| Disallowed regions (%) | 0 |
| B factor, all atoms (Å$^2$) | 23 |

$^1R_{factor} = \Sigma_{hkl}||F_o| - |F_c||/\Sigma_{hkl}|F_o|$ for all data excluding the 5% that comprised the $R_{free}$ used for cross-validation.

**Extended Data Table 2 | List of sgRNA guides for CRISPR-mediated deletion of CD44**

| sgRNA guide | Source |
| --- | --- |
| *Cd44* sgRNA1: 5' CAGUCCGGGAGAUACUGUAG 3' | Synthego |
| *Cd44* sgRNA2: 5' CGAGGAUAUAUACUCCUGUG 3' | Synthego |
| non-targeting sgRNA: 5' GCACUACCAGAGCUAACUCA 3' | Synthego |

**Extended Data Table 3 | List of antibodies used for flow cytometry**

| Reagent or Resource | Source | Identifier | Dilution |
|---|---|---|---|
| BD Horizon™ BUV661 Rat Anti-Mouse CD4 | BD Biosciences | 612974 | 1:400 |
| BD Horizon™ BUV395 Rat Anti-Mouse CD8 | BD Biosciences | 563786 | 1:400 |
| BD Horizon™ BUV805 Rat Anti-Mouse CD8 | BD Biosciences | 612898 | 1:200 |
| BD Horizon™ BV650 Rat Anti-Mouse CD8 | BD Biosciences | 563234 | 1:200 |
| BD Horizon™ BV786 Rat Anti-Mouse CD8 | BD Biosciences | 563332 | 1:200 |
| BD Pharmingen™ Alexa Fluor® 700 Rat Anti-CD11b | BD Biosciences | 557960 | 1:500 |
| BD Pharmingen™ APC Hamster Anti-Mouse CD11c | BD Biosciences | 550261 | 1:100 |
| APC/Cyanine7 Hamster Anti-Mouse CD11c | Biolegend | 117324 | 1:100 |
| BD Pharmingen™ PerCP-Cy™5.5 Rat Anti-Mouse CD19 | BD Biosciences | 551001 | 1:100 |
| eBioscience™ CD31 Monoclonal Antibody (390), PE | Thermo Fisher Scientific | 12-0311-82 | 1:400 |
| BD OptiBuild™ BUV496 Rat Anti-Mouse CD44 | BD Biosciences | 741057 | 1:200 |
| BD OptiBuild™ BUV805 Rat Anti-Mouse CD44 | BD Biosciences | 741921 | 1:400 |
| BD Horizon™ BV711 Rat Anti-Mouse CD45 | BD Biosciences | 563709 | 1:400 |
| BD Pharmingen™ APC-Cy7™ Rat Anti-Mouse CD45 | BD Biosciences | 561037 | 1:800 |
| BD Pharmingen™ APC-Cy7™ Rat Anti-Mouse CD45.1 | BD Biosciences | 560579 | 1:100 |
| BD Pharmingen™ PE-Cy7™ Rat Anti-Mouse CD45.2 | BD Biosciences | 560696 | 1:100 |
| BD Horizon™ BUV737 Rat Anti-Mouse CD45R/B220 | BD Biosciences | 612838 | 1:800 |
| BD Horizon™ BV421 Hamster Anti-Mouse CD80 | BD Biosciences | 562611 | 1:800 |
| Brilliant Violet 785™ Anti-Mouse CD86 | Biolegend | 105043 | 1:200 |
| BD OptiBuild™ BV421 Rat Anti-Mouse CD106 | BD Biosciences | 740019 | 1:1600 |
| Alexa Fluor® 647 Anti-Mouse CD127 | Biolegend | 135020 | 1:100 |
| BD Horizon™ BV786 Rat Anti-Mouse CD127 | BD Biosciences | 563748 | 1:100 |
| BD Pharmingen™ APC Rat Anti-Mouse CD140a | BD Biosciences | 562777 | 1:100 |
| BD OptiBuild™ BUV615 Rat Anti-Mouse CD146 | BD Biosciences | 751402 | 1:200 |
| BD OptiBuild™ R718 Mouse Anti-Mouse CD157 | BD Biosciences | 741012 | 1:400 |
| BD Horizon™ BV421 Rat Anti-Mouse CD335 | BD Biosciences | 562850 | 1:100 |
| Brilliant Violet 711™ anti-mouse F4/80 Antibody | Biolegend | 123147 | 1: 100 |
| BD Horizon™ BV650 Rat Anti-Mouse I-A/I-E | BD Biosciences | 563415 | 1:400 |
| BD Horizon™ BUV395 Mouse Anti-Ki-67 | BD Biosciences | 564071 | 1:200 |
| BV570™ Rat Anti-Mouse Ly-6C | Biolegend | 128030 | 1:200 |
| BD Horizon™ BV605 Rat Anti-Mouse Ly-6G | BD Biosciences | 563005 | 1:100 |
| BD OptiBuild™ Rat Anti-Mouse MAdCAM-1 | BD Biosciences | 752198 | 1:100 |
| Mouse Anti-Immediate-Early 1 protein (Clone 6/58/1) | In house | - | 1:100 |
| BD Horizon™ BV421 Mouse Anti-Mouse NK-1.1 | BD Biosciences | 562921 | 1:100 |
| Alexa Fluor® 488 Anti-Mouse Podoplanin | Biolegend | 127406 | 1:100 |
| eBioscience™ Siglec H Monoclonal antibody (eBio44c), FITC | Thermo Fisher Scientific | 11-0333-82 | 1:200 |
| BD Pharmingen™ FITC Hamster Anti-Mouse TCRβ Chain | BD Biosciences | 553171 | 1:100 |
| PerCP/Cyanin5.5 Anti-Mouse TCRβ chain | Biolegend | 109228 | 1:100 |
| BD Pharmingen™ PerCP-Cy™5.5 Rat Anti-Mouse TER119 | BD Biosciences | 560512 | 1:100 |
| Rat Anti-MCMV vCD44BP (7G5) | In house | - | 1:100 |
| Biotinylated Versican G1 Domain | Echelon Biosciences | G-HA02 | 3 μg/mL |
| Biotin-SP (long spacer) AffiniPure™ Donkey Anti-Rat IgG (H+L) | Jackson ImmunoResearch | 712-065-153 | 1:500 |
| Biotin conjugated Goat Anti-Human Fc | Jackson ImmunoResearch | 109-066-098 | 1:1000 |
| BD Horizon™ BUV395 Streptavidin | BD Biosciences | 564176 | 1:100 |
| BD Pharmingen™ PE-Cy™7 Streptavidin | BD Biosciences | 557598 | 1:100 |

**Extended Data Table 4 | List of antibodies used for immunofluorescence**

| Reagent or Resource | Source | Identifier | Dilution |
|---|---|---|---|
| Alexa Fluor® 647 Rat Anti-Mouse CD8α | Biolegend | 100724 | 1:100 |
| Biotin Rat Anti-Mouse CD8α | Biolegend | 100704 | 1:100 |
| Alexa Fluor® 647 Rat Anti-Mouse/Human CD45R/B220 | Biolegend | 103226 | 1:200 |
| Purified Hamster Anti-Mouse CD11c (N418) | †Gift | - | 1:100 |
| Biotin Hamster Anti-Mouse CD11c | Biolegend | 117304 | 1:100 |
| Biotin Rat Anti-Mouse XCR1 | Biolegend | 148212 | 1:100 |
| BD Pharmingen™ Purified Rat Anti-Mouse CD44 | BD Biosciences | 550538 | 1:100 |
| Alexa Fluor® 647 Rat Anti-Mouse/Human CD44 | Biolegend | 103018 | 1:100 |
| Mouse Anti-Immediate-Early 1 protein (clone 6/58/1) | In house | - | 1:200 |
| Rhodamine Phalloidin Reagent | Abcam | Ab235138 | 1:500 |
| APC Hamster Anti-Mouse Podoplanin | Biolegend | 127410 | 1:100 |
| Purified Hamster Anti-Mouse Podoplanin | Biolegend | 127401 | 1:100 |
| Rat Anti-MCMV vCD44BP (7G5) | In house | - | - |
| Mouse Anti-MCMV vCD44BP (M-627) | In house | - | - |
| Anti-Rat IgG Alexa Fluor™ 647 | Biolegend | 405416 | 1:200 |
| Biotin-SP (long spacer) AffiniPure™ Donkey Anti-Rat IgG (H+L) | Jackson ImmunoResearch | 712-065-153 | 1:500 |
| Invitrogen Goat Anti-Mouse IgG (H+L) Alexa Fluor™ 546 | Thermo Fisher Scientific | A-11030 | 1:200 |
| Alexa Fluor™ 488 Streptavidin | Biolegend | 405235 | 1:800 |
| Streptavidin Alexa Fluor™ 555 | Thermo Fisher Scientific | S21381 | 1:500 |
| Streptavidin Brilliant Violet 421™ | Biolegend | 405225 | 1:500 |
| Goat Anti-Hamster IgG (H+L) Alexa Fluor™ 647 | Thermo Fisher Scientific | A21451 | 1:200 |

†Provided by Meredith O'Keefe, Monash University, Australia

**Extended Data Table 5 | Software and Algorithms used in the study**

## Software and Algorithms

| | | |
|---|---|---|
| Buster release20180515 | https://www.globalphasing.com/buster/ | RRID:SCR_015653 |
| CCP4 v7.0.023 | http://www.ccp4.ac.uk/ | RRID:SCR_007255 |
| CellProfiler v4.2.6 | https://cellprofiler.org/ | RRID:SCR_007358 |
| Coot v0.8.7 | http://www2.mrc-lmb.cam.ac.uk/personal/pemsley/coot/ | RRID:SCR_014222 |
| FlowJo v10 | https://www.flowjo.com | RRID: SCR_008520 |
| Graphpad Prism v10.6 | https://www.graphpad.com/scientific-software/prism/ | RRID: SCR_002798 |
| IDEAS® v6.2.187.0 | https://www.merckmillipore.com/ | — |
| Image J v1.54f or v2.14 | https://imagej.net/ij/index.html | RRID:SCR_003070 |
| Imaris v10.2.0 | https://imaris.oxinst.com/packages | RRID: SCR_007370 |
| Phaser v2.7.17 | https://www.phenix-online.org/documentation/reference/phaser.html | RRID:SCR_014219 |
| PyMOL v2.1.1 | http://www.pymol.org/ | RRID:SCR_000305 |
| Scrubber v2.0 | http://www.biologic.com.au/scrubber.html | RRID:SCR_015745 |

# Reporting Summary

## Statistics

For all statistical analyses, confirm that the following items are present in the figure legend, table legend, main text, or Methods section.

| n/a | Confirmed | |
|---|---|---|
| ☐ | ☒ | The exact sample size (*n*) for each experimental group/condition, given as a discrete number and unit of measurement |
| ☐ | ☒ | A statement on whether measurements were taken from distinct samples or whether the same sample was measured repeatedly |
| ☐ | ☒ | The statistical test(s) used AND whether they are one- or two-sided<br>*Only common tests should be described solely by name; describe more complex techniques in the Methods section.* |
| ☒ | ☐ | A description of all covariates tested |
| ☐ | ☒ | A description of any assumptions or corrections, such as tests of normality and adjustment for multiple comparisons |
| ☐ | ☒ | A full description of the statistical parameters including central tendency (e.g. means) or other basic estimates (e.g. regression coefficient) AND variation (e.g. standard deviation) or associated estimates of uncertainty (e.g. confidence intervals) |
| ☐ | ☒ | For null hypothesis testing, the test statistic (e.g. *F*, *t*, *r*) with confidence intervals, effect sizes, degrees of freedom and *P* value noted<br>*Give P values as exact values whenever suitable.* |
| ☒ | ☐ | For Bayesian analysis, information on the choice of priors and Markov chain Monte Carlo settings |
| ☒ | ☐ | For hierarchical and complex designs, identification of the appropriate level for tests and full reporting of outcomes |
| ☒ | ☐ | Estimates of effect sizes (e.g. Cohen's *d*, Pearson's *r*), indicating how they were calculated |

*Our web collection on statistics for biologists contains articles on many of the points above.*

## Software and code

Policy information about availability of computer code

Data collection
BIAcore 3000 system (GE Healthcare)
Quantum-315 CCD detector at the MX2 beamline of the Australian Synchrotron
FACSymphony A3 analyser running FACSDiva software v9.1 (BD Biosciences)
Olympus BX60 epifluorescence microscope and Olympus DP-controller 3.1.1.267 (Olympus)
DMi8 Inverted Microscope (Leica Microsystems) with Leica Application Suite 2.7.3.9723
Leica SP5 and SP8 (Leica Microsystems) with Leica Application Suite v2.7.3.9723
Nikon AX R Ti2-E confocal microscope (Nikon) with NIS Elements v5.42.01
Zeiss LSM 980 with Airyscan Confocal Microscope (ZEISS) with Zen3.9
Amnis INSPIRE ImageStreamX (Cytek Biosciences)
Opera Phenix Plus High-Content Screening System (PerkinElmer)

Data analysis
Macromolecular Analysis:
CCP4 suite of programs  v7.0.023
Scrubber2.0 (BioLogic Software, Campbell, ACT, Australia)
Coot v0.8.7
Phaser v2.7.17
Pymol v2.1.1
Buster Release 20180515

Flow Cytometry:
FlowJo Version v10 (BD Biosciences)

Amnis imageStreamX:
Image Data Exploration and Analysis Software (IDEAS) Version 6.2.187.0
CellProfiler Version 4.2.6

Imaging:
Fiji ImageJ2, Version 2.14. or 1.54f
Imaris, Version 10.2.0

Data Analysis and Visualisation:
GraphPad Prism, Version 10.6

For manuscripts utilizing custom algorithms or software that are central to the research but not yet described in published literature, software must be made available to editors and reviewers. We strongly encourage code deposition in a community repository (e.g. GitHub). See the Nature Portfolio guidelines for submitting code & software for further information.

## Data

Policy information about availability of data

All manuscripts must include a data availability statement. This statement should provide the following information, where applicable:
- Accession codes, unique identifiers, or web links for publicly available datasets
- A description of any restrictions on data availability
- For clinical datasets or third party data, please ensure that the statement adheres to our policy

The data supporting the findings reported in this paper are included in the article and extended data. Source data for all relevant figures are included with the article. The structure factor file and associated atomic coordinates have been deposited in the Protein Data Bank, accession code 9EJW.
Data relating to the structure of murine CD44 (PDB code 2JCP) and m04 (PDB code 4PN6) are accessible from the Protein Data Bank (https://www.rcsb.org/)
The m11 sequence is available from GenBank (https://www.ncbi.nlm.nih.gov/) accession code: CAP08055.1
Requests for materials can be directed to M. Degli-Esposti.

## Research involving human participants, their data, or biological material

Policy information about studies with human participants or human data. See also policy information about sex, gender (identity/presentation), and sexual orientation and race, ethnicity and racism.

| | |
|---|---|
| Reporting on sex and gender | N/A |
| Reporting on race, ethnicity, or other socially relevant groupings | N/A |
| Population characteristics | N/A |
| Recruitment | N/A |
| Ethics oversight | N/A |

Note that full information on the approval of the study protocol must also be provided in the manuscript.

# Field-specific reporting

Please select the one below that is the best fit for your research. If you are not sure, read the appropriate sections before making your selection.

☒ Life sciences    ☐ Behavioural & social sciences    ☐ Ecological, evolutionary & environmental sciences

For a reference copy of the document with all sections, see nature.com/documents/nr-reporting-summary-flat.pdf

# Life sciences study design

All studies must disclose on these points even when the disclosure is negative.

| | |
|---|---|
| Sample size | Group sample sizes were chosen based on standards in the field and our previous studies where we used power analysis to estimate group sizes that would provide at least 80% power to detect statistically significant differences. |
| Data exclusions | Exclusion criteria were pre-established i.e. data were only excluded if outliers were identified using the ROUT test. |
| Replication | All experiments were replicated at least twice, except for (i) Extended data Fig. 2 where infections were conducted in triplicate within the |

| Replication | same experiment and (ii) Fig. 5d and Extended data Fig. 7f where 56-97 white pulp areas were examined from 3 mice/group within one experiment. There were no failures of replication. |
| --- | --- |
| Randomization | Age- and sex-matched mice were randomly allocated to groups.<br><br>For in vitro experiments randomization was not required given there are no relevant covariates i.e. all samples were treated at the same time and analysed using the same equipment. |
| Blinding | Microscopy image analysis was performed in a blinded fashion by multiple investigators.<br>All other analyses were strictly quantitative; hence no blinding was required. |

# Reporting for specific materials, systems and methods

We require information from authors about some types of materials, experimental systems and methods used in many studies. Here, indicate whether each material, system or method listed is relevant to your study. If you are not sure if a list item applies to your research, read the appropriate section before selecting a response.

## Materials & experimental systems

| n/a | Involved in the study |
| --- | --- |
| ☐ | ☒ Antibodies |
| ☐ | ☒ Eukaryotic cell lines |
| ☒ | ☐ Palaeontology and archaeology |
| ☐ | ☒ Animals and other organisms |
| ☒ | ☐ Clinical data |
| ☒ | ☐ Dual use research of concern |
| ☒ | ☐ Plants |

## Methods

| n/a | Involved in the study |
| --- | --- |
| ☒ | ☐ ChIP-seq |
| ☐ | ☒ Flow cytometry |
| ☒ | ☐ MRI-based neuroimaging |

## Antibodies

| Antibodies used | Refer to tables "List of antibodies used for flow cytometry" and "List of antibodies used for immunofluorescence" provided in the manuscript. |
| --- | --- |
| Validation | All antibodies except for the anti-vCD44BP (M-627, 7G5), anti-IE1 (clone 6/58/1) and anti-CD11c (clone N148) are from commercial sources and validation is provided on the manufacturer's website.<br><br>The specificity of anti-vCD44BP antibodies was verified by comparing cells infected with virus expressing vCD44BP and a mutant lacking vCD44BP, referred to as ΔvCD44BP, or cells transfected with vCD44BP.<br><br>The anti-IE1 (clone 5/58/1) antibody was originally described in Koszonowski et al, J. Virol., 1987. Specificity was verified by the absence of staining of uninfected cells.<br><br>Anti-CD11c (clone N418) was provided by Meredith O'Keefe, Monash University, Australia and verified as described in doi: 10.1084/jem.20021031.<br><br>Antibody dilutions used in our experiments are reported in the Methods section, however they are only relevant to the specific batch used, and therefore investigators should perform their own titrations prior to use. |

## Eukaryotic cell lines

Policy information about cell lines and Sex and Gender in Research

| Cell line source(s) | CV-1/EBNA<br>COS-7<br>EL4<br>IC-21<br>M2-10B4<br>SF9<br>Fibroblastic reticular cell line derived from murine C57BL/6 (FRC2) |
| --- | --- |
| Authentication | CV1/EBNA - sourced from ATCC (CCL-70)<br>COS-7 - sourced from ATCC (CRL-1651)<br>EL4 - sourced from ATCC (TIB-39)<br>IC-21- sourced from ATCC (TIB-186)<br>M2-10B4 sourced from ATCC (CRL-1972)<br>SF9 sourced from Thermo Fischer Scientific Cat# 11496-015<br><br>FRC2 was authenticated based on morphological assessment and flow cytometric analysis. |

| Mycoplasma contamination | The cell lines tested negative for mycoplasma. |
|---|---|
| Commonly misidentified lines (See ICLAC register) | No commonly misidentified cell lines were used. |

# Animals and other research organisms

Policy information about studies involving animals; ARRIVE guidelines recommended for reporting animal research, and Sex and Gender in Research

| Laboratory animals | BALB/c, C57BL/6J and C57BL/6.CD45.1 were from the Animal Resources Centre, Ozgene ARC (Perth, Western Australia) or the Walter and Eliza Hall Institute of Medical Research (Melbourne, Victoria).<br>B6.Cd44-/- and B6.BALB-TC1 (TC1) (H2b NK1.1+ Ly49H-) were bred at Perkins Bioresources Facility (Perth, Western Australia).<br>BALB/c.Ifng-/- mice and BALB/c.Prf1-/- were obtained from the Animal Services Facility at QIMR Berghofer Medical Research Institute (Queensland, Australia).<br>Age-matched adult female mice (8-12 weeks old) were used for all experiments.<br><br>Mice were housed in Techniplast IVC Greenline cages. Room temperature 18-24 degrees Celsius with humidity 40-70% and a 12-hour light cycle (7am-7pm). |
|---|---|
| Wild animals | The study did not use wild mice. |
| Reporting on sex | Refer to Methods (Mice). Female mice were used. |
| Field-collected samples | No field collected samples were used in this study. |
| Ethics oversight | All animal experimentation was performed with ethics approval from Monash University Ethics Committee (MARP2); Perkins Animal Ethics Committees (for the Lions Eye Institute); University of Western Australia Animal Ethics Committee (for the Lions Eye Institute) and in accordance with NHMRC Australia Code of Practice for the Care and Use of Animals for Scientific Purposes. |

Note that full information on the approval of the study protocol must also be provided in the manuscript.

# Flow Cytometry

## Plots

Confirm that:

☒ The axis labels state the marker and fluorochrome used (e.g. CD4-FITC).

☒ The axis scales are clearly visible. Include numbers along axes only for bottom left plot of group (a 'group' is an analysis of identical markers).

☒ All plots are contour plots with outliers or pseudocolor plots.

☒ A numerical value for number of cells or percentage (with statistics) is provided.

## Methodology

| Sample preparation | Refer to Methods: Isolation of leukocytes and fibroblastic reticular cells, cell staining and flow cytometric analysis |
|---|---|
| Instrument | BD FACSymphony A3 (Special Order Research Product), Serial Number: R66093723001 |
| Software | BD FACSDiva Software, Version 9.1 |
| Cell population abundance | Abundance of cell populations are indicated in the figures of the manuscript. |
| Gating strategy | The relevant gating strategies are presented in Extended data Figure 10. |

☒ Tick this box to confirm that a figure exemplifying the gating strategy is provided in the Supplementary Information.

