## [Peer Review File · Nature]

Fibroblastic reticular cells direct initiation of T cell responses via CD44

Corresponding Author: Professor Mariapia Degli-Esposti

Version 0:

Reviewer comments:

Referee #1

(Remarks to the Author)

Degli-Esposti and colleagues propose a new mechanisms whereby the mouse cytomegalovirus (MCMV) subverts adaptive immune recognition by encoding a CD44-binding protein, and claim to reveal an unrecognized biological function of CD44. The authors first identify the MCMV m11 gene as a viral encoded CD44-binding partner, referred to as vCD44BP. The elucidation of MCMV-encoded m11 is a CD44 binding protein that competes for binding to the HA-binding site is well substantiated. The authors next generate a mutant virus lacking vCD44BP to assess the role of vCD44BP. The authors observe a CD8+ T cell-dependent role for vCD44BP to dampen viral replication, which is associated with restricted DC migration into the splenic white pulp T cell zone.

In general, the manuscript is well written, data are well-presented, appropriate statistical analyses are performed and the relevant literature is well cited. The major claims of conceptual advance are: i) the discovery of a new viral evasion strategy and ii) revealing a previously unknown biological function of CD44 on fibroblasts. Nevertheless, there are substantial gaps in the mechanistic validation of the role by which vCD44BP interacts with fibroblast CD44 and how this relates to perturbed initiation of adaptive CD8+ T cell responses. The main weaknesses of the study are that the authors fail to demonstrate the binding of vCD44BP to fibroblast CD44, and to validate the functional role of fibroblast-expressed CD44 as a universal factor that promotes the initiation of CD8+ T cell responses. Hence the two main claims of the study are at the moment insufficiently substantiated.

vCD44BP binding to fibroblast CD44: The authors infer from the observed expression of CD44 on an FRC cell line and apparent morphological changes in CD44 distribution in cultured cells, that vCD44BP leads to a redistribution of CD44 and concomitant morphological changes in FRCs in vivo, which in turn likely affects their ability to bind to DCs through competitive binding to DC-produced HA. Several aspects of this proposed mechanism remain suggestive but have not been validated:

The co-expression of vCD44BP and CD44 expressed by FRCs has not been shown in vivo. While both infected and non-infected FRC cell lines express CD44, this does not necessitate that vCD44BP binds to CD44 on FRCs in vivo, although it is implied through Cd44^{-/-} bone marrow chimeras. Nevertheless, these latter experiments do not formally show that vCD44BP binds to CD44 on FRCs rather than other non-hematopoietic cells or incompletely sensitized, hematopoietic cells (indeed, the reciprocal chimeras have not been performed). Thus, the co-expression and interaction of vCD44BP and CD44 in FRCs must be validated in vivo.

Assuming vCD44BP binds surface CD44 in fibroblasts, it remains unclear if this is binding to CD44 is on the infected or neighbouring FRCs. As these early time points represent lytic MCMV infection, infected FRCs will likely be rapidly depleted, leading to perturbed antiviral immunity (PMIDs: 18425132, 27415420). In light of this, the relevance of the reported morphological changes in FRC cell lines remain unclear. Does vCD44BP binding to host FRC CD44 further disrupt the FRC network, accentuating the impairment in antiviral immune responses in a similar manner as described in LCMV infection. Or does vCD44BP binding to neighbouring CD44+ FRCs promote FRC proliferation, thereby extending the reservoir of FRCs to be infected. The authors have not shown how binding of vCD44BP to CD44+ FRCs alters FRC

morphology, proliferation and gene expression in vivo.

The proposed mechanistic link of FRC-intrinsic, CD44-mediated remodelling and DC interaction is not sufficiently validated. The authors suggest that vCD44BP interferes with FRC-DC interaction via blocking FRC CD44 binding to HA-produced by DCs. The authors have not shown: i) that DCs physically interact with infected CD44+ FRCs, ii) that by interfering with FRC-produced CD44, there is a morphological remodelling of the FRC network in vivo, and iii) a consequent impairment in the interaction with DCs.

The authors show a correlation between DC location in the T cell zone of the splenic white pulp and CD8+ T cell responses and viral titers. However, the authors have not functionally validated that the localisation of DCs in the splenic WP formally links the differentiation of CD8+ SLECs and viral clearance.

5. The second main claim of the paper is that the authors have uncovered a universal mechanism by which CD44 on FRCs promotes efficient CD8+ T cell activation (in the setting of the vCD44BP mutant MCMV). However to substantiate that the CD44-mediated interaction of FRCs and DCs is a broadly-relevant feature of FRCs, the authors should examine the effect on FRC morphology, proliferation, gene expression, DC interaction/migration and CD8+ T cell activation in response to other viral infections. To understand the mechanistic advantage from the point of the virus (ie. whether vCD44BP binding to CD44 on infected FRCs accelerates dissolution of the FRC network), it may be helpful to compare a virus that infects FRCs (such as LCMV) and a virus that does not readily target FRCs.

Minor comment: Sitnik K et al., recently described red pulp fibroblasts to be the main fibroblasts supportive of latent MCMV infection. The authors should substantiate the marker-based conclusion that lytic MCMV is mostly targeted to T cell zone FRCs.

Referee #2

(Remarks to the Author)

In this manuscript by Sng, Voigt and colleagues, the authors investigate the mechanisms by which viral infection impacts functional interactions between dendritic cells and T cells within secondary lymphoid organs. To this end, the authors focused on MCMV infection and specifically on the viral cell surface protein m11. They convincingly demonstrate that m11 binds to cellular CD44, and performed a series of experiments which convincingly implicate the CD44 binding domain of m11 in suppression of antiviral immune responses. This physical and functional relationship is novel and well-documented in the manuscript. The authors go on to propose a model wherein MCMV infection of fibroblastic reticular cells restricts trafficking of CD44 to filopodia in a manner dependent on m11 binding to CD44, such that infection with MCMV harboring a mutant m11 unable to bind CD44 results in redistribution of CD44, increased dendritic cell migration into the white pulp of the spleen, and an increased antiviral CD8 T cell response as a consequence of these DC-T cell interactions. They also claim that m11 restricts CD44 binding to its natural ligand, hyaluronic acid. While the results in the manuscript are generally consistent with this model, the data in the present manuscript fall short of a compelling demonstration of the functional consequences of m11 interaction with CD44. Additional mechanistic detail would help to clarify and strengthen the authors' central claims.

Specific comments:

- 1) The authors' model and present results suggest that CD44 binding to vCD44BP on FRCs limits dendritic cell migration but the cellular basis for this remains unclear. Does MCMV infection with and without intact vCD44BP differentially impact associations between CLEC-2 on DCs and podoplanin on FRCs? Does vCD44BP binding impact chemokine production by CD44-expressing FRCs?
- 2) Are CD44 and m11/vCD44BP interacting in cis or trans? Do these molecules interact on the surface of the same cells?
- 3) The cellular redistribution of CD44 on MCMV-infected FRCs by vCD44BP is somewhat difficult to appreciate from the results presented in Figure 4f and 4g. It would be helpful here to co-stain for CD44 and markers of filopodia and quantify filopodia-associated CD44 in the context of viral infection with and without intact vCD44BP.
- 4) The mutual exclusivity of CD44 interactions with HA and vCD44BP is interesting but the functional significance of this pattern would be strengthened with in vivo evidence. For example, does hyaluronidase treatment to degrade high molecular weight HA or genetic inhibition of hyaluronan synthase in FRCs reduce dendritic cell migration and/or exacerbate viral infection, and is increased binding of CD44 to vCD44BP evident (perhaps from morphological changes, or other FRC responses to this interaction) in the context of HA depletion upon MCMV infection?

Referee #3

(Remarks to the Author)

This study reports the finding that the MCMV m11 protein binds CD44, competing with HA for binding. A high-resolution structure establishes properties of the binding interface and confirms overlap with the CD44 HA binding site. Virus deficient in the m11 protein (renamed vCD44BP) replicates similarly in vitro and during the first 4 days in vivo but is then controlled more effectively. The control depends on CD8 T cells, and the mutant virus elicits a stronger virus specific short lived effector CD8 T cell response. Examining the basis for this effect, the Δ vCD44BP virus is shown to increase the efficiency of cDC1

mobilization into the splenic T zone and this is suggested to lead to improved CD8 T cell priming or expansion. MCMV is confirmed to infect T zone reticular cells (TRCs) in the spleen and microscopy of FRC type cells infected in vitro provides evidence that the Δ vCD44BP virus causes less rounding of the cells. Using BM chimeras, mice lacking CD44 in stromal cells are shown to support less growth of WT MCMV while the Δ vCD44BP virus grows with similar (lower) efficiency in both groups. Overall, the study is notable for its identification of a novel viral CD44 interacting protein and demonstration that it contributes to restraining the magnitude of the CD8 T cell response. The study provides some support for a model for how CD44 engagement may normally promote the CD8 response (and thus how CD44 antagonism may benefit viral growth) but there are weaknesses to this part of the study that need to be addressed.

Specific comments:

1. In Fig 3f-l, the authors claim that cDC1 movement into the T zone is augmented but cDC1 numbers in the spleen are not changed. While the data look quite suggestive, there is concern that the amount of background XCR1 staining in the T zone may have gone up in the inflamed spleens (the red staining seems more broad than might be expected solely from cDC1). Some additional support for the cDC1 differential localization model is needed. For example, intravascular pulse labeling can be used to preferentially label cDC that are blood exposed (in the red pulp) allowing quantitation of the fraction of cells (protected from antibody) in the white pulp. According to the model, there should be a strong increase in unlabeled cDC1 in the Δ vCD44BP infected mice. Data on cDC2 should be provided to help determine how specific the effect is for cDC1. Secondly, based on the authors model that the effect of vCD44BP is via CD44 on stromal cells, given the known role of CCL21 in cDC recruitment into the T zone, and the ability of WT MCMV to strongly downregulate splenic CCL21 expression (ref 27) it might be predicted that CCL21 expression is stronger in Δ vCD44BP versus WT MCMV infected spleens. This could be shown by QPCR and/or tissue staining for CCL21. Additionally, given the emphasis of the model on a cDC1 localization mechanism, literature should be cited that have established MCMV control depends on cDC1 (e.g., studies in *Batf3* KO mice).
2. In Fig 4j, k, there is increased CD8 expansion in control chimeras infected with Δ vCD44BP virus and this fits with the reduced PFU in panel i. The idea here is that without vCD44BP blocking CD44 function (HA binding) in stromal cells, there is induction of a more robust protective CD8 T cell response. As in, the presence of CD44 is suggested to be needed for the more robust CD8 response. With this model, it would be predicted that mice lacking CD44 in stromal cells would have a reduced CD8 response, irrespective of which virus infects them, and that is what is seen. But what is puzzling is why this is not associated with a high PFU/spleen in the stromal CD44-deficient mice matching that observed in TC1 \rightarrow WT mice. This discrepancy with the model needs explanation.
3. Given the broad expression of CD44 on hematopoietic cells (including cDCs and CD8 T cells) and established HA-dependent functions in these cells (e.g. ref 41) it seems unlikely that vCD44BP would only act via stromal cells. The authors may not have had CD44 $^{-/-}$ mice on the correct genetic background to perform CD44 $^{-/-}$ \rightarrow WT BM chimeras, but these would provide valuable complementary data if available.
4. HA binding to CD44 is blocked by anti-CD44 (Fig. 1g). Given the overlap in vCD44BP and HA binding sites, it should be determined whether vCD44BP binding is blocked by the anti-CD44 antibody. This affects interpretation of data in Fig. 4d, e, where CD44 is stained in cells that are expressing vCD44BP. If the antibody and vCD44BP binding sites are competitive, one might expect expression of vCD44BP to alter CD44 staining.
5. For vCD44BP to be altering CD44 distribution on cultured FRC (Fig. 4f) it is presumably interacting with the CD44 ectodomain in the cultures. Is the inference that this is a cis interaction between vCD44BP and CD44 on the same cell membrane? In Fig 4f and g, the morphology of cultured uninfected FRC needs to be shown (and it should be indicated if these are primary FRC or an FRC cell line). Was any attempt made to examine TRC morphology in vivo in the WT vs Δ vCD44BP infected mice?
6. The rationale for using TC1 strain bone marrow in Fig 4h is not adequately introduced. In particular, the influence of Ly49H (and how its presence affected the earlier experiments) is not described. The strains of mice used (e.g., Balb/c vs C57BL/6) and rationale for them needs to be stated more clearly throughout.
7. The author's model is that CD44 in stromal cells promotes cDC1 movement into the splenic T cell zone of mice infected with Δ vCD44BP MCMV. This leads to the prediction that in the TC1 \rightarrow Cd44 $^{-/-}$ BM chimeras, Δ vCD44BP MCMV should fail to induce increased cDC1 movement into the T zone. Was that seen?
8. In Fig 4l-n, the nature of the 'untreated' group needs clarification. It would seem that this should be the adjuvant (Montanide) treated group that was untreated with Fc fusion. This then makes it confusing why hTRAIL-Fc causes an increase in the response, and leaves it unclear if vCD44BP-Fc had any effect. If 'untreated' actually means no immunization, then a control 'adjuvant' only immunization group is needed (minimally to ensure that the hTRAIL-Fc is without effect).

Version 1:

Reviewer comments:

Referee #1

(Remarks to the Author)

The authors were asked to demonstrate the binding of vCD44BP to FRC-expressed CD44 in vivo and validate the function

of FRC-provided CD44 as universal factor that determines the efficiency of CD8+ T cell responses. The reviewer appreciates that in the revised manuscript the authors have made a considerable effort to use a combination of approaches to strengthen the evidence vCD44BP binds to FRC CD44. Nevertheless, a key concern remains that the authors cannot formally demonstrate that interference of CD44-mediated FRC-DC interaction by vCD44BP is how this viral protein delays CD8+ T cell responses. The reviewer does not dispute that in vitro infected FRC cell lines or primary splenic FRCs co-express CD44 and vCD44BP (newly added Figure 4g and 4h). Rather, the added in vitro approaches do not exclude that in vivo trans-binding of vCD44BP in infected FRCs acts uniquely by interrupting FRC-expressed CD44 binding to surface HA on DCs (as opposed to CD44 on other cells or secreted HA).

Data on the 3D cultures is compelling; however, the control experiments should also have been performed using pre-incubation of DCs with vCD44BP-Fc. Should this have induced a similar effect in reducing DC migration then this draws into question how vCD44BP can interact with neighboring cells in vivo, and the effect of therapeutic vCD44BP-Fc treatment in the vaccination and flu infection settings (Figure 5f-m). While the authors argue that reduced expression of vCD44BP in Cd44^{-/-} cells indicates a cis-interaction between FRC CD44 and vCD44BP, it remains unclear how CD44 co-expression is needed to stabilize a transmembrane protein (Figure 4i).

The authors emphasize the importance of white pulp (WP) FRCs – DC interaction. FRCs are relatively rare in the spleen, making the proportion of infected FRCs high relative to the more abundant red pulp (RP) fibroblasts (Ext. Fig. 4e). However, it is important to note that a similar number of WP and RP fibroblasts are infected by the virus (Extended Fig. 4d). Moreover, histological representation shows that the infected RP FRCs are highly abundant in proximity to the marginal zone, a site that supports DCs and T cells entering the splenic WP (Ext. Fig. 4f). Without the use of FRC-specific Cre-drivers, the authors cannot exclude the contribution of overall fibroblast vs FRC CD44 in driving antiviral T cell responses. The inability to pinpoint a function to WP FRCs is also relevant for the interpretation of Cd44^{-/-} BM chimera experiments.

Minor comments:

In MCMV-infected primary splenic FRCs (Figure 4h), there does not seem to be a redistribution of surface CD44 to FRC filopodia as seen of the FRC cell line (Figure 4e). It is also unclear why this antibody does not work for in situ imaging of splenic sections from MCMV-infected mice.

The authors have added images to demonstrate the proximity of viral IE1 and PDPN. In Figure 4e, a nuclear marker and 3D reconstruction or separate channels should be shown to support that these are FRCs and not closely interacting cells.

All representative FACS plots should be shown for the quantified data (ie. including tetramer staining).

Referee #2

(Remarks to the Author)

The authors have meaningfully and thoroughly responded to my comments from the original submission.

Referee #3

(Remarks to the Author)

The authors have made extensive revisions in response to the reviewer comments and the manuscript is improved. However, some specific concerns remain:

It remains unclear why FRC would be 'the main population able to bind vCD44BP-Fc after infection'. Do FRC really express more CD44 than various hematopoietic cells (e.g., effector CD8 cells)? This comparison is not shown. If CD44 levels are not higher on FRC but they bind more vCD44BP-Fc (than hematopoietic cells) is this because they have a form of CD44 that binds better?

Given that the authors have not been able to test the role of CD44 on hematopoietic cells in vivo (as they apparently have not intercrossed the CD44 KO and TC1 B6 mouse lines), they have not excluded an action of vCD44BP via CD44 on hematopoietic cells. Indeed, although the result in Fig. 4b is taken to argue that all the vCD44BP action depends on stromal CD44, there remains an effect of vCD44BP deletion in the TC1-> Cd44 KO chimeras that approaches that of the WT. Moreover, the authors show in other parts of the manuscript that vCD44BP can act 'in trans' when provided in a soluble format. How can it be excluded that even in transmembrane form (whether in FRC or other cells) it is not acting to engage CD44 on other cells? In the absence of data in full CD44 KO mice (and CD44 KO TC1 -> WT TC1 chimeras), this possibility must at least be discussed.

Line 231: 'This would account for the increased numbers of virus specific SLEC CD8 T cells observed after infection with the ΔvCD44BP virus.' In this sentence, 'would' should be changed to 'could'.

The data in figure 4j, k appear quite striking. The term 'branch length' should be clarified. Is this referring to the length of individual FRC membrane processes? Moreover, if the processes are longer, it is not intuitive (and not obvious from the image) how this would lead to such a marked increase in 'branches per FOV'. (Longer processes might be expected to correspond to fewer branch points).

The type of DC used for the in vitro assays is mentioned in the context of Versican staining. It should be clarified in the legend whether BMDC were used for all the in vitro assays (e.g., Fig. 4l, m).

The text (line 296) states 'DCs in the co-cultures expressed the CD44 ligand HA'. However ED Fig. 4r shows staining for Versican. Versican is not mentioned in the text and it is not explained why it is a sufficient marker to determine HA levels.

Version 3:

Reviewer comments:

Referee #1

(Remarks to the Author)

In the rebuttal, the authors have addressed one of the two remaining major revision points.

The added experiments using Crispr-Cas9 knockdown of Cd44 now formally demonstrate that the effect of vCD44BP on viral titers and CD8+ T cell responses is independent of CD44 on the hematopoietic compartment. However, the observation that the absence of CD44 on the hematopoietic compartment is dispensable for viral control, begs the question whether CD44-deficient DCs are still capable of binding to HA, as this is the primary molecule thought to mediate HA binding. The authors must show to what extent HA-binding on (activated) splenic DCs is dependent on CD44.

This relates to the outstanding point concerning the mechanism of how vCD44BP binds stromal cell CD44 to delay antiviral CD8+ T cell responses. The authors deduce that vCD44BP inhibits DC migration by blocking the interaction of FRC-CD44 with DC-HA based on several observations including:

- Crystal structures of purified vCD44BP and CD44 showing binding to the HA-cleft;
- In vitro DC migration assays showing a statistical difference in migration when FRCs are pretreated with vCD44BP-FC;
- In situations visualization of differences in DC accumulation in the T cell zone

Nevertheless, it remains that this mechanistic deduction is shown in parts (purified molecules, in vitro assays, and functional in vivo readouts), and experimental evidence demonstrating that attenuated DC-migration in vivo is mediated by interfering with FRC-CD44 to DC-HA is lacking.

A key feature in vivo that cannot be recapitulated in vitro is the lack of a reticular - FRC network in vitro and overall compositional differences in FRC-associated ECM. Particularly, in vivo, HA is integrated in the ECM. Thus, vCD44BP may interfere with FRC-ECM interactions, which may then induce a distinct transcriptional reprogramming of FRCs that indirectly influence DC recruitment (only CCL19 and CCL21 expression have been examined). These limitations must be considered in relation to the conclusions that can be drawn from in vitro DC migration studies (Figure 4 l, m).

Additionally, the connection of FRCs to a continuous reticular and cellular network means that there are no filipodia in vivo (presumably this is what is meant by apical surface in reference to Figure 4h). Therefore, it remains unclear whether a redistribution of CD44 on FRCs is at all relevant in vivo.

Similarly, it is unclear whether cell morphology changes in an FRC cell line have any relevance in vivo (Figure 4f). Similarly, whether there is any relevance of FRC morphological differences on the interaction with DCs remain speculative. Indeed, an alternative explanation to the more extended morphology may simply be the increased cellularity observed following infection with vCD44BP-deficient MCMV (Figure 3a, and 5 h, l). Indeed, this resembles the stretching of the FRC network observed in inflamed LNs upon infection, but is not necessarily a consequence of altered CD44-HA interaction between FRCs and DCs. As the relevance of this data remains speculative, Figures 4 f, g, j, k should be moved to the extended data).

With these considerations, and without a demonstration that HA in the ECM is irrelevant for DC migration in vitro or that FRC-DC interactions are truly disrupted in vivo, the conclusions must be reworded to imply that this is a possible but not formally demonstrated mechanism.

Minor comments:

i. The "apical distribution" of CD44 remains unclear. The term apical is usually reserved to describe the lumen-facing side of polarized cell types such as epithelial cells. If this refers to the filipodia in vitro, this should be stated, along with the limitation

that it remains unclear whether in vivo FRCs have similar structures in vivo.

ii. Representative images for Figure 5d must be provided.

iii. Figures 5i-j: The authors have now added the representative FACS plots for the quantification of OVA tetramer+ cells in the vaccination setting. In the OVA vaccination setting, it is clear that the antigen-specific effect size is minimal (maximally a difference of 0.1% of CD8+ T cells in PBS treatment compared with vCD44BP-Fc treatment). With populations of < 100 cells, a few cells can quickly account for such small differences between conditions. Rather the differences in CD8+ and antigen-specific T cells appear to be driven by overall cellularity and not proportional differences between treatment groups. This dataset should therefore be moved to the extended data.

iv. For data with a non-normal distribution, the geometric mean rather than mean should be shown. (ie. Fig. 5c).

v. p values should be provided for all datasets, especially instead of ns so that the reader can judge how close to the 0.05 cutoff the data are.

vi. The reviewer appreciates that the authors have added Figure 4e to show MCMV infection of PDPN+ cells. However, Fig. 4e must also be provided for MCMV infected splenic FRCs infected with vCD44BP-deficient MCMV, and the proportion of infected PDPN+ cells following WT or vCD44BP-deficient MCMV infection histologically quantified. Indeed, in Fig. 4d, the distribution of viral IE1 in relation to PDPN appears somewhat distinct for the two viruses. This is relevant, as the tropism of vCD44BP-deficient MCMV is not shown in vivo (presumably due to the lack of a vCD44BP-deficient, mCherry recombinant virus).

Referee #3

(Remarks to the Author)

In their further revised manuscript, the authors have included data that addresses several earlier concerns. I am surprised by the efficiency of CD44 KO in hematopoietic cells generated by CRISPR RNP treatment of HSC – without a selection step for cells that had received RNPs one might have expected a less complete KO (and thus a more bimodal staining pattern in ED Fig. 5h). Was this efficiency of KO observed in all recipients in both experiments? At what cell density were the purified progenitor cells incubated in StemSpan SFEM II medium? Did all mice survive reconstitution with 2×10^5 progenitor cells?

The data in Fig 3g suggest that MCMV normally limits cDC migration into the T zone by inhibiting CD44's ability to bind HA. That is, infection with WT MCMV is suggested to inhibit CD44 and reduce cDC migration and this is overcome in the Δ vCD44BP MCMV infected mice where CD44 function is not blocked. According to this model, the rescued cDC migration in Δ vCD44BP mice should be lost in CD44 KO mice. Is this the case?

Referee #4

(Remarks to the Author)

I restrict my comments to the characterization of the CD44 m11 interaction. The crystal structure is technical sound, at high resolution, well refined and supports the conclusions that m11 and HA compete for an overlapping binding site on CD44. The SPR data also appear solid, measuring the interaction at KD 14uM. It is rather puzzling that both m11 and HA have low affinity for CD44, the concentration of m11 would need to be very high in vivo in order to block HA binding which is on cells and has an entropic advantage. This merits some comment in the text, the numbers would seem to work against m11. Do the authors think the inhibition is partial or incomplete ?

Version 4:

Reviewer comments:

Referee #1

(Remarks to the Author)

The reviewer appreciates the added explanation as to why the deletion of HA synthesis in DCs is currently not possible, and appreciates the text added to the discussion adding further insights into the mechanisms of how stromal cells alter DC migration through CD44 - HA interactions.

The reviewer acknowledges that the authors changed the manuscript title from Fibroblastic reticular cells to fibroblastic stromal cells to account for infected fibroblasts within the red pulp. However, as the conceptualized FRC - DC - T cell interaction centers in the white pulp, the reviewer would recommend using the FRC nomenclature for the manuscript title.

The reviewer appreciates that the authors have added representative images and quantification of MCMV infected PDPN+ cells in Figure 4. It remains very difficult to visualize that the IE1+ cells in Figure 4e are PDPN+ FRCs - FRCs tend to have an elongated cell body, and when a nucleus is captured in the z-stack, a PDPN signal should surround the nucleus. While

the flow cytometric quantification suggests that hematopoietic cells are rarely infected with MCMV, in both MCMV infected mice and in mice infected with the vCD44BP-deficient MCMV, the PDPN signal appears adjacent to an MCMV infected cell rather than outlining infected cells. It is strongly recommended that the authors provide a better visualization of MCMV infected FRC in the splenic white pulp rather than CD45+ hematopoietic cells. This may be as simple as showing the single channel PDPN if the IEL1 signal is simply too strong relative to the PDPN for improved visualization.

Referee #3

(Remarks to the Author)

The authors have adequately addressed my principle concerns.

I remain surprised by the efficiency of the RNP-induced CD44 KO in HSC. Moreover, the amounts of Cas9 protein and sgRNA complex are less than in published studies (e.g., Khoo et al., JOVE 2023) - the details of the protocol should be checked to ensure that others can reproduce these findings (that are impressive if reproducible).

Referee #4

(Remarks to the Author)

The authors response to my question about relative affinities of M11 versus HA is reasonable, although speculative. The possibility of cis CD44/M11 interaction does increase the effective concentration of m11 enormously and could explain why it effectively inhibits the HA interaction.

Authors should cite Borowska et al. Science Immunology, 2024 as a highly related case where a virus has targeted a cell surface phosphatase on immune cells, although using a different mechanism.

We appreciate the Referees' enthusiasm about the conceptual advancements provided by our study and their recognition of the quality of the supporting data. We are grateful for their helpful comments, all of which have been addressed in detail as outlined below. The revised manuscript includes extensive new data that reinforce the role of fibroblast CD44 in promoting the initiation of adaptive T cell responses and substantiate the mechanisms involved.

Referee 1

Degli-Esposti and colleagues propose a new mechanism whereby the mouse cytomegalovirus (MCMV) subverts adaptive immune recognition by encoding a CD44-binding protein, and claim to reveal an unrecognized biological function of CD44. The authors first identify the MCMV m11 gene as a viral encoded CD44-binding partner, referred to as vCD44BP. The elucidation of MCMV-encoded m11 is a CD44 binding protein that competes for binding to the HA-binding site is well substantiated. The authors next generate a mutant virus lacking vCD44BP to assess the role of vCD44BP. The authors observe a CD8⁺ T cell-dependent role for vCD44BP to dampen viral replication, which is associated with restricted DC migration into the splenic white pulp T cell zone. In general, the manuscript is well written, data are well-presented, appropriate statistical analyses are performed and the relevant literature is well cited. The major claims of conceptual advance are: i) the discovery of a new viral evasion strategy and ii) revealing a previously unknown biological function of CD44 on fibroblasts. Nevertheless, there are substantial gaps in the mechanistic validation of the role by which vCD44BP interacts with fibroblast CD44 and how this relates to perturbed initiation of adaptive CD8⁺ T cell responses.

1. The main weaknesses of the study are that the authors fail to demonstrate the binding of vCD44BP to fibroblast CD44, and to validate the functional role of fibroblast expressed CD44 as a universal factor that promotes the initiation of CD8⁺ T cell responses. Hence the two main claims of the study are at the moment insufficiently substantiated.

The Referee asks for more evidence that vCD44BP binds to fibroblast CD44 and that impairing fibroblasts' CD44 function affects the initiation of CD8⁺ T cell responses. These points have been addressed by showing that:

- (i) vCD44BP directly binds to CD44 on FRCs (**Rebuttal Fig. 1**) and the binding shows higher preference for FRCs than endothelial cells
- (ii) vCD44BP and CD44 colocalise in infected FRCs, including infected primary FRCs (**New Fig. 4g and h; and New Extended data Fig. 4l and m**)
- (iii) CD44 stabilises vCD44BP expression in primary infected FRCs (**New Fig. 4i**)
- (iv) lack of stromal CD44 results in impaired DC localization (**New Fig. 5e**) and altered CD8⁺ T cell responses (**Fig. 5c and d**) in the chimera model
- (v) vCD44BP-Fc impairs antigen-specific CD8⁺ T cell responses generated in response to vaccination (**New Fig. 5f-i**) and influenza challenge (**New Fig. 5j-m**).

Please also see the response to comment 2 for further details.

Rebuttal Fig. 1. FRCs are the main population binding vCD44BP-Fc.

a. Concatenated pseudocolor plots of vCD44BP-Fc binding to splenic FRC (left) or CD31⁺ endothelial cells (EC) (right).

b. Proportion of FRCs and EC that bind vCD44BP-Fc.

2: vCD44BP binding to fibroblast CD44: The authors infer from the observed expression of CD44 on an FRC cell line and apparent morphological changes in CD44 distribution in cultured cells, that

vCD44BP leads to a redistribution of CD44 and concomitant morphological changes in FRCs in vivo, which in turn likely affects their ability to bind to DCs through competitive binding to DC-produced HA. Several aspects of this proposed mechanism remain suggestive but have not been validated.

The co-expression of vCD44BP and CD44 expressed by FRCs has not been shown in vivo. While both infected and non-infected FRC cell lines express CD44, this does not necessitate that vCD44BP binds to CD44 on FRCs in vivo, although it is implied through Cd44-/- bone marrow chimeras. Nevertheless, these latter experiments do not formally show that vCD44BP binds to CD44 on FRCs rather than other non-hematopoietic cells or incompletely sensitized, hematopoietic cells (indeed, the reciprocal chimeras have not been performed). Thus, the co-expression of vCD44BP and CD44 expressed by FRCs must be validated in vivo.

Co-expression of vCD44BP and CD44 expressed by FRCs: The co-expression of vCD44BP and CD44 has been confirmed including in infected primary splenic FRCs (**New Fig. 4g and h; and New Extended data Fig. 4I**). Furthermore, we show that expression of vCD44BP in primary splenic FRCs requires CD44 (**New Fig 4i**), supporting the critical importance of this cis interaction in infected FRCs.

Staining of vCD44BP *in vivo* has not been possible. We generated monoclonal (x3) and polyclonal (x1) antibodies against vCD44BP but could not detect vCD44BP in spleens of infected mice by immunofluorescence microscopy - this is not unusual for highly glycosylated proteins such as vCD44BP. We also attempted to examine vCD44BP by flow cytometry on FRCs isolated from the spleen of infected mice; however, the enzymes required for the preparation of the stromal cell fractions cleave vCD44BP – this was demonstrated by examining primary FRC infected *in vitro* and treated with relevant enzymes individually.

The Referee is particularly interested in evidence that vCD44BP causes significant morphological changes in FRC *in vivo*. We have identified a differential effect of WT and Δ vCD44BP viruses on the FRC network *in vivo* and this is discussed in detail below (response to comment 3). We have also demonstrated that FRCs in the white pulp are infected (**New Fig. 4d; and Extended data Fig. 4f**). In fact, WP FRCs are preferentially targeted by MCMV *in vivo* (**New Extended data Fig. 4e**), consistent with a critical role for these cells in anti-viral immunity and the need for CMV to target their functions, including via the expression of vCD44BP.

Show that vCD44BP binds to CD44 on FRCs rather than other non-hematopoietic cells or incompletely sensitized, hematopoietic cells: The role of FRCs rather than other stromal cells is supported by the finding that FRCs are the main population able to bind vCD44BP-Fc after infection (**Rebuttal Fig. 1**) and confirmed by data showing that the endothelial cells that are infected by MCMV *in vivo* do not express CD44 (**Rebuttal Fig. 2**).

Rebuttal Fig 2. A minor fraction of CD44-negative endothelial cells is infected by CMV *in vivo*. **a.** Percentage of CMV infected (mCherry⁺) endothelial cells. **b.** Concatenated pseudocolor plots of CD31⁺ endothelial cells from naïve (left) or MCMV:mCherry infected (right) BALB/c mice at 24hr post-infection are shown. Endothelial cells that are infected (mCherry⁺) do not express CD44.

Therefore, in contrast to FRCs, most endothelial cells lack CD44 expression and consequently do not bind vCD44BP (see **Rebuttal Fig. 1** above). Notably, CMV infects endothelial cells that do not express CD44 (**Rebuttal Fig. 2**).

The possibility that vCD44BP binds to CD44 on incompletely sensitized hematopoietic cells in the chimera system has been ruled out by the data below which show greater than 95% chimerism in the hematopoietic compartment (**Rebuttal Fig. 3**). These data confirm that the results obtained in the chimera are due to the interaction of vCD44BP with CD44 on non-hematopoietic stromal cells.

Rebuttal Fig 3. Donor ($CD45.2^+$) cells constitute the majority (95%) of live cells in chimeric mice. **a.** Concatenated pseudocolor plot of live cells from B6.TC1 \rightarrow B6.CD45.1 (left) and frequency of CD45.1 (recipient) or CD45.2 (donor) live cells (right). The frequency of macrophages, the most radioresistant cells, is also shown (far right graph).

Altogether these data establish that *in vivo* vCD44BP affects immune responses due to impacts on FRCs. This tenet is further supported by the fact that, after WT MCMV infection, altered DC distribution is observed in WT chimera but not in stromal *Cd44*^{-/-} chimeras (**New Fig. 5e** –see response to R3, comments #5 and 6 for further details, including reason for the absence of reciprocal chimeras), and that this results in differences in antiviral CD8 T cell responses (**Fig. 5c and d**).

3: Assuming vCD44BP binds surface CD44 in fibroblasts, it remains unclear if this is binding to CD44 on the infected or neighbouring FRCs. As these early time points represent lytic MCMV infection, infected FRCs will likely be rapidly depleted, leading to perturbed antiviral immunity (PMIDs: 18425132, 27415420). In light of this, the relevance of the reported morphological changes in FRC cell lines remain unclear. Does vCD44BP binding to host FRC CD44 further disrupt the FRC network, accentuating the impairment in antiviral immune responses in a similar manner as described in LCMV infection. Or does vCD44BP binding to neighbouring CD44⁺ FRCs promote FRC proliferation, thereby extending the reservoir of FRCs to be infected. The authors have not shown how binding of vCD44BP to CD44⁺ FRCs alters FRC morphology, proliferation and gene expression *in vivo*.

Does vCD44BP bind CD44 on the infected or neighbouring FRCs and are infected FRCs rapidly depleted? As discussed above, vCD44BP and CD44 are co-expressed and co-localise in the same cell in infected FRCs, including primary splenic FRCs (**New Fig. 4g and h**) and expression of vCD44BP in infected FRCs requires CD44 (**New Fig. 4i**).

We have shown that FRCs are a primary target of MCMV infection *in vivo* (24hrs) (**New Fig. 4b-c; and Extended data Fig. 4c-e**) and now provide additional evidence, including sustained T cell zone reticular cells (TRC) infection (day 2), indicating that infected FRCs are not rapidly depleted (**New Fig. 4d; and Extended data Fig. 4f**).

We also show that at day 2 pi FRC numbers in mice infected with either WT MCMV or Δ vCD44BP viruses are equivalent, and similar to those observed in naive mice (**New Extended data Fig. 4n**). Thus, lytic infection of FRCs cannot account for the differences induced by vCD44BP.

Extended data Fig 4n. Number of stromal cells remain unchanged early in infection (day 2) and are equivalent in mice infected with WT MCMV (grey bar) and Δ vCD44BP (blue bar) viruses. Naïve = white bar.

FRC numbers decrease later in infection but there is no difference between WT MCMV and Δ vCD44BP viruses. Notably, vCD44BP leads to changes in the FRC network as discussed below.

Does vCD44BP promote FRC proliferation? MCMV infection results in reduced FRC proliferation, however this is not affected by vCD44BP (**New Extended data Fig. 4o**).

Extended data Fig 4o. *FRC proliferation is decreased after MCMV infection independently of vCD44BP.* Concatenated pseudocolor plots of Ki67 expression on FRCs (Lin-Ter119-CD45-CD31- PDPN+) from naïve or infected mice (d2 pi, left). Frequency of Ki67⁺ stromal cells (right).

Does binding of vCD44BP to CD44⁺ FRCs alter the FRC network? Examination of the splenic FRC network after infection with WT MCMV and $\Delta vCD44BP$ revealed a number of differences. Firstly, high-resolution imaging highlighted that FRCs in the splenic white pulp are readily infected by MCMV (**New Fig. 4d**).

Skeleton analysis revealed a differentially remodeled FRC network in spleens from mice infected with $\Delta vCD44BP$ (**New Fig. 4j**), characterized by increased branch lengths corresponding to an increased number of elongated and more complex PDPN⁺ fibers (**New Fig. 4j and k**), see part of Fig. 4 below. This FRC network remodeling is critical to generate effective immune responses [refs 1-5].

New Fig. 4k. *vCD44BP affects remodelling of the FRC network.* The podoplanin⁺ FRC network in the spleens of WT MCMV and $\Delta vCD44BP$ infected mice was examined by skeleton analysis. Length of branches in the white pulp at day 2 pi are plotted. The data are representative of 450 (WT MCMV and $\Delta vCD44BP$) and 360 (naïve) podoplanin⁺ skeleton branches derived from the spleen of 2 naïve mice and 3 infected mice per group.

4: *The proposed mechanistic link of FRC-intrinsic, CD44-mediated remodelling and DC interaction is not sufficiently validated. The authors suggest that vCD44BP interferes with FRC-DC interaction via blocking FRC CD44 binding to HA-produced by DCs. The authors have not shown: i) that DCs physically interact with infected CD44⁺ FRCs, ii) that by interfering with FRC-produced CD44, there is a morphological remodelling of the FRC network in vivo, and iii) a consequent impairment in the interaction with DCs.*

Do DCs physically interact with infected CD44⁺ FRCs: Because of the widespread distribution of CD44 and its highly glycosylated structure, imaging for this protein *in vivo* is technically very difficult. However, we show that, as predicted from our model, DCs interact closely with infected cells in the white pulp (**New Fig. 3i**). Notably, we have shown that the infected cells in the white pulp are principally stromal TRCs (**New Fig. 4c and d; and Extended data Fig. 4c-e**); 4d shown below.

New Fig. 3i. *Close interactions between MCMV infected cells, DCs and CD8 T cells in the white pulp.* Image of cells in the splenic white pulp of MCMV infected mice at day 2 pi. Green = infected cells identified by IE1 staining, blue = CD11c DCs, pink = CD8 T cells. White circle shows example of interactions.

New Fig. 4d. *MCMV infects FRCs in the white pulp.* Image of podoplanin+ cells in the splenic white pulp of MCMV infected mice at day 2 pi. Green = infected cells identified by IE1 staining, pink = T cell zone FRCs identified by podoplanin staining.

By interfering with FRC-produced CD44, is there a morphological remodelling of the FRC network in vivo and consequent impairment in the interaction with DCs: As discussed above, we have added new data that demonstrate that the FRC network is differentially remodelled in the presence of vCD44BP *in vivo* (**New Fig. 4j and k**).

We previously showed that DC trafficking to the white pulp is impaired by vCD44BP (**Fig. 3g and h**) and now provide evidence in the chimera system that this differential DC trafficking depends on expression of CD44 on stromal cells (**New Fig. 5e** – please see Response to Referee 3 Comment 5 for further details).

We have also added new data that directly show that CD44 expressed by FRCs is integral to the trafficking of DCs on FRCs. Time-lapse imaging demonstrated that the movement of DCs on FRCs in 3D cultures is significantly reduced (i) when FRCs are pre-treated with vCD44BP-Fc prior to co-culture with DCs (**New Fig. 4l** – distance graph shown below), and (ii) when FRCs are CD44-deficient, a situation that mimics the unavailability of CD44-HA interactions when vCD44BP is expressed in infected FRCs (**New Fig. 4m** – distance graph shown below).

New Fig. 4l and m (subpanels). *The movement of DCs is affected by FRC CD44.* **l.** Primary splenic FRCs pre-treated with vCD44BP-Fc (40 minutes at 4°C and washed). **m.** WT or CD44-deficient primary splenic FRCs. In both assays FRCs were co-cultured with DCs in 3D gels and the distance travelled by DCs interacting with FRC was examined by time-lapse imaging 24 hours later. Data are pooled from 2 independent experiments.

Since interaction between CLEC-2 on DCs and podoplanin on FRCs can affect DC migration, we examined podoplanin expression on the CD44-deficient FRCs used in our assay and found it was not affected (**Extended data Fig. 4q**). Podoplanin partners with CD44 to execute its functions [refs 6,7], with the interactions mediated by the transmembrane and cytosolic regions [ref 8]. Since expression of podoplanin on the CD44-deficient FRCs used in our assay was not affected (**Extended data Fig. 4q**) and the CD44–vCD44BP-Fc interaction involves the extracellular domain of CD44 (where the HA binding cleft is located), we conclude that CD44 can direct DC migration independently of podoplanin–CLEC-2 interactions. Consistent with DC migration being directed by CD44, the DC in the co-cultures expressed the CD44 ligand HA (**Extended data Fig. 4r**).

In summary, the extensive new data added to the manuscript provide evidence that vCD44BP affects the FRC network via binding to CD44, a molecule which in FRCs we show to be important for DC migration.

5: *The authors show a correlation between DC location in the T cell zone of the splenic white pulp and CD8+ T cell responses and viral titers. However, the authors have not functionally validated that the localisation of DCs in the splenic WP formally links the differentiation of CD8+ SLECs and viral clearance.*

The crucial role of DCs in initiating anti-MCMV CD8 T cell responses is well documented [refs 9-11], as is the effect of the CD8 T cell response on viral titers [refs 12-15]. We have shown that FRCs within the white pulp are infected at day 2 pi (**New Fig. 4d**) and found that infected cells in the white pulp are in close contact with DCs, which in turn closely interact with CD8 T cells (**New Fig. 3i**).

To further address the question raised above would require experimental systems that enable time-lapse analysis of DCs and antigen specific CD8 T cells by two-photon laser scanning microscopy in the spleen, the site of priming in the MCMV infection model. Thus, DC reporter and transgenic CD8 T cell mice would be required; neither of these tools are available on a BALB/c background where the generation of effective CD8 T cell responses is key for MCMV control. It is also worth noting that in our model of CMV infection, vCD44BP acts to slow, rather than eliminate the generation of early anti-viral CD8 T cell responses; thus, the magnitude of the difference in WT versus Δ vCD44BP infections *in vivo* may be difficult to capture. For these reasons we trust that the Referee agrees that additional studies are not possible at present.

6: *The second main claim of the paper is that the authors have uncovered a universal mechanism by which CD44 on FRCs promotes efficient CD8+ T cell activation (in the setting of the vCD44BP mutant MCMV). However to substantiate that the CD44-mediated interaction of FRCs and DCs is a broadly-relevant feature of FRCs, the authors should examine the effect on FRC morphology, proliferation, gene expression, DC interaction/migration and CD8+ T cell activation in response to other viral infections.*

In our original manuscript we showed that vCD44BP could alter adjuvant-driven lymph node expansion, a key step in the generation of adaptive immune responses, independently of infection (now in **Extended data Fig. 5e-g**). We have now broadened our analyses and show that treatment with vCD44BP-Fc specifically impairs an antigen-specific CD8 T cell response in a vaccination model (**New Fig. 5f – i**; h and i shown below).

New Fig. 5h and i. *vCD44BP-Fc impairs antigen-specific CD8 T cell responses in a vaccination model.* The magnitude of an antigen-specific response was assessed by examining CD8 T cell numbers after immunisation with antigen (OVA) and adjuvant (Montanide) in the draining lymph nodes. Data are combined from 3 independent experiments.

h. number of CD8+ T cells
i. number of OVA-specific CD8+ T cells

In addition, we have also examined the impact of treatment with vCD44BP-Fc in a model of influenza infection and observed a similar impairment in flu-specific CD8 T cell responses (**New Fig. 5j – m**), also shown below.

New Fig. 5j - m. *vCD44BP-Fc impairs flu-specific CD8 T cell responses.* The magnitude of a flu CD8 T response was assessed in mediastinal lymph nodes. Data are combined from 3 independent

experiments. **j.** model and interventions; **k.** total cell numbers in mediastinal lymph nodes; **l.** number of CD8+ T cells; **m.** percentage and number of influenza NP-specific CD8+ T cells.

In combination, these data provide evidence that altering CD44 functionality affects the generation of antigen-specific CD8 T cell responses.

***Minor comment:** Sitnik K et al., recently described red pulp fibroblasts to be the main fibroblasts supportive of latent MCMV infection. The authors should substantiate the marker-based conclusion that lytic MCMV is mostly targeted to T cell zone FRCs. If MCMV causes lytic infection of TRCs, then there should be decline in number at d2 whereas other populations (RPRCs?) may not be affected. This assumes that there is no proliferation of FRCs in that time.*

While we detect MCMV-infected red pulp fibroblasts (**Extended data Fig. 4a-d**), our data clearly establish that FRCs in the T cell zone are a principal target of early MCMV infection (**New Fig. 4d and Extended data Fig. 4c-e**). Furthermore, as shown above (**Extended data Fig. 4n and o**), there was no loss of FRCs at day 2 after infection, and no difference in FRC numbers or proliferation were observed in mice infected with WT MCMV or Δ vCD44BP viruses. It is worth noting that Sitnik et al examined lytic infection by measuring the expression of the immediate-early (ie1/ie3), early (M38) or late (M48/SCP) viral transcripts and did not assess viability of the infected stromal subsets *in vivo*. The Sitnik study utilized C57BL/6 mice, a system where virally infected cells are rapidly eliminated by NK cells; thus, it would be very difficult to differentiate lytic infection resulting in apoptosis of infected cells from immune-mediated elimination *in vivo*.

Importantly, the Sitnik study shows that MCMV persists long term in splenic fibroblasts, including those in the white pulp, indicating that not all infected cells will necessarily die.

In this manuscript by Sng, Voigt and colleagues, the authors investigate the mechanisms by which viral infection impacts functional interactions between dendritic cells and T cells within secondary lymphoid organs. To this end, the authors focused on MCMV infection and specifically on the viral cell surface protein m11. They convincingly demonstrate that m11 binds to cellular CD44, and performed a series of experiments which convincingly implicate the CD44 binding domain of m11 in suppression of antiviral immune responses. This physical and functional relationship is novel and well-documented in the manuscript. The authors go on to propose a model wherein MCMV infection of fibroblastic reticular cells restricts trafficking of CD44 to filopodia in a manner dependent on m11 binding to CD44, such that infection with MCMV harboring a mutant m11 unable to bind CD44 results in redistribution of CD44, increased dendritic cell migration into the white pulp of the spleen, and an increased antiviral CD8 T cell response as a consequence of these DC-T cell interactions. They also claim that m11 restricts CD44 binding to its natural ligand, hyaluronic acid. While the results in the manuscript are generally consistent with this model, the data in the present manuscript fall short of a compelling demonstration of the functional consequences of m11 interaction with CD44. Additional mechanistic detail would help to clarify and strengthen the authors' central claims.

1: The authors' model and results suggest that CD44 binding to vCD44BP on FRCs limits dendritic cell migration but the cellular basis for this remains unclear. Does MCMV infection with and without intact vCD44BP differentially impact associations between CLEC-2 on DCs and podoplanin on FRCs? Does vCD44BP binding impact chemokine production by CD44-expressing FRCs?

Does MCMV infection with and without intact vCD44BP differentially impact associations between CLEC-2 on DCs and podoplanin on FRCs? The Referee raises a point that we believe highlights the significance of our study. In the context of FRC-DC interactions, the published literature has primarily concentrated on the role of podoplanin–CLEC-2 interactions. Although evidence suggests that CD44 may also play a role, its function has not been investigated and consequently has been considered secondary to that of podoplanin. As discussed above (response to Reviewer 1, comment 4), to validate the role of CD44 as a key mechanism in FRC-DC interactions necessary for DC migration, we investigated DC migration in co-cultures with primary splenic FRCs lacking CD44. Our data demonstrate that DC migration on CD44^{-/-} FRCs is impaired (**New Fig. 4m**). Since expression of podoplanin on the CD44-deficient FRCs used in our assay was not affected (**Extended data Fig. 4q**) and the CD44–vCD44BP-Fc interaction involves the extracellular domain of CD44 (where the HA binding cleft is located) rather than the transmembrane and cytosolic regions that mediate podoplanin–CD44 interaction, we conclude that CD44 can direct DC migration independently of podoplanin–CLEC-2 interactions. Consistent with DC migration being directed by CD44, the DCs in the co-cultures expressed the CD44 ligand HA (**Extended data Fig. 4r**).

To validate the impact of vCD44BP binding to CD44 on FRCs as necessary to limit DC migration, we pre-treated primary splenic FRCs with vCD44BP-Fc and, after removing unbound Fc, co-cultured them with DCs and measured DC migration. As shown above (response to R1, # 4) the movement of DCs on FRCs is significantly reduced when FRCs are pre-treated with vCD44BP-Fc (**New Fig. 4l**) – (n.b., we confirmed that vCD44BP-Fc remains bound to FRCs during the co-culture period and that DCs in the co-cultures express HA, the CD44 ligand, **New Extended data Fig. 4r**). Thus, when the HA binding site of CD44 on FRC is occupied by vCD44BP, DC migration on FRCs is compromised.

During MCMV infection *in vivo*, migration of DCs into the T cell zone is impaired but not eliminated in the presence of vCD44BP indicating that other interactions (e.g., CLEC-2–PDPN) and chemotactic gradients are not specifically altered by vCD44BP – see below.

Does vCD44BP binding impact chemokine production by CD44-expressing FRCs? There are no differences in the expression of CCL19 and CCL21 (chemokines primarily produced by FRCs – ref 16) between WT MCMV and Δ vCD44BP infections. Relative to naïve levels, CCL19 showed a slight increase following infection, while expression of CCL21 decreased (**New Extended Data Fig. 4p**). These data have been added to the revised manuscript.

New Extended Data Fig. 4p. Chemokine expression is equivalent in WT and $\Delta vCD44BP$ infections. CCL19 and CCL21 mRNA levels in spleen homogenates (day 2 pi) are shown. Data are combined from 3 experiments.

2: Are CD44 and m11/vCD44BP interacting in cis or trans? Do these molecules interact on the surface of the same cells?

vCD44BP and CD44 are co-expressed and co-localise in infected FRCs, including infected primary FRCs (**New Fig. 4g and h**; and **Extended data Fig. 4I**), and cell surface expression of vCD44BP is reduced in the absence of CD44 (**New Fig. 4i**); these data support the critical importance of a CD44 - vCD44BP cis interaction in infected FRCs.

3: The cellular redistribution of CD44 on MCMV-infected FRCs by vCD44BP is somewhat difficult to appreciate from the results presented in Figure 4f and 4g. It would be helpful here to co-stain for CD44 and markers of filopodia and quantify filopodia-associated CD44 in the context of viral infection with and without intact vCD44BP.

We have included better images to show redistribution of CD44 on MCMV-infected FRCs as suggested. We have found it difficult to co-stain for CD44 and filopodia as the permeabilization required to stain for actin filaments affects the distribution of CD44.

4: The mutual exclusivity of CD44 interactions with HA and vCD44BP is interesting but the functional significance of this pattern would be strengthened with *in vivo* evidence. For example, does hyaluronidase treatment to degrade high molecular weight HA or genetic inhibition of hyaluronan synthase in FRCs reduce dendritic cell migration and/or exacerbate viral injection, and is increased binding of CD44 to vCD44BP evident (perhaps from morphological changes, or other FRC responses to this interaction) in the context of HA depletion upon MCMV infection?

The *in vivo* experiments suggested by the Referee would be extremely challenging to conduct without a genetically inducible model of hyaluronan synthase, something that will necessitate the *de novo* generation of the required mouse strain on a BALB/c background.

As HA forms an important glycocalyx on almost all cell types, hyaluronidase treatment to degrade high molecular weight HA would make results very difficult to interpret especially in the presence of subsequent viral infection. Furthermore, the literature [ref 17] reports that "administration of HA tetrasaccharides, before or simultaneously with antigen application, recapitulated phenotypes observed in *HYAL1*-expressing animals, suggesting that the generation of small HA fragments, rather than the loss of large HA molecules, promotes DC migration and subsequent modification of allergic responses."

To address the relevance of CD44-HA-vCD44BP we have added new data that show that DC migration on FRCs is compromised *in vitro* when the HA binding site on CD44 on FRC is occupied by vCD44BP (**New Fig. 4I**). The differences in DC migration to the WP observed in the CD44KO BM chimeras (**New Fig. 5e** – see Response to Referee 3 Comment 5 for the data) provide further *in vivo* evidence.

This study reports the finding that the MCMV m11 protein binds CD44, competing with HA for binding. A high-resolution structure establishes properties of the binding interface and confirms overlap with the CD44 HA binding site. Virus deficient in the m11 protein (renamed vCD44BP) replicates similarly in vitro and during the first 4 days in vivo but is then controlled more effectively. The control depends on CD8 T cells, and the mutant virus elicits a stronger virus specific short lived effector CD8 T cell response. Examining the basis for this effect, the Δ vCD44BP virus is shown to increase the efficiency of cDC1 mobilization into the splenic T zone and this is suggested to lead to improved CD8 T cell priming or expansion. MCMV is confirmed to infect T zone reticular cells (TRCs) in the spleen and microscopy of FRC type cells infected in vitro provides evidence that the Δ vCD44BP virus causes less rounding of the cells. Using BM chimeras, mice lacking CD44 in stromal cells are shown to support less growth of WT MCMV while the Δ vCD44BP virus grows with similar (lower) efficiency in both groups. Overall, the study is notable for its identification of a novel viral CD44 interacting protein and demonstration that it contributes to restraining the magnitude of the CD8 T cell response. The study provides some support for a model for how CD44 engagement may normally promote the CD8 response (and thus how CD44 antagonism may benefit viral growth) but there are weaknesses to this part of the study that need to be addressed.

*1: Some additional support for the cDC1 differential localization model is needed. For example, intravascular pulse labeling can be used to preferentially label cDC that are blood exposed (in the red pulp) allowing quantitation of the fraction of cells (protected from antibody) in the white pulp. According to the model, there should be a strong increase in unlabeled cDC1 in the Δ vCD44BP infected mice. Data on cDC2 should be provided to help determine how specific the effect is for cDC1. Secondly, based on the authors model that the effect of vCD44BP is via CD44 on stromal cells, given the known role of CCL21 in cDC recruitment into the T zone, and the ability of WT MCMV to strongly downregulate splenic CCL21 expression (ref 27) it might be predicted that CCL21 expression is stronger in Δ vCD44BP versus WT MCMV infected spleens. This could be shown by QPCR and/or tissue staining for CCL21. Additionally, given the emphasis of the model on a cDC1 localization mechanism, literature should be cited that have established MCMV control depends on cDC1 (e.g., studies in *Batf3* KO mice).*

Some additional support for the cDC1 differential localization model is needed. As suggested by the Referee, we have added new data to the revised manuscript that supports differential DC localization post infection with Δ vCD44BP. These data include:

- (i) Quantification of DC distribution in the spleen by microscopy has been conducted in 3 independent experiments (All DCs using CD11c: **Fig 3g and h**, and **New Extended data Fig. 3f**; cDC1 using XCR1: **Extended data Fig. 3g and New Extended data Fig. 3h**; cDC2 using 33D1: **New Extended data Fig. 3i**)
- (ii) Quantification of DCs in the white versus red pulp by intravascular pulse labelling (**New Fig. 3f**) – see below.

Figure 3f. *vCD44BP interferes with DC migration into the splenic white pulp.* The frequency of IV⁻ DCs in the spleen of uninfected or mice MCMV or Δ vCD44BP infected mice at 36 hr pi. Data are pooled from 2 independent experiments (naïve, n = 5; MCMV and Δ vCD44BP infected, n = 8)

It is worth noting that standard lymphocyte isolation methods do not recover all the cells from tissues, and even in the spleen, Masopust and colleagues reported a two-fold difference in T cell recovery [ref 18]. Recovery becomes even more complicated when there are strong associations between cells, as is the case for FRCs and DCs. Given this caveat, it is impressive that we observed a

significant difference in the percentage of DCs within the white pulp (IV^{neg}) when spleens were processed for stromal preparation to assist dissociation of lymphocytes from stroma.

Data on cDC2 should be provided to help determine how specific the effect is for cDC1 and Given the emphasis of the model on a cDC1 localization mechanism, literature should be cited that have established MCMV control depends on cDC1. We apologise as it seems we created confusion about the role of cDC1 versus cDC2 in priming anti-viral responses in the MCMV infection model. Both cDC1 and cDC2 are involved in priming anti-viral CD8 T cell responses to MCMV [refs 19-23]. In our original manuscript we presented immunofluorescence imaging data for CD11c, which stains both DC subsets, and XCR1 which marks cDC1. We are now showing quantification for both cDC1 and cDC2 in the white pulp as described above. We have modified the text to clarify that both DC subsets are affected (page 7).

Is CCL21 expression stronger in $\Delta vCD44BP$ versus WT MCMV infected spleens?

As discussed above (see response to R2, #1), and as expected from the published literature [refs 24-26], CCL21 decreased post infection; however, there is no difference between WT MCMV versus $\Delta vCD44BP$ infection (**New Extended data Fig. 4p**). This result is consistent with our conclusion that vCD44BP has an impact on FRC-DC contact by interfering with CD44-HA interactions.

2: In Fig 4j, k, there is increased CD8 expansion in control chimeras infected with $\Delta vCD44BP$ virus and this fits with the reduced PFU in panel i. The idea here is that without vCD44BP blocking CD44 function (HA binding) in stromal cells, there is induction of a more robust protective CD8 T cell response. As in, the presence of CD44 is suggested to be needed for the more robust CD8 response. With this model, it would be predicted that mice lacking CD44 in stromal cells would have a reduced CD8 response, irrespective of which virus infects them, and that is what is seen. But what is puzzling is why this is not associated with a high PFU/spleen in the stromal CD44-deficient mice matching that observed in TC1- \rightarrow WT mice. This discrepancy with the model needs explanation.

We have added additional data for the experiments conducted in the CD44-deficient chimeras and the revised figure now includes data combined from 3 independent experiments. Replication of MCMV or $\Delta vCD44BP$ in mice lacking CD44 expression in the stromal compartment is equivalent to that of WT chimeras infected with MCMV. Importantly, as we found previously, in the TC1 \rightarrow *Cd44*^{-/-} chimeras there is no significant difference in viral loads between WT MCMV and $\Delta vCD44BP$ infections (**Revised Fig. 5b**) – see below.

Revised Fig. 5b. Viral loads in the spleen of chimeric mice. The viral loads for WT MCMV and $\Delta vCD44BP$ viruses are equivalent in the TC1 \rightarrow *Cd44*^{-/-} chimeras and comparable to those observed after WT MCMV infection of TC1 \rightarrow WT chimeras.

3: Given the broad expression of CD44 on hematopoietic cells (including cDCs and CD8 T cells) and established HA-dependent functions in these cells (e.g. ref 41) it seems unlikely that vCD44BP would only act via stromal cells. The authors may not have had CD44^{-/-} mice on the correct genetic background to perform CD44^{-/-} \rightarrow WT BM chimeras, but these would provide valuable complementary data if available.

Unfortunately, as noted by the Referee, we cannot generate reciprocal CD44^{-/-} \rightarrow WT chimera as it would require TC1 mice lacking CD44. We have clarified this point in the manuscript (page 10). Although it is possible that vCD44BP might affect additional HA-dependent functions, our current study highlights the remarkable impact of vCD44BP on FRC-DC interactions. As discussed, we have

added new data to the revised manuscript that show reduced DC migration in the chimera system when stromal CD44 is lacking (**New Fig. 5e**, see above and responses to comment 7 below for further details). Notably, we also found that binding of vCD44BP to FRCs is sufficient to reduce DC migration (**New Fig. 4I**), as is the absence of CD44 expression by FRCs (**New Fig. 4m**).

4: HA binding to CD44 is blocked by anti-CD44 (Fig. 1g). Given the overlap in vCD44BP and HA binding sites, it should be determined whether vCD44BP binding is blocked by the anti-CD44 antibody. This affects interpretation of data in Fig. 4d, e, where CD44 is stained in cells that are expressing vCD44BP. If the antibody and vCD44BP binding sites are competitive, one might expect expression of vCD44BP to alter CD44 staining.

We examined the question raised here in cells expressing CD44, including IC-21 and primary splenic FRCs, and found that prebinding of CD44 by vCD44BP-Fc does not alter the expression or detection of CD44 using the IM7 antibody in flow cytometry experiments (**Rebuttal Fig. 4**). For clarity, the anti-CD44 KM114 mAb that masks the HA-binding site was never used for detection of CD44. These data can be incorporated into Extended data Figure 4 if considered necessary.

Rebuttal Fig. 4 Anti-CD44 IM7 antibody and vCD44BP binding sites are not competitive. The IC21 macrophage cell line or primary FRCs were treated with vCD44BP-Fc prior to addition of the IM7 anti-CD44 antibody and expression levels of CD44 examined by flow cytometric analysis.

Thus, vCD44B binding does not impact detection of CD44 in our experiments.

5: For vCD44BP to be altering CD44 distribution on cultured FRC (Fig. 4f) it is presumably interacting with the CD44 ectodomain in the cultures. Is the inference that this is a cis interaction between vCD44BP and CD44 on the same cell membrane? In Fig 4f and g, the morphology of cultured uninfected FRC needs to be shown (and it should be indicated if these are primary FRC or an FRC cell line). Was any attempt made to examine TRC morphology in vivo in the WT vs ΔvCD44BP infected mice?

The questions raised here have been addressed in the responses to R1 and R2 above. To summarise:

- (i) vCD44BP and CD44 colocalise in infected primary splenic FRCs (**New Figure 4g and h**; and **New Extended data Figure 4I**)
- (ii) CD44 stabilises vCD44BP expression in infected primary splenic FRCs (**New Figure 4i**)
- (iii) FRCs in the splenic white pulp are readily infected by MCMV (**New Figure 4d**)
- (iv) the TRC network in the splenic white pulp of mice infected with ΔvCD44BP is differentially remodeled with increased branch lengths and an increased number of elongated and more complex PDPN+ fibers (**New Figure 4j and k**).

The data for naïve FRCs has been included alongside with information about the nature of the FRCs.

6: The rationale for using TC1 strain bone marrow in Fig 4h is not adequately introduced. In particular, the influence of Ly49H (and how its presence affected the earlier experiments) is not described. The strains of mice used (e.g., Balb/c vs C57BL/6) and rationale for them needs to be stated more clearly throughout.

We have modified the text to explain the rationale for the use of BALB/c mice, as in this strain the control of acute viral infection relies primarily of CD8 T cell responses rather than Ly49H-mediated NK cell responses (which occurs in C57BL/6 mice). The B6.TC1 congenic mouse strain, which lacks Ly49H, was used for the chimera as the CD44 KO mice are on a B6 background. In the absence of Ly49H, MCMV control in TC1 mice relies on CD8 T cell responses, similar to BALB/c mice. These points have been addressed in the revised manuscript (page 10).

7: The author's model is that CD44 in stromal cells promotes cDC1 movement into the splenic T cell zone of mice infected with $\Delta vCD44BP$ MCMV. This leads to the prediction that in the TC1 \rightarrow Cd44 $^{-/-}$ BM chimeras, $\Delta vCD44BP$ MCMV should fail to induce increased cDC1 movement into the T zone. Was that seen?

As predicted by the Referee, in TC1 \rightarrow Cd44 $^{-/-}$ chimera, $\Delta vCD44BP$ MCMV infection failed to induce increased DC migration into the splenic WP relative to WT MCMV.

The chimera experiments also showed that stromal CD44 is crucial for DC migration at steady state, as reduced DC localisation to the WP was observed in naïve TC1 \rightarrow Cd44 $^{-/-}$ chimera relative to the naïve TC1 \rightarrow WT. These important new data are presented in **New Fig. 5e** – see below.

New Fig. 5e. DC localization in the spleen is affected by stromal CD44. Spleen sections from chimera mice infected with MCMV or $\Delta vCD44BP$ at day 2 pi were stained with anti-CD11c to identify DCs and their location; the mean CD11c staining intensities within T cell zones are plotted. The number of white pulp areas examined are: TC1 \rightarrow WT (n = 79 naïve, n = 97 MCMV, n = 65 $\Delta vCD44BP$); TC1 \rightarrow Cd44 $^{-/-}$ (n = 58 naïve, n = 80 MCMV, n = 56 $\Delta vCD44BP$)

In summary, the DC trafficking data in the chimera independently demonstrate that CD44 expression by FRCs is required for optimal DC migration and that by limiting FRC-DC interactions, vCD44BP reduces DC migration thereby impairing early CD8 T cell responses.

8: In Fig 4l-n, the nature of the 'untreated' group needs clarification. It would seem that this should be the adjuvant (Montanide) treated group that was untreated with Fc fusion. This then makes it confusing why hTRAIL-Fc causes an increase in the response, and leaves it unclear if vCD44BP-Fc had any effect. If 'untreated' actually means no immunization, then a control 'adjuvant' only immunization group is needed (minimally to ensure that the hTRAIL-Fc is without effect).

We have now included new data so that in the mice that received Montanide there are 3 groups: PBS, TRAIL-Fc and vCD44BP-Fc. The untreated group received PBS only (no Montanide). We have added this revised Figure (to replace the previous Figure 4l-n) but moved it to Extended Data (**Revised Extended Data Fig. 5e-g**) and replaced the main Figure with new data from experiments that examined antigen-specific CD8 T cell responses as requested by Referee 1 (**New Fig. 5f-i**). These new data show that treatment with vCD44BP-Fc specifically impairs an antigen-specific CD8 T cell response in a vaccination model.

For clarity we have also amended the methods to clarify that the hTRAIL-Fc is a human protein that does not bind in the mouse.

References

1. Kumar V, Scandella E, Danuser R, Onder L, Nitschké M, Fukui Y, Halin C, Ludewig B, Stein JV. Global lymphoid tissue remodeling during a viral infection is orchestrated by a B cell-lymphotoxin-dependent pathway. *Blood*. 2010 Jun 10;115(23):4725-33.
2. Yang CY, Vogt TK, Favre S, Scarpellino L, Huang HY, Tacchini-Cottier F, Luther SA. Trapping of naive lymphocytes triggers rapid growth and remodeling of the fibroblast network in reactive murine lymph nodes. *PNAS USA*. 2014 Jan 7;111(1):E109-18.

3. Acton SE, Farrugia AJ, Astarita JL, Mourão-Sá D, Jenkins RP, Nye E, Hooper S, van Blijswijk J, Rogers NC, Snelgrove KJ, Rosewell I, Moita LF, Stamp G, Turley SJ, Sahai E, Reis e Sousa C. Dendritic cells control fibroblastic reticular network tension and lymph node expansion. *Nature*. 2014 Oct 23;514(7523):498-502.
4. Astarita JL, Cremasco V, Fu J, Darnell MC, Peck JR, Nieves-Bonilla JM, Song K, Kondo Y, Woodruff MC, Gogineni A, Onder L, Ludewig B, Weimer RM, Carroll MC, Mooney DJ, Xia L, Turley SJ. The CLEC-2-podoplanin axis controls the contractility of fibroblastic reticular cells and lymph node microarchitecture. *Nat Immunol*. 2015 Jan;16(1):75-84.
5. Novkovic M, Onder L, Cupovic J, Abe J, Bomze D, Cremasco V, Scandella E, Stein JV, Bocharov G, Turley SJ, Ludewig B. Topological Small-World Organization of the Fibroblastic Reticular Cell Network Determines Lymph Node Functionality. *PLoS Biol*. 2016 Jul 14;14(7):e1002515.
6. Martin-Villar E, Fernandez-Munoz B, Parsons M, Yurrita MM, Megias D, Perez-Gomez E, Jones GE, Quintanilla M. Podoplanin associates with CD44 to promote directional cell migration. *Mol Biol Cell*. 2010 21, 4387-4399. 10.1091/mbc.E10-06-0489.
7. de Winde CM, Makris S, Millward LJ, Cantoral-Rebordinos JA, Benjamin AC, Martinez VG, Acton SE. Fibroblastic reticular cell response to dendritic cells requires coordinated activity of podoplanin, CD44 and CD9. *2021 J Cell Sci* 134. 10.1242/jcs.258610.
8. Montero-Montero L, Renart J, Ramirez A, Ramos C, Shamhood M, Jarcovsky R, Quintanilla M, Martin-Villar E. Interplay between Podoplanin, CD44s and CD44v in Squamous Carcinoma Cells. *Cells* 2020 9. 10.3390/cells9102200.
9. Dalod M, Hamilton T, Salomon R, Salazar-Mather TP, Henry SC, Hamilton JD, Biron CA. Dendritic cell responses to early murine cytomegalovirus infection: subset functional specialization and differential regulation by interferon alpha/beta. *J Exp Med*. 2003 Apr 7;197(7):885-98.
10. Alexandre YO, Cocita CD, Ghilas S, Dalod M. Deciphering the role of DC subsets in MCMV infection to better understand immune protection against viral infections. *Front Microbiol*. 2014 Jul 29;5:378.
11. Wikstrom ME, Fleming P, Kuns RD, Schuster IS, Voigt V, Miller G, Clouston AD, Tey SK, Andoniou CE, Hill GR, Degli-Esposti MA. Acute GVHD results in a severe DC defect that prevents T-cell priming and leads to fulminant cytomegalovirus disease in mice. *Blood*. 2015 Sep 17;126(12):1503-14.
12. Koszinowski UH, Del Val M, Reddehase MJ. Cellular and molecular basis of the protective immune response to cytomegalovirus infection. *Curr Top Microbiol Immunol*. 1990;154:189-220.
13. Lathbury LJ, Allan JE, Shellam GR, Scalzo AA. Effect of host genotype in determining the relative roles of natural killer cells and T cells in mediating protection against murine cytomegalovirus infection. *J Gen Virol*. 1996 Oct;77 (Pt 10):2605-13.
14. Sumaria N, van Dommelen SL, Andoniou CE, Smyth MJ, Scalzo AA, Degli-Esposti MA. The roles of interferon-gamma and perforin in antiviral immunity in mice that differ in genetically determined NK-cell-mediated antiviral activity. *Immunol Cell Biol*. 2009 Oct;87(7):559-66.
15. Andrews DM, Estcourt MJ, Andoniou CE, Wikstrom ME, Khong A, Voigt V, Fleming P, Tabarias H, Hill GR, van der Most RG, Scalzo AA, Smyth MJ, Degli-Esposti MA. Innate immunity defines the capacity of antiviral T cells to limit persistent infection. *J Exp Med*. 2010 Jun 7;207(6):1333-43.

16. Onder L, Cheng HW, Ludewig B. Visualization and functional characterization of lymphoid organ fibroblasts. *Immunol Rev* 2022 306, 108-122. 10.1111/imr.13051.
17. Muto J, Morioka Y, Yamasaki K, Kim M, Garcia A, Carlin AF, Varki A, Gallo RL. Hyaluronan digestion controls DC migration from the skin. *J Clin Invest*. 2014 Mar;124(3):1309-19.
18. Steinert EM, Schenkel JM, Fraser KA, Beura LK, Manlove LS, Igyártó BZ, Southern PJ, Masopust D. Quantifying Memory CD8 T Cells Reveals Regionalization of Immunosurveillance. *Cell*. 2015 May 7;161(4):737-49.
19. Dalod M, Hamilton T, Salomon R, Salazar-Mather TP, Henry SC, Hamilton JD, Biron CA. Dendritic cell responses to early murine cytomegalovirus infection: subset functional specialization and differential regulation by interferon alpha/beta. *J Exp Med*. 2003 Apr 7;197(7):885-98.
20. Torti N, Walton SM, Murphy KM, Oxenius A. Batf3 transcription factor-dependent DC subsets in murine CMV infection: differential impact on T-cell priming and memory inflation. *Eur J Immunol*. 2011 Sep;41(9):2612-8.
21. Busche A, Jirno AC, Welten SP, Zischke J, Noack J, Constabel H, Gatzke AK, Keyser KA, Arens R, Behrens GM, Messerle M. Priming of CD8+ T cells against cytomegalovirus-encoded antigens is dominated by cross-presentation. *J Immunol*. 2013 Mar 15;190(6):2767-77.
22. Alexandre YO, Cocita CD, Ghilas S, Dalod M. Deciphering the role of DC subsets in MCMV infection to better understand immune protection against viral infections. *Front Microbiol*. 2014 Jul 29;5:378.
23. Holtappels R, Büttner JK, Freitag K, Reddehase MJ, Lemmermann NA. Modulation of cytomegalovirus immune evasion identifies direct antigen presentation as the predominant mode of CD8 T-cell priming during immune reconstitution after hematopoietic cell transplantation. *Front Immunol*. 2024 Feb 15;15:1355153.
24. Benedict CA, De Trez C, Schneider K, Ha S, Patterson G, Ware CF. Specific remodeling of splenic architecture by cytomegalovirus. *PLoS Pathog*. 2006 Mar;2(3):e16.
25. Mueller SN, Hosiawa-Meagher KA, Konieczny BT, Sullivan BM, Bachmann MF, Locksley RM, Ahmed R, Matloubian M. Regulation of homeostatic chemokine expression and cell trafficking during immune responses. *Science*. 2007 Aug 3;317(5838):670-4.
26. Bekiaris V, Timoshenko O, Hou TZ, Toellner K, Shakib S, Gaspal F, McConnell FM, Parnell SM, Withers D, Buckley CD, Sweet C, Yokoyama WM, Anderson G, Lane PJ. Ly49H+ NK cells migrate to and protect splenic white pulp stroma from murine cytomegalovirus infection. *J Immunol*. 2008 May 15;180(10):6768-76.

Dear Ursula,

Thank you for the opportunity to address the reviewers' comments on our manuscript 2024-01-00034 "**Fibroblastic reticular cells control the initiation of adaptive T cell responses via CD44**".

In this study we discovered that CD44 expressed by stromal cells is critical for the efficient trafficking of DCs and the subsequent generation of adaptive T cell responses. Reviewer 2 was entirely satisfied with the revised manuscript, and we consider the primary concerns raised by reviewers 1 and 3 to be addressable in a timely manner.

- 1. Rev 1: Cannot exclude that in vivo trans-binding of vCD44BP in infected FRCs acts uniquely by interrupting FRC-expressed CD44 binding to surface HA on DCs as opposed to CD44 on other cells. A similar point was raised by Rev 3.**

The chimera (CD44 KO TC1 -> WT TC1) that would formally exclude the possibility that vCD44BP targets CD44 expressed by haematopoietic cells could not be generated as CD44 KO TC1 mice do not exist. Since viral replication, T cell responses and DC migration in the TC1-> CD44 KO chimeric mice infected with WT MCMV were not significantly different to those induced by infection with the Δ vCD44BP mutant we felt this provided compelling evidence that vCD44BP functions by targeting CD44 expressed by stromal cells.

However, we can formally test whether vCD44BP mediates its effects by binding to CD44 expressed by haematopoietic cells by deleting CD44 in TC1-derived haematopoietic stem cells via CRISPR and generating appropriate chimeras.

- 2. Rev 1: Data on the 3D cultures is compelling; however, the control experiments should also have been performed using pre-incubation of DCs with vCD44BP-Fc.**

This experiment can be performed and, in combination with the experiment proposed to address point 1 as discussed above, will definitively establish whether vCD44BP can intrinsically affect DCs.

- 3. Rev 1: Without the use of FRC-specific Cre-drivers, the authors cannot exclude the contribution of overall fibroblast vs FRC CD44 in driving antiviral T cell responses.**

We emphasised the impact of vCD44BP on white pulp FRCs (TRC) as these cells are specifically targeted by MCMV and play a critical role in DC migration and the generation of adaptive immune responses. When we refer to FRCs, we include all reticular fibroblasts and it is possible that vCD44BP may also target CD44 expressed by other fibroblast populations, such as red pulp fibroblasts. This does not detract from one of our major findings i.e. that CD44 expressed by stromal fibroblastic cells is critical for the efficient trafficking of DCs and the subsequent generation of adaptive T cell responses. We will address this point to clarify the contribution of fibroblasts, rather than a particular fibroblast subset.

A detailed response to all the points raised by the reviewers is provided below.

Referee #1 (Remarks to the Author):

The authors were asked to demonstrate the binding of vCD44BP to FRC-expressed CD44 in vivo and validate the function of FRC-provided CD44 as universal factor that determines the efficiency of CD8+ T cell responses. The reviewer appreciates that in the revised manuscript the authors have made a considerable effort to use a combination of approaches to strengthen the evidence vCD44BP binds to FRC CD44. Nevertheless, a key concern remains that the authors cannot formally demonstrate that interference of CD44-mediated FRC-DC interaction by vCD44BP is how this viral protein delays CD8+ T cell responses. The reviewer does not dispute that in vitro infected FRC cell lines or primary splenic FRCs co-express CD44 and vCD44BP (newly added Figure 4g and 4h). Rather, the added in vitro approaches do not exclude that in vivo trans-binding of vCD44BP in infected FRCs acts uniquely by interrupting FRC-expressed CD44 binding to surface HA on DCs (as opposed to CD44 on other cells or secreted HA).

The reviewer is asking for additional evidence to show that vCD44BP in infected FRCs does not act by affecting CD44 activities on cells other than FRCs.

Priming of CD8 T cells requires interactions with DCs in the splenic white pulp. DCs traffic to the white pulp via fibroblast networks [*Nat Rev Immunol* doi: 10.1038/s41577-023-00857-x; *JEM* doi.org/10.1084/jem.20221220].

We have demonstrated that DC trafficking is reduced in the presence of vCD44BP both in vitro and in vivo. In vivo we have demonstrated this by comparing infection of WT MCMV with a virus lacking vCD44BP in WT hosts (Fig. 3g-h and Extended data Fig. 3f-i) and in chimeric hosts lacking stromal CD44 expression (Fig. 5e). The analysis of chimeric mice clearly demonstrates that CD44 expression by stromal cells is essential for optimal DC migration (Fig. 5e).

Independently of MCMV infection, we have demonstrated a role of CD44, by blocking its capacity to bind to HA, both in inflammation (Fig. 5f-i and Extended data Fig. 5d) and immunization (Fig. 5j-m and Extended data Fig. 5e-g) models.

In vitro, we have also demonstrated that interfering with FRC CD44 and DC-HA binding (by pre-incubation of FRC with vCD44BP-Fc) severely interferes with DC trafficking (Fig. 4l-m).

The reviewer seeks evidence to exclude that in vivo trans-binding of vCD44BP in infected FRCs functions by interfering with CD44 binding on cells other than FRCs. To explain our results, vCD44BP would have to bind to CD44 on DCs, but this has been excluded by the results obtained in our chimera system which will be further validated by **creating a reverse chimera i.e. CD44^{-/-} TC1 into WT using a CRISPR approach to delete CD44 from the TC1 hematopoietic stem cells that will be used to generate the chimera. This experiment will also address R3's main outstanding concern.**

Furthermore, as shown below:

- Most splenic fibroblasts express CD44 (~80%) compared to a small fraction of endothelial cells (~15%)
- Splenic fibroblasts express significantly more CD44 than splenic DCs (both cDC1 and cDC2)

[Redacted]

[Figure Redacted]

Thus, fibroblasts are the main cell population expressing CD44 in the spleen. **The additional experiments proposed above will conclusively define the potential role of hematopoietic cells, including DCs.**

[Redacted]

[Redacted]

[Figure Redacted]

[Redacted]

Since the spleen lacks afferent lymphatics, and endothelial cells in the spleen express negligible levels of CD44, the differences in DC trafficking observed after WT MCMV versus Δ vCD44BP infection cannot be attributed to endothelium-mediated effects.

Data on the 3D cultures is compelling; however, the control experiments should also have been performed using pre-incubation of DCs with vCD44BP-Fc. Should this have induced a similar effect in reducing DC migration then this draws into question how vCD44BP can interact with neighboring cells in vivo, and the effect of therapeutic vCD44BP-Fc treatment in the vaccination and flu infection settings (Figure 5f-m). While the authors argue that reduced expression of vCD44BP in Cd44^{-/-} cells indicates a cis-interaction between FRC CD44 and vCD44BP, it remains unclear how CD44 co-expression is needed to stabilize a transmembrane protein (Figure 4i).

The migration experiments can be repeated as suggested by the Reviewer i.e expose DCs to vCD44BP-Fc prior to co-culture with FRCs.

How CD44 stabilises expression of vCD44BP requires extensive biochemical studies that we feel are beyond the scope of the current report which focuses on how the vCD44BP viral protein interferes with stromal based mechanisms to impair T cell responses.

The authors emphasize the importance of white pulp (WP) FRCs – DC interaction. FRCs are relatively rare in the spleen, making the proportion of infected FRCs high relative to the more abundant red pulp (RP) fibroblasts (Ext. Fig. 4e). However, it is important to note that a similar number of WP and RP fibroblasts are infected by the virus (Extended Fig. 4d). Moreover, histological representation shows that the infected RP FRCs are highly abundant in proximity to the marginal zone, a site that supports DCs and T cells entering the splenic WP (Ext. Fig. 4f). Without the use of FRC-specific Cre-drivers, the authors cannot exclude the contribution of overall fibroblast vs FRC CD44 in driving antiviral T cell responses. The inability to pinpoint a function to WP FRCs is also relevant for the interpretation of Cd44^{-/-} BM chimera experiments.

The reviewer is correct, and it is possible that RP FRCs are also involved. When we use the term FRCs we include all reticular fibroblasts. **We will modify the text, and in particular the Discussion, to clarify that the effects described here define the contribution of fibroblasts, rather than a particular fibroblast subset, in driving antiviral T cell responses. The novelty of our study lies in the role of fibroblastic CD44 in initiating adaptive immune responses and how this has been targeted by a viral immune evasin.**

The Reviewer suggests that FRC-specific Cre-drivers are required to exclude the contribution of overall fibroblast vs FRC CD44 in driving antiviral T cell responses. These mice do not exist - examining T cell responses in our infection model requires mice on a BALB/c background and FRC-specific-cre and CD44-flox mouse lines do not exist. Importantly, as noted above, our focus is on fibroblasts rather than a specific subset, and the role for fibroblastic CD44 can be addressed by additional chimera experiments as proposed.

Minor comments:

In MCMV-infected primary splenic FRCs (Figure 4h), there does not seem to be a redistribution of surface CD44 to FRC filopodia as seen of the FRC cell line (Figure 4e). It is also unclear why this antibody does not work for in situ imaging of splenic sections from MCMV-infected mice.

In Figure 4h cells were permeabilised after staining for vCD44BP to examine the actin cytoskeleton and therefore CD44 redistribution is difficult to assess. The purpose of the experiment shown in this Figure was to determine whether CD44 and vCD44BP interact in the same cell after infection.

We can specifically examine CD44 distribution in primary FRCs after infection with MCMV and vCD44BP as done in Figure 4e.

We do not specifically know why the anti-vCD44BP antibodies don't work for in situ imaging of splenic sections; however, this is not unusual for highly glycosylated proteins (e.g. PLoS One. 2018 Feb 9;13(2):e0192506).

The authors have added images to demonstrate the proximity of viral IE1 and PDPN. In Figure 4e, a nuclear marker and 3D reconstruction or separate channels should be shown to support that these are FRCs and not closely interacting cells.

We have already performed high-resolution single molecule analyses that can be added to the manuscript.

All representative FACS plots should be shown for the quantified data (ie. including tetramer staining).

All FACS plots are available and can be added to the relevant Figures.

Referee #2 (Remarks to the Author):

The authors have meaningfully and thoroughly responded to my comments from the original submission.

Referee #3 (Remarks to the Author):

The authors have made extensive revisions in response to the reviewer comments and the manuscript is improved. However, some specific concerns remain:

It remains unclear why FRC would be 'the main population able to bind vCD44BP-Fc after infection'. Do FRC really express more CD44 than various hematopoietic cells (e.g., effector CD8 cells)? This comparison is not shown. If CD44 levels are not higher on FRC but they bind more vCD44BP-Fc (than hematopoietic cells) is this because they have a form of CD44 that binds better?

As shown (see response to R1 and below) splenic FRCs express significantly higher levels of CD44 compared to hematopoietic cells that can participate in the initiation of immune responses i.e DCs.

Notably, expression of CD44 on splenic fibroblastic cells is significantly higher than on lymphocytes, including effector T cells (see below).

Given that the authors have not been able to test the role of CD44 on hematopoietic cells in vivo (as they apparently have not intercrossed the CD44 KO and TC1 B6 mouse lines), they have not excluded an action of vCD44BP via CD44 on hematopoietic cells. Indeed, although the result in Fig. 4b is taken to argue that all the vCD44BP action depends on stromal CD44, there remains an effect of vCD44BP

deletion in the TC1-> Cd44 KO chimeras that approaches that of the WT. Moreover, the authors show in other parts of the manuscript that vCD44BP can act 'in trans' when provided in a soluble format. How can it be excluded that even in transmembrane form (whether in FRC or other cells) it is not acting to engage CD44 on other cells? In the absence of data in full CD44 KO mice (and CD44 KO TC1 -> WT TC1 chimeras), this possibility must at least be discussed.

As noted in the response to R1's comments, we will generate a reverse chimera i.e. **CD44^{-/-} TC1 into WT using a retrogenic approach to delete CD44 from the TC1 hematopoietic stem cells.** This experiment will address the role of vCD44BP on CD44 expressed by hematopoietic cells.

In addition, as suggested by the Reviewer we can discuss how vCD44BP might also function "in trans".

Line 231: 'This would account for the increased numbers of virus specific SLEC CD8 T cells observed after infection with the Δ vCD44BP virus.' In this sentence, 'would' should be changed to 'could'.

This will be changed as requested.

The data in figure 4j, k appear quite striking. The term 'branch length' should be clarified. Is this referring to the length of individual FRC membrane processes? Moreover, if the processes are longer, it is not intuitive (and not obvious from the image) how this would lead to such a marked increase in 'branches per FOV'. (Longer processes might be expected to correspond to fewer branch points).

Branch length refers to the length of individual podoplanin processes. The number of branches per FOV was calculated using a skeleton analysis plug-in, as detailed in the methods. As well as longer branches, there are also more complex branches in the Δ vCD44BP spleen which the plug-in estimates as more branch points. This will be clarified.

The type of DC used for the in vitro assays is mentioned in the context of Versican staining. It should be clarified in the legend whether BMDC were used for all the in vitro assays (e.g., Fig. 4l, m).

BMDC were used for all *in vitro* assays, and this will be clarified in the relevant figure legends.

The text (line 296) states 'DCs in the co-cultures expressed the CD44 ligand HA'. However, ED Fig. 4r shows staining for Versican. Versican is not mentioned in the text and it is not explained why it is a sufficient marker to determine HA levels.

Versican is the only marker available to determine HA levels. Relevant references and an explanation for the use of this reagent will be added to the text as suggested.

Dear Dr Weiss,

Thank you for the opportunity to address the Reviewers' comments on our manuscript 2024-01-00034 "Fibroblastic reticular cells control the initiation of adaptive T cell responses via CD44".

Reviewer 2 was entirely satisfied with the revised manuscript, and we have carefully addressed the outstanding comments raised by Reviewers 1 and 3 in the current revision.

We thank all the Reviewers for their valuable insights and feedback, which have helped to clarify and strengthen the work presented in our manuscript. A detailed response to each of the points raised by the Reviewers is provided below.

Referee #1

R1 #1. The authors were asked to demonstrate the binding of vCD44BP to FRC-expressed CD44 in vivo and validate the function of FRC-provided CD44 as universal factor that determines the efficiency of CD8+ T cell responses. The reviewer appreciates that in the revised manuscript the authors have made a considerable effort to use a combination of approaches to strengthen the evidence vCD44BP binds to FRC CD44. Nevertheless, a key concern remains that the authors cannot formally demonstrate that interference of CD44-mediated FRC-DC interaction by vCD44BP is how this viral protein delays CD8+ T cell responses. The reviewer does not dispute that in vitro infected FRC cell lines or primary splenic FRCs co-express CD44 and vCD44BP (newly added Figure 4g and 4h). Rather, the added in vitro approaches do not exclude that in vivo trans-binding of vCD44BP in infected FRCs acts uniquely by interrupting FRC-expressed CD44 binding to surface HA on DCs (as opposed to CD44 on other cells or secreted HA).

Response: The reviewer is asking for additional evidence to show that vCD44BP in infected FRCs does not act by affecting CD44 activities on cells other than fibroblasts/FRCs. To explain our results, vCD44BP would have to bind to CD44 on DCs. We have addressed this point by undertaking additional experiments as suggested by the reviewer.

Firstly, we have repeated the DC migration experiments in 3D cultures where we pre-incubated DCs with vCD44BP-Fc prior to co-culture with FRCs. Unlike pre-incubation of vCD44BP-Fc with FRCs, which reduced DC migration, pre-incubation with DCs had no impact on DC migration in the 3D co-cultures (**revised Fig. 4I** – see below). **These data exclude the possibility that vCD44BP acts directly on DCs to influence their migration.**

Figure 4 (I) Primary splenic FRCs from B6 mice or BM-derived DCs were pre-treated with vCD44BP-Fc (40min, 4°C) or left untreated, combined, and grown in a 3-dimensional (3D) gel culture system. The distance and velocity of movement of DCs that were in contact with FRCs quantified by time-lapse live imaging are shown. Data are pooled from three independent experiments with n = 62 untreated, n = 60 vCD44BP-Fc treated FRC, and n = 58 vCD44BP-Fc treated DC.

We have now also confirmed that the effects of vCD44BP are not mediated by interactions with hematopoietic cells, including DCs, *in vivo*. This conclusion is based on new experiments in which we generated CD44 chimeric mice using a CRISPR-Cas9 based approach. Specifically, we targeted CD44 in hematopoietic stem cells from TC1 mice, generated chimeric mice and demonstrated specific loss of CD44 expression in leukocytes (CD44 guides vs scrambled guides; **Extended data Fig. 5h**). Chimeric mice were then infected with WT or Δ vCD44BP MCMV, and CD8 T cell responses and the capacity to limit MCMV infection examined. We previously showed that the improved antiviral CD8 T cell responses and reduced viral loads observed after infection with Δ vCD44BP were lost when the host lacked CD44 on stromal cells, and that finding was confirmed in these new experiments (**Fig 5e and f**; gScr \rightarrow Cd44^{-/-}). In contrast, when CD44 was absent on hematopoietic cells (gCd44 \rightarrow WT), mice infected with Δ vCD44BP had significantly lower viral loads and better antiviral CD8 T cell responses compared to WT MCMV (**Fig. 5e and f** – see below), a response equivalent to that observed in chimeric mice with normal CD44 expression on hematopoietic cells (generated using a scrambled guide that does not target CD44 i.e. gScr \rightarrow WT). **These findings confirm that vCD44BP delays the anti-viral CD8 T cell response by targeting CD44 expressed by stromal cells.**

Figure 5 (e). Viral loads in the spleens of chimeric mice infected with MCMV (grey) or Δ vCD44BP (blue) are shown. **(f)** Number of virus-specific m45⁺ CD8 T cells at day 7 PI in the spleens of chimeric mice infected with MCMV (grey), Δ vCD44BP (blue) or uninfected (white) are shown. The data in (e and f) are combined from two independent experiments (gScr \rightarrow WT, n = 7 – 8; gCd44 \rightarrow WT, n = 8; gScr \rightarrow Cd44^{-/-} n = 8; naïve n = 3 – 4/ group).

The additional experiments have investigated and conclusively excluded a role for CD44 on hematopoietic cells, including DCs, in controlling DC migration and the initiation of anti-viral CD8 T cell responses.

Since the spleen lacks afferent lymphatics, and only a very small fraction of endothelial cells expresses CD44, (see response to R3 and **Extended data Figure 5l and m**), the differences in DC trafficking observed after WT MCMV versus Δ vCD44BP infection cannot be attributed to activities mediated by endothelial cells.

R1 #2. Data on the 3D cultures is compelling; however, the control experiments should also have been performed using pre-incubation of DCs with vCD44BP-Fc. Should this have induced a similar effect in reducing DC migration then this draws into question how vCD44BP can interact with neighboring cells *in vivo*, and the effect of therapeutic vCD44BP-Fc treatment in the vaccination and flu infection settings (Figure 5f-m). While the authors argue that reduced expression of vCD44BP in Cd44^{-/-} cells indicates a cis-interaction between FRC CD44 and vCD44BP, it remains unclear how CD44 co-expression is needed to stabilize a transmembrane protein (Figure 4i).

Response: As discussed above, the control experiment suggested by the Reviewer – i.e. pre-incubation of DCs with vCD44BP-Fc, has been conducted and the results (shown above R1 #1) have been included in **revised Fig. 4l**.

Elucidating how CD44 stabilises expression of vCD44BP requires extensive biochemical studies that we feel are beyond the scope of the current report which focuses on how the vCD44BP viral protein interferes with stromal based mechanisms to impair T cell responses.

R1 #3. The authors emphasize the importance of white pulp (WP) FRCs – DC interaction. FRCs are relatively rare in the spleen, making the proportion of infected FRCs high relative to the more abundant red pulp (RP) fibroblasts (Ext. Fig. 4e). However, it is important to note that a similar number of WP and RP fibroblasts are infected by the virus (Extended Fig. 4d). Moreover, histological representation shows that the infected RP FRCs are highly abundant in proximity to the marginal zone, a site that supports DCs and T cells entering the splenic WP (Ext. Fig. 4f). Without the use of FRC-specific Cre-drivers, the authors cannot exclude the contribution of overall fibroblast vs FRC CD44 in driving antiviral T cell responses. The inability to pinpoint a function to WP FRCs is also relevant for the interpretation of Cd44^{-/-} BM chimera experiments.

Response: The novelty of our study resides in identifying a previously unrecognised role for CD44 expressed by fibroblastic stromal cells in initiating adaptive immune responses, and in demonstrating how this process is subverted by a viral immune evasin.

The new migration assays and chimeric mouse experiments have conclusively excluded a role for hematopoietic and DC-derived CD44 in modulating CD8 T cell responses. **Multiple lines of evidence support a principal role for CD44 expressed by FRCs.** Specifically:

- The virus exhibits a strong tropism for FRCs, with ~60% of cells in the white pulp becoming infected, compared to less than 10% of fibroblasts in the red pulp.
- Extensive remodelling of the podoplanin-positive FRC network within the white pulp is observed after MCMV infection, and this remodelling is modulated by vCD44BP.

We agree that in the absence of FRC-specific Cre-drivers we cannot exclude a contribution from other fibroblastic stromal cells (e.g marginal zone fibroblasts) – see revised Discussion. Examining antiviral T cell responses in MCMV infection requires mice on a BALB/c background and the FRC-specific Cre-drivers suggested by the Reviewer, e.g. ccl19-cre and CD44-flox mouse lines, do not exist on this genetic background.

Most importantly, our findings reveal **a novel role for fibroblastic CD44 in regulating CD8 T cell responses and uncover an unappreciated immunomodulatory function of this stromal compartment.** To better reflect this newly appreciated role of fibroblastic CD44, and in line with the Reviewer's suggestion, we have revised the title to "**Fibroblastic stromal cells control the initiation of adaptive T cell responses via CD44**".

R1. Minor comments:

R1 M1. In MCMV-infected primary splenic FRCs (Figure 4h), there does not seem to be a redistribution of surface CD44 to FRC filopodia as seen of the FRC cell line (Figure 4e). It is also unclear why this antibody does not work for in situ imaging of splenic sections from MCMV-infected mice.

Response: In Figure 4h, cells were permeabilised after staining for vCD44BP to allow visualization of the actin cytoskeleton, making it difficult to assess CD44 redistribution. The primary objective of this experiment was to determine whether CD44 and vCD44BP colocalize within the same cell after infection.

CD44 distribution in primary FRCs after infection with MCMV and vCD44BP is shown below and is equivalent to that observed in the FRC cell line shown in Figure 4e – now Extended Data Figure 4k.

Rebuttal Figure 1. Primary splenic FRC were infected with the respective viruses and stained for CD44. The proportion of infected FRC was measured by flow cytometry and was ~60%.

We do not know why the anti-vCD44BP antibodies fail to work for in situ imaging of splenic sections; however, this is not unusual for highly glycosylated proteins, as glycosylation can limit antibody accessibility (PMID: 29425242). It is worth noting that in situ imaging of CD44 in splenic sections is similarly difficult and, to our knowledge, not reported in the published literature.

R1 M2. The authors have added images to demonstrate the proximity of viral IE1 and PDPN. In Figure 4e, a nuclear marker and 3D reconstruction or separate channels should be shown to support that these are FRCs and not closely interacting cells.

Response: We thank the Reviewer for highlighting this; high-resolution orthogonal images, which include a nuclear marker, have been added (revised **Figure 4e**) – see below.

Figure 4(e). Representative confocal 3D orthogonal views of infected IE1+ (green), PDPN+ FRCs (red) and surrounding CD11c+ DCs (white); blue = nuclei. Representative of data from two independent experiments.

R1 M3. All representative FACS plots should be shown for the quantified data (ie. including tetramer staining).

Response: All flow cytometry plots have been added - see Extended Data Figures 3, 4 and 5.

Referee #2

The authors have meaningfully and thoroughly responded to my comments from the original submission.

Response: We thank the Reviewer for their previous comments.

Referee #3

The authors have made extensive revisions in response to the reviewer comments and the manuscript is improved. However, some specific concerns remain:

R3 #1. It remains unclear why FRC would be 'the main population able to bind vCD44BP-Fc after infection'. Do FRC really express more CD44 than various hematopoietic cells (e.g., effector CD8 cells)? This comparison is not shown. If CD44 levels are not higher on FRC but they bind more vCD44BP-Fc (than hematopoietic cells) is this because they have a form of CD44 that binds better?

Response: As shown right (**revised Extended data Figure 5I**), splenic fibroblastic stromal cells (FC) express significantly higher levels of CD44 compared to endothelial cells (EC) as well as hematopoietic cells that can participate in the initiation of immune responses i.e. DC.

Rebuttal Figure 2. CD44 expression on splenic fibroblastic stromal cells (FC), effector CD8 T cells and activated CD4 T cells, purified from day 7 post-MCMV infection, are shown. The dashed line shows the peak CD44 expression on CD44⁺ FC; the solid line separates CD44 positive and negative populations.

Notably, expression of CD44 on splenic fibroblastic stromal cells is also higher than on lymphocytes, including effector CD8 and CD4 T cells (**Rebuttal Figure 2**).

It is worth noting that vCD44BP delays, rather than completely inhibits, the antiviral CD8 T cell response. **Its selective targeting of a specific cell subset, fibroblastic cells, rather than widespread effects on multiple cell populations, reflects a highly focused immune evasion strategy which highlights the critical role of fibroblastic CD44 in orchestrating effective immune responses and shows that CMV has evolved a mechanism to specifically target this critical checkpoint.**

R3 #2. Given that the authors have not been able to test the role of CD44 on hematopoietic cells in vivo (as they apparently have not intercrossed the CD44 KO and TC1 B6 mouse lines), they have not excluded an action of vCD44BP via CD44 on hematopoietic cells. Indeed, although the result in Fig. 4b is taken to argue that all the vCD44BP action depends on stromal CD44, there remains an effect of vCD44BP deletion in the TC1-> Cd44 KO chimeras that approaches that of the WT. Moreover, the authors show in other parts of the manuscript that vCD44BP can act 'in trans' when provided in a soluble format. How can it be excluded that even in transmembrane form (whether in FRC or other cells) it is not acting to engage CD44 on other cells? In the absence of data in full CD44 KO mice (and CD44 KO TC1 -> WT TC1 chimeras), this possibility must at least be discussed.

Response: As noted in our response to Reviewer 1, we generated chimeric mice lacking hematopoietic CD44 using a CRISPR-Cas9 approach. These experiments ruled out any effect of

vCD44BP via CD44 on hematopoietic cells and conclusively demonstrated that CD44 on stromal cells is required for vCD44BP to delay the anti-viral CD8 T cell response.

Altogether the available data suggest that the primary function of vCD44BP is to target fibroblastic CD44 as a specific immune evasion strategy as discussed above (R3 #1). This important point has been discussed in the revised manuscript (page 13).

R3 #3. Line 231: 'This would account for the increased numbers of virus specific SLEC CD8 T cells observed after infection with the Δ vCD44BP virus.' In this sentence, 'would' should be changed to 'could'.

Response: Modified as requested.

R3 #4. The data in figure 4j, k appear quite striking. The term 'branch length' should be clarified. Is this referring to the length of individual FRC membrane processes? Moreover, if the processes are longer, it is not intuitive (and not obvious from the image) how this would lead to such a marked increase in 'branches per FOV'. (Longer processes might be expected to correspond to fewer branch points).

Response: Branch length refers to the length of individual podoplanin processes. This has been clarified in the revised Figure 4k legend. The number of branches per FOV was calculated using a skeleton analysis plug-in, as detailed in the methods. As well as longer branches, there are also more complex branches in the Δ vCD44BP spleen which the plug-in estimates as more branch points. This has been clarified in the results section on page 9.

R3 #5. The type of DC used for the *in vitro* assays is mentioned in the context of Versican staining. It should be clarified in the legend whether BMDC were used for all the *in vitro* assays (e.g., Fig. 4l, m).

Response: BMDC were used for all *in vitro* assays, and this has been clarified in the relevant figure legends as suggested.

R3 #6. The text (line 296) states 'DCs in the co-cultures expressed the CD44 ligand HA'. However, ED Fig. 4r shows staining for Versican. Versican is not mentioned in the text and it is not explained why it is a sufficient marker to determine HA levels.

Response: The most reliable method to measure HA expression is by using HA-binding proteins or protein domains that specifically bind HA with high affinity. Versican is a large proteoglycan whose G1 domain binds HA with high affinity and it the best marker available to determine HA levels. As suggested by the Reviewer, the above details have been added to the legend for Extended Data Fig. 4r.

Dear Dr Weiss,

Thank you for the opportunity to address the remaining Referees' comments on our manuscript 2024-01-00034 "Fibroblastic reticular cells control the initiation of adaptive T cell responses via CD44".

We sincerely thank the Referees for their valuable insights, which have improved the clarity of our findings and further strengthened the manuscript. In the current version of the manuscript, we have carefully addressed all their outstanding comments.

A detailed point-by-point response to the Referees' comments is provided below.

Referee #1:

The added experiments using Crispr-Cas9 knockdown of Cd44 now formally demonstrate that the effect of vCD44BP on viral titers and CD8+ T cell responses is independent of CD44 on the hematopoietic compartment. However, the observation that the absence of CD44 on the hematopoietic compartment is dispensable for viral control, begs the question whether CD44-deficient DCs are still capable of binding to HA, as this is the primary molecule thought to mediate HA binding. The authors must show to what extent HA-binding on (activated) splenic DCs is dependent on CD44.

We used a flow cytometry-based approach, as previously described [e.g., DOI: 10.1126/science.278.5338.672; DOI: 10.1186/1471-2121-5-10], to examine the extent of HA-binding on activated splenic DCs isolated after infection from WT or CD44 KO mice. DCs isolated from either WT or CD44 KO showed no HA binding (**Rebuttal Fig. 1**). [Redacted]

EL4 cells stimulated with PMA and ionomycin served as our positive control (**Rebuttal Fig. 1c**); we used these cells to show that vCD44BP-Fc or the KM114 anti-CD44 mAb prevent CD44 from binding HA (**manuscript Fig. 1f and g**). These results indicate that DCs are unable to bind HA, a finding in agreement with the published literature [DOI: 10.4049/jimmunol.1402506; DOI: 10.3389/fimmu.2015.00150; DOI: 10.1097/00002371-200401000-00001]. These findings further strengthen our conclusions from the in vitro assays and in vivo chimera data, demonstrating that vCD44BP exerts its effects by interfering with CD44 function in stromal fibroblasts.

[Figure Redacted]

Rebuttal Figure 1. Representative histogram overlays showing CD44 expression and HA binding on DCs isolated by mechanical dissociation from the spleens of naïve mice and mice infected with MCMV as indicated (d1 pi). (a) Data are from C57BL/6 and CD44 KO mice. [Redacted] (c) Data are from EL4 cells. In a and b, the histograms are concatenated from 2x naïve, 3-4x infected mice per condition. Data are representative of 2-4 independent experiments.

This relates to the outstanding point concerning the mechanism of how vCD44BP binds stromal cell CD44 to delay antiviral CD8+ T cell responses. The authors deduce that vCD44BP inhibits DC migration by blocking the interaction of FRC-CD44 with DC-HA based on several observations including:

- *Crystal structures of purified vCD44BP and CD44 showing binding to the HA-cleft*
- *In vitro DC migration assays showing a statistical difference in migration when FRCs are pretreated with vCD44BP-FC*
- *In situ visualization of differences in DC accumulation in the T cell zone*

Nevertheless, it remains that this mechanistic deduction is shown in parts (purified molecules, in vitro assays, and functional in vivo readouts), and experimental evidence demonstrating that attenuated DC-migration in vivo is mediated by interfering with FRC-CD44 to DC-HA is lacking.

Our chimera experiments provide crucial in vivo data on the role of CD44 in DC trafficking. Specifically, our studies with stromal CD44KO hosts demonstrated the following:

- in naïve mice, localisation of DCs to the splenic white pulp is reduced in the absence of stromal CD44 (compare pink plots, **Rebuttal Fig. 2**)
- when the HA binding site on CD44 is blocked by vCD44BP (WT MCMV infection), DC migration to the white pulp in WT mice is equivalent to that seen when CD44 is absent from the stroma (green plots, **Rebuttal Fig. 2**)
- in the absence of vCD44BP (Δ vCD44BP infection), infection increases DC migration, and this effect strictly depends on stromal CD44 (compare orange plots, **Rebuttal Fig. 2**).
- in the absence of stromal CD44, infection with either virus (green and orange plots on the right, **Rebuttal Fig. 2**) increases DC migration relative to the naïve controls (pink plot on the right, **Rebuttal Fig. 2**), indicating the presence of additional mechanisms

Rebuttal Figure 2. The mean CD11c staining intensities within the WP of chimeric mice infected with MCMV (green), Δ vCD44BP (orange) at day 2 PI, or naïve (pink) controls are plotted.

These data identify two distinct instances where DC localisation to the splenic white pulp is affected: (i) when CD44 is absent from stromal cells; (ii) when vCD44BP is expressed by MCMV. Since vCD44BP blocks HA binding to activated CD44 (**manuscript Fig. 1f and g**) and occupies the HA binding site in CD44 (**manuscript Fig. 1i-q**), these findings are consistent with vCD44BP having the same effect on WT stroma as eliminating CD44 expression. Additional discussion of the chimera data is provided in our response to Referee 3.

Our results also demonstrate that DC trafficking to the white pulp is reduced, but not abolished, by either vCD44BP or CD44 deficiency, consistent with the contribution of additional interactions to this process, which provides context for the role of the CD44–HA axis.

To provide additional experimental evidence as requested by the Referee would require selective ablation of HA synthesis in DCs, necessitating complex conditional deletion of multiple (3) HA synthetases; these models do not currently exist. Importantly, our data provide definitive evidence

that stromal fibroblastic CD44 is essential for DC trafficking in vivo, and that this process is specifically targeted by vCD44BP, a viral protein that selectively blocks HA binding to CD44. In sum, our findings firmly establish stromal CD44 as a key regulator of DC trafficking and the initiation of T cell responses. Importantly, they reveal a previously unrecognised function of stromal CD44 in the initiation of adaptive immunity, one that is essential but operates in concert with other mechanisms (such as chemokines and the podoplanin-CLEC2 axis) to maintain robustness in a system critical for host survival. By targeting stromal CD44 through vCD44BP, CMV delays but does not abolish T cell responses, an evolutionarily advantageous strategy that enhances viral replication while preserving host survival. This concept is now further highlighted in the revised manuscript (**first marked paragraph on page 14, Discussion**).

A key feature in vivo that cannot be recapitulated in vitro is the lack of a reticular - FRC network in vitro and overall compositional differences in FRC-associated ECM. Particularly, in vivo, HA is integrated in the ECM. Thus, vCD44BP may interfere with FRC-ECM interactions, which may then induce a distinct transcriptional reprogramming of FRCs that indirectly influence DC recruitment (only CCL19 and CCL21 expression have been examined). These limitations must be considered in relation to the conclusions that can be drawn from in vitro DC migration studies (Figure 4 l, m).

While we cannot definitively exclude the possibility that vCD44BP may induce transcriptional reprogramming of FRCs, our data strongly support that direct engagement of CD44 by vCD44BP is the primary mechanism driving its effect on DC trafficking. The principal FRC-derived chemokines driving DC recruitment, CCL19 and CCL21, showed no differences, expression of PDPN (a previously described driver of FRC-DC interactions) was not altered, and there were no changes in FRC numbers or proliferation rate. Importantly, however, vCD44BP alters the intracellular localisation of CD44 and remodels the stromal cell network in vivo. These effects, together with the modulation of DC migration via CD44-HA interactions between stromal cells and DCs, provide a compelling mechanistic rationale for the observed effects. Nevertheless, since we cannot formally exclude a role for ECM-integrated HA, we have noted it as a possibility (**second marked paragraph on page 14, Discussion**).

Additionally, the connection of FRCs to a continuous reticular and cellular network means that there are no filipodia in vivo (presumably this is what is meant by apical surface in reference to Figure 4h). Therefore, it remains unclear whether a redistribution of CD44 on FRCs is at all relevant in vivo.

Considering all the in vivo data presented in the revised manuscript, we had already chosen to move these descriptive findings to the Extended Data Figures. While not central, these data align with previous literature linking protein redistribution to protrusions and filopodia of FRCs in vitro with significant in vivo functionality.

Similarly, it is unclear whether cell morphology changes in an FRC cell line have any relevance in vivo (Figure 4f).

The FRC morphology index data has been moved to **Extended data Fig. 4m**. It is worth noting that multiple high impact studies [e.g. DOI:10.1038/ni.3035; DOI:10.1038/nature13814; DOI: 10.1038/s41590-022-01272-5] link FRC morphology in vitro with in vivo immunological outcomes.

Similarly, whether there is any relevance of FRC morphological differences on the interaction with DCs remain speculative. Indeed, an alternative explanation to the more extended morphology may simply be the increased cellularity observed following infection with vCD44BP-deficient MCMV (Figure 3a, and 5 h, l). Indeed, this resembles the stretching of the FRC network observed in inflamed LNs upon infection but is not necessarily a consequence of altered CD44-HA interaction between

FRCs and DCs. As the relevance of this data remains speculative, Figures 4f, g, j, k should be moved to the extended data).

Although changes in the FRC network may not be the sole, direct and immediate consequence of interactions with DCs, extensive published evidence supports the influence of FRC morphology on DC behaviour and vice versa.

Importantly, the data in Figures 4g, i and j (previously 4j and k) provide the following key insights:

- Fig 4g – demonstrates that CD44 and vCD44BP are colocalised in infected cells, highlighting a spatial interaction that is key to understanding how vCD44BP influences CD44-mediated processes during infection.
- Fig 4i and j – provide important information about changes in the FRC network after infection in the presence and absence of vCD44BP. We were grateful to the Referee for suggesting we undertake this analysis as it revealed differential remodelling of the FRC network in spleens from mice infected with the Δ vCD44BP virus. This FRC network remodelling is critical to generate effective immune responses, and as noted by the Referee, these changes resemble the stretching of the FRC network observed in inflamed LNs upon infection. These changes generally precede lymphocyte influx to secondary lymphoid organs and the accompanying increased cellularity. Indeed, the network analysis presented in Fig 4i and j is from day 2 post-infection when overall cellularity is equivalent in the presence or absence of vCD44BP.
- As noted above, Figure 4f has been moved to the Extended data as suggested.

With these considerations, and without a demonstration that HA in the ECM is irrelevant for DC migration in vitro or that FRC-DC interactions are truly disrupted in vivo, the conclusions must be reworded to imply that this is a possible but not formally demonstrated mechanism.

Please see above.

Minor comments:

i. The "apical distribution" of CD44 remains unclear. The term apical is usually reserved to describe the lumen-facing side of polarized cell types such as epithelial cells. If this refers to the filipodia in vitro, this should be stated, along with the limitation that it remains unclear whether in vivo FRCs have similar structures in vivo.

This has been addressed above.

ii. Representative images for Figure 5d must be provided.

Representative images have been added to **Extended Data Fig. 5f**.

iii. Figures 5i-j: The authors have now added the representative FACS plots for the quantification of OVA tetramer+ cells in the vaccination setting. In the OVA vaccination setting, it is clear that the antigen-specific effect size is minimal (maximally a difference of 0.1% of CD8+ T cells in PBS treatment compared with vCD44BP-Fc treatment). With populations of < 100 cells, a few cells can quickly account for such small differences between conditions. Rather the differences in CD8+ and antigen-specific T cells appear to be driven by overall cellularity and not proportional differences between treatment groups. This dataset should therefore be moved to the extended data.

Remodelling of the FRC network is critical for the generation of effective immune responses, and this includes facilitating the access of lymphocytes into secondary lymphoid organs. We therefore would certainly expect differences in the generation of CD8 T cell responses in the presence of vCD44BP-Fc to be amplified by increases in overall cellularity.

To make these points more clearly, we have moved the OVA vaccination data to Extended Data and put the adjuvant experiments that demonstrate the impact of vCD44BP on lymph node expansion in the main Figures.

iv. For data with a non-normal distribution, the geometric mean rather than mean should be shown. (ie. Fig. 5c).

We thank the Referee for this helpful suggestion. Following consultation with a Biostatistician, we were advised that the use of the arithmetic mean is acceptable, provided that all individual data points are presented to illustrate the distribution. Importantly, the statistical analyses and reported significance values are not affected by this choice. We acknowledge that geometric means are recommended when the data span several orders of magnitude or require a logarithmic transformation for appropriate interpretation. As our cell numbers do not require logarithmic transformation, we trust that presenting the arithmetic mean together with the individual data points is a reasonable option that maintains consistency in the data presentation throughout the manuscript.

v. p values should be provided for all datasets, especially instead of ns so that the reader can judge how close to the 0.05 cutoff the data are.

p values have been added.

vi. The reviewer appreciates that the authors have added Figure 4e to show MCMV infection of PDPN+ cells. However, Fig. 4e must also be provided for MCMV infected splenic FRCs infected with vCD44BP-deficient MCMV, and the proportion of infected PDPN+ cells following WT or vCD44BP-deficient MCMV infection histologically quantified. Indeed, in Fig. 4d, the distribution of viral IE1 in relation to PDPN appears somewhat distinct for the two viruses. This is relevant, as the tropism of vCD44BP-deficient MCMV is not shown in vivo (presumably due to the lack of a vCD44BP-deficient, mCherry recombinant virus).

Representative images for both WT and Δ vCD44BP infected mice have been added to **Fig. 4**, and the quantification of infected PDPN+ cells has been added to **Extended Data Fig. 4**; please see below.

Figure 4. (e) Representative confocal 3D orthogonal views of infected IE1+ (green), PDPN+ FRCs (red) and surrounding CD11c+ DCs (white) in the spleens of MCMV or Δ vCD44BP infected mice at day 2 PI; blue = nuclei. Representative of data from two independent experiments.

Extended Data Figure 4. (g) Quantification of infected FRC in mice infected with MCMV or Δ vCD44BP at day 2 pi.

Referee #3:

In their further revised manuscript, the authors have included data that addresses several earlier concerns. I am surprised by the efficiency of CD44 KO in hematopoietic cells generated by CRISPR RNP treatment of HSC – without a selection step for cells that had received RNPs one might have expected a less complete KO (and thus a more bimodal staining pattern in ED Fig. 5h).

The CRISPR-mediated deletion of *Cd44* was highly efficient due to the use of a dual-guide strategy, please see method on page 29. For each experiment multiple electroporations were performed to generate a sufficient number of cells. Two days after electroporation with *gCd44*, hematopoietic stem cells from individual cultures were pooled and CD44 expression assessed by flow cytometry. As illustrated by the histograms in **Rebuttal Figure 4**, deletion of *Cd44* was almost complete in stem cells treated with *gCd44*. Two independent *gCd44* stem cell pools were generated approximately 1 month apart (**Rebuttal Figure 4A and B**). As shown in **Rebuttal Figure 4** the efficiency of *Cd44* deletion was highly reproducible.

Given the efficiency of this protocol additional purification or selection steps were not required prior to injection of these cells into irradiated recipients.

Rebuttal Figure 4. Histograms showing CD44 expression by hematopoietic stem cells two days after electroporation with *gScr* or *gCd44*. Two independent stem cell pools (**A**) and (**B**) were generated and used to produce two cohorts of chimeric mice. The proportion of live cells within the CD44⁺ gate for cells treated with *gScr* or *gCd44* is shown.

Was this efficiency of KO observed in all recipients in both experiments?

Yes, the efficiency of CD44 KO was similar in all recipient mice in both experiments. In the manuscript, the portion of CD45.2⁺ DCs (i.e. those derived from the donor stem cells) for each individual mouse is plotted (**Extended data Fig 5h**). These data demonstrate that >90% of DCs are donor derived. Representative histograms of CD44 expression by DCs in the various experimental groups are also shown in the manuscript (**Extended data 5i**). When taken together, the data provided in **Extended data Fig 5h and i** illustrate the consistency in the efficiency of KO in our experiments.

As part of the analysis of chimeric mice (**Figs 5e and f in the manuscript**), CD44 expression on DCs was examined in each mouse. The percentage of DCs expressing CD44 in individual mice from the chimera experiments is shown below in **Rebuttal Fig 5**.

Data from mice infected with MCMV, Δ CD44BP (day 7 pi), as well as naïve controls, were pooled for each experimental group. As shown, the majority of DCs in the *gScr* → WT and *gScr* → *Cd44*^{-/-} groups express CD44. Conversely, CD44 expression was virtually absent on DC in the *gCd44* → WT group. This pattern was consistent across mice from the two independent experiments.

Rebuttal Figure 5. Percentages of splenic DCs that express CD44 in all mice used for the chimera experiments (**Figs 5e and f in the manuscript**) are plotted. Expression was assessed in naïve mice and in mice infected with either MCMV or the Δ CD44BP (day 7 pi). Each dot represents an individual mouse.

At what cell density were the purified progenitor cells incubated in StemSpan SFEM II medium?

Progenitor cells were seeded into 24 well plates at 1×10^6 cells per well in a total of 1.5 ml of StemSpan II medium. This information has been added to the Methods section (**page 29 and marked in the revised manuscript**).

Did all mice survive reconstitution with 2×10^5 progenitor cells?

Two independent cohorts of chimeric mice were generated. A total of 60 mice were irradiated and reconstituted with stem cells (treated with either *gScr* or *gCD44*) and of these, 59 survived.

Cohort 1: *gScr* → WT (10/10 mice survived); *gCd44* → WT (10/10 mice survived); *gScr* → *Cd44*^{-/-} (9/10 mice survived).

Cohort 2: *gScr* → WT (10/10 mice survived); *gCd44* → WT (10/10 mice survived); *gScr* → *Cd44*^{-/-} (10/10 mice survived).

The data in Fig 3g suggest that MCMV normally limits cDC migration into the T zone by inhibiting CD44's ability to bind HA. That is, infection with WT MCMV is suggested to inhibit CD44 and reduce cDC migration and this is overcome in the $\Delta vCD44BP$ MCMV infected mice where CD44 function is not blocked. According to this model, the rescued cDC migration in $\Delta vCD44BP$ mice should be lost in CD44 KO mice. Is this the case?

That is correct and the data are shown in **Figure 5d** (the relevant groups are highlighted in orange in the reproduced figure below where $p < 0.0001$). We have revised **Figure 5d** and added all comparisons and p values to highlight the relevant differences and clarify this point. The text has also been amended as follows: “the enhanced migration of DCs into the splenic white pulp observed in TC1 → WT chimeras infected with the $\Delta vCD44BP$ virus (relative to WT MCMV) was lost in TC1 → *Cd44*^{-/-} chimeras (**Fig. 5d and Extended data Fig. 5f**). Indeed, in TC1 → *Cd44*^{-/-} chimeras, DC migration after $\Delta vCD44BP$ infection was comparable to that observed in WT MCMV infection and significantly lower ($p < 0.0001$) than in $\Delta vCD44BP$ infected TC1 → WT chimeras (**Fig. 5d and Extended data Fig. 5f**).” (**page 10 and marked in revised manuscript**).

Figure 5d. The mean CD11c staining intensities within the WP of chimeric mice infected with MCMV, $\Delta vCD44BP$ at day 2 PI or naïve controls are plotted.

Regarding any observations in global CD44KO mice, we have not investigated infection in these mice as they are on a B6 background where the early role of T cells is minimal because of the NK cell-mediated virus control [DOI:10.1038/icb.2009.41].

Referee #4

The crystal structure is technically sound, at high resolution, well refined and supports the conclusions that m11 and HA compete for an overlapping binding site on CD44. The SPR data also appear solid, measuring the interaction at KD 14uM.

It is rather puzzling that both m11 and HA have low affinity for CD44, the concentration of m11 would need to be very high in vivo in order to block HA binding which is on cells and has an entropic advantage. This merits some comment in the text, the numbers would seem to work against m11. Do the authors think the inhibition is partial or incomplete?

As noted by the Referee, both m11 (vCD44BP) and HA have a reported low affinity interaction with CD44 (m11-CD44, Kd 14uM; CD44-HA, Kd 24uM). It is worth noting however that, as shown in **Figure 1**, in vitro m11-Fc inhibits the capacity of HA to interact with CD44 similarly to the HA blocking anti-CD44 antibody KM144, which is likely to have a high affinity interaction with CD44. Thus, there appears to be some advantage to m11 that may be independent of affinity; in support of this, our in vivo data indicate that the CD44-m11 interaction occurs in cis in infected fibroblasts to reduce the interaction between fibroblast CD44 and HA on a second cell (DC) and/or in the extracellular matrix. A pre-existing CD44-m11 cis interaction gives HA less opportunities to bind CD44 irrespective of entropy, and m11 would not need to be at a very high concentration in order to block HA.

Importantly, we have shown that m11 interacts with CD44 in such a way that physically occludes the HA binding site in CD44. Thus, m11 functions as an effective natural inhibitor thereby bypassing the advantages of the natural ligand HA. As suggested by the Referee we have included a comment about this in the revised Discussion (**page 12 and marked in revised manuscript**).

It is important to note that CD44 is just one of the stromal cell-expressed proteins required for DC trafficking. Our data demonstrate that the viral protein m11 effectively interferes with stromal CD44 function to reduce, but not ablate, DC migration to the white pulp, which in turn, diminishes early T cell activation. If m11 completely blocked DC trafficking, it would jeopardise host survival, an outcome that would not be evolutionarily advantageous for the virus. Nonetheless, the targeting of CD44 by m11 has identified CD44 as being essential for the functioning of the fibroblastic stromal cell network and has revealed a previously unrecognised stroma-dependent mechanism crucial for the generation of effective T cell responses.

Dear Dr Weiss,

We have addressed the remaining Referees' comments on our manuscript 2024-01-00034 "**Fibroblastic reticular cells control the initiation of adaptive T cell responses via CD44**" as detailed below.

Response to the Referees' comments.

Referee #1

The reviewer appreciates the added explanation as to why the deletion of HA synthesis in DCs is currently not possible, and appreciates the text added to the discussion adding further insights into the mechanisms of how stromal cells alter DC migration through CD44 - HA interactions.

The reviewer acknowledges that the authors changed the manuscript title from Fibroblastic reticular cells to fibroblastic stromal cells to account for infected fibroblasts within the red pulp. However, as the conceptualized FRC - DC - T cell interaction centers in the white pulp, the reviewer would recommend using the FRC nomenclature for the manuscript title.

As recommended, we have changed the title to use the FRC nomenclature. To comply with the requested character limit, the revised title reads: "Fibroblastic reticular cells direct initiation of T cell responses via CD44."

The reviewer appreciates that the authors have added representative images and quantification of MCMV infected PDPN+ cells in Figure 4. It remains very difficult to visualize that the IE1+ cells in Figure 4e are PDPN+ FRCs - FRCs tend to have an elongated cell body, and when a nucleus is captured in the z-stack, a PDPN signal should surround the nucleus. While the flow cytometric quantification suggests that hematopoietic cells are rarely infected with MCMV, in both MCMV infected mice and in mice infected with the vCD44BP-deficient MCMV, the PDPN signal appears adjacent to an MCMV infected cell rather than outlining infected cells. It is strongly recommended that the authors provide a better visualization of MCMV infected FRC in the splenic white pulp rather than CD45+ hematopoietic cells. This may be as simple as showing the single channel PDPN if the IEL1 signal is simply too strong relative to the PDPN for improved visualization.

We thank the Reviewer for their careful evaluation and suggestions regarding Figure 4e. Visualising IE1+ PDPN+ FRCs can be challenging due to the elongated morphology of FRCs and the strength of the IE1 signal relative to PDPN. We attempted to construct a figure using single-channel images; however, while this improves the visibility of the PDPN signal, it makes it difficult to clearly identify which cells are infected.

It is worth noting that hematopoietic cells are also infected by MCMV after day 1. The focus in Fig 4e is on infection of PDPN+ FRCs, which represent the first cells infected by the virus as shown in Fig 4c.

Referee #3

The authors have adequately addressed my principle concerns.

I remain surprised by the efficiency of the RNP-induced CD44 KO in HSC. Moreover, the amounts of Cas9 protein and sgRNA complex are less than in published studies (e.g., Khoo et al., JOVE

2023) - the details of the protocol should be checked to ensure that others can reproduce these findings (that are impressive if reproducible).

The high efficiency in deleting CD44 in HSCs reported in our study is fully supported by the complete dataset provided in our previous response and included in the manuscript. These data clearly and directly document the editing outcomes.

We note that the Reviewer may not have realised that we used a dual-sgRNA (2-guide) strategy to target CD44. This approach, detailed in the Methods, substantially increases the likelihood of generating disruptive deletions and, in our hands, it improved editing efficiency compared to using each guide individually. To eliminate any possible ambiguity and to facilitate reproducibility, we have now added further methodological details to ensure that others can readily follow the protocol.

We trust that these clarifications resolve any concerns and underscore that the results and methodology are robust and thoroughly documented.

Referee #4

The authors response to my question about relative affinities of M11 versus HA is reasonable, although speculative. The possibility of cis CD44/M11 interaction does increase the effective concentration of m11 enormously and could explain why it effectively inhibits the HA interaction. Authors should cite Borowska et al. Science Immunology, 2024 as a highly related case where a virus has targeted a cell surface phosphatase on immune cells, although using a different mechanism.

The Borowska paper has been added to the references and is cited in the Main text as follows “Viruses hijack cellular functions to their advantage. Over millennia of co-evolution, viruses, and cytomegaloviruses (CMV) in particular, have evolved mechanisms to subvert host immunity, including the expression of proteins that bind and modify the functions of cellular receptors^{10,11}” where ¹¹ is Borowska et al, 2024.